# UncertaINR: Uncertainty Quantification of End-to-End Implicit Neural Representations for Computed Tomography

**Francisca Vasconcelos**[*][†]                                    *francisca@berkeley.edu*
*Department of Electrical Engineering and Computer Science*
*University of California, Berkeley*

**Bobby He**[*]                                                    *bobby.he@stats.ox.ac.uk*
*Department of Statistics*
*University of Oxford*

**Nalini Singh**                                                  *nmsingh@mit.edu*
*Computer Science and Artificial Intelligence Laboratory*
*Massachusetts Institute of Technology*

**Yee Whye Teh**                                                  *y.w.teh@stats.ox.ac.uk*
*Department of Statistics*
*University of Oxford*

**Reviewed on OpenReview:** *https://openreview.net/forum?id=jdGMBgYvfX*

## Abstract

Implicit neural representations (INRs) have achieved impressive results for scene reconstruction and computer graphics, where their performance has primarily been assessed on reconstruction accuracy. As INRs make their way into other domains, where model predictions inform high-stakes decision-making, uncertainty quantification of INR inference is becoming critical. To that end, we study a Bayesian reformulation of INRs, UncertaINR, in the context of computed tomography, and evaluate several Bayesian deep learning implementations in terms of accuracy and calibration. We find that they achieve well-calibrated uncertainty, while retaining accuracy competitive with other classical, INR-based, and CNN-based reconstruction techniques. Contrary to common intuition in the Bayesian deep learning literature, we find that INRs obtain the best calibration with computationally efficient Monte Carlo dropout, outperforming Hamiltonian Monte Carlo and deep ensembles. Moreover, in contrast to the best-performing prior approaches, UncertaINR does not require a large training dataset, but only a handful of validation images.

## 1 Introduction

Implicit neural representations (INRs) are a recent technique for capturing complex, coordinate-based signals, achieving impressive results in novel view synthesis (Mildenhall et al., 2020; Niemeyer et al., 2020; Saito et al., 2019; Sitzmann et al., 2019), shape representation (Chen & Zhang, 2019; Deng et al., 2020; Genova et al., 2019; 2020; Jiang et al., 2020; Park et al., 2019), and texture synthesis (Henzler et al., 2020; Oechsle et al., 2019). In the context of image reconstruction, INRs represent images as continuous functions mapping coordinates to pixel values, $f : (x, y) \rightarrow [0, 1]$. Typically $f$ is parameterized by a small neural network (NN) and trained with gradient-based optimization, given observations $S$. This continuous INR formulation offers several benefits to discrete array representations, e.g. enabling signal processing at arbitrary resolutions (thereby avoiding the so-called "curse of discretization" (Mescheder, 2020)) and improved memory efficiency (Dupont et al., 2021).

---

[*]Equal contribution.
[†]This work was done while the author was at the University of Oxford.

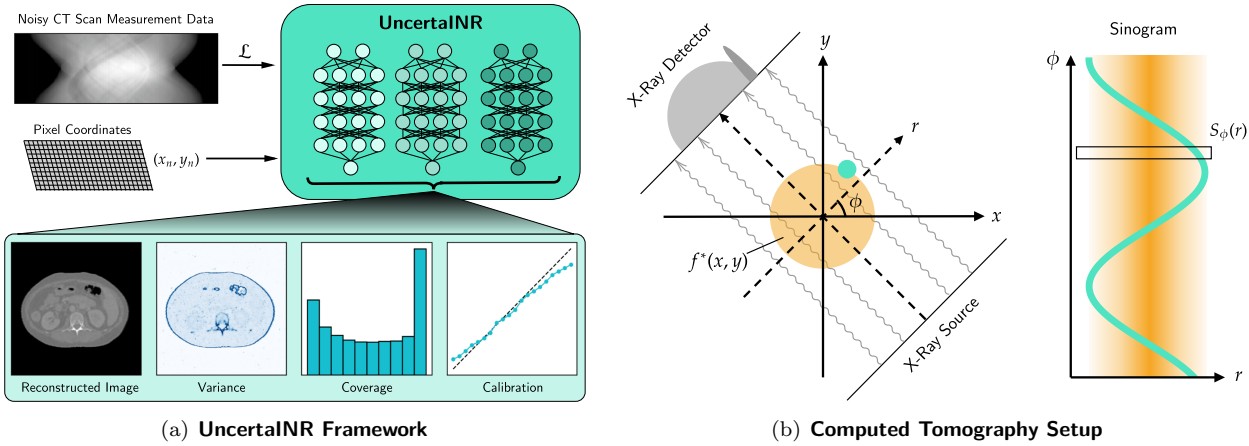

(a) **UncertaINR Framework**    (b) **Computed Tomography Setup**

Figure 1: **(a)** From a noisy CT scan, UncertaINR reconstructs a high-quality image of the 2D imaging subject cross-section. Further, UncertaINR achieves well-calibrated UQ over the reconstructed image, enabling doctors to use the predicted variance to determine regions where the reconstruction may have failed. **(b)** A CT scan of attenuation-coefficient image $f^*(x, y)$ results in sinogram $S_\phi(r)$.

Despite these appealing properties, existing INR applications have primarily focused their scope on either: 1) predictive representation accuracy or 2) visual plausibility of predictions extrapolated from a given training signal $S$. However, there are settings – e.g. Magnetic Resonance Imaging (MRI) and Computed Tomography (CT) (Tancik et al., 2020) – in which INRs have proven promising and for which uncertainty quantification (UQ) is highly desirable. In such applications, only an underdetermined set of observations $S$ is available, which is exacerbated by the high-dimensional, non-convex nature of NN parameter landscapes. As a result, many different INR parameter values may fit the observed data equally well, but yield vastly different predictions (some of which may be poor, e.g. Figure 4 of Tancik et al. (2021)) when extrapolating to unobserved regions of coordinate space. Hence, in such scenarios, calibrated predictive uncertainty is crucial alongside high reconstruction accuracy.

This work assesses the UQ capabilities of INRs in the applied setting of medical imaging, specifically computed tomography (CT). This setting is apt for INR UQ due to its high-stakes nature – even a small image artifact could result in misdiagnosis – and underdetermination of CT sinogram observations. Moreover, UQ could be leveraged to reduce healthcare costs via automated triage, e.g. by assigning images with varying degrees of uncertainty to healthcare providers of relevant expertise. Furthermore, each CT scan measurement exposes the patient to harmful radiation[1], meaning calibrated model uncertainty could enable techniques, such as active learning (Cohn et al., 1996), to inform more efficient measurement collection and reduce negative radiation exposure effects. Although uncertainty has been quantified in deep-learning based medical imaging (Barbano et al., 2021; Laves et al., 2022), medical datasets are often small and apparatus-specific, posing significant generalization challenges for large data-driven approaches (Abramoff et al., 2018; De Fauw et al., 2018; Hosny et al., 2018; Yasaka & Abe, 2018; Zhang et al., 2020). Meanwhile, INRs require little to no training data and have previously demonstrated decent CT reconstruction accuracy (Tancik et al., 2020), which we now complement with well-calibrated UQ.

In light of the aforementioned need for well-calibrated INRs, we propose UncertaINR (Figure 1a): a Bayesian reformulation of INR-based image reconstruction. We demonstrate effective application of Bayesian deep-learning (BDL) principles to INRs, in the underdetermined CT reconstruction context, and study the performance of various established BDL approximate inference approaches: Bayes-by-backprop (BBB) (Blundell et al., 2015), Monte Carlo dropout (MCD) (Gal & Ghahramani, 2016), and deep ensembles (DEs) (Lakshminarayanan et al., 2017), as well as the 'gold-standard' but computationally intensive Hamiltonian Monte Carlo (HMC) (Neal et al., 2011). We find that the simple, yet efficient MCD procedure is highly effective at

---

[1]An estimated 29,000 current or future cancer cases are linked to CT scans performed in the United States of America in 2007 alone (De González et al., 2009)

achieving calibrated INR UQ. MCD outperforms deep ensembles, countering findings in other BDL setups, like image classification – where ensembling is considered a state-of-the-art NN UQ approach (Ovadia et al., 2019). MCD also rivals (sometimes outperforming) HMC, which is far more computationally demanding and difficult to hypertune. Ensembling INRs with MCD achieves further performance gains. Overall, UncertaINR attains well-calibrated uncertainty estimates without sacrificing reconstruction quality relative to other classical, INR-based, and CNN-based reconstruction techniques on realistic, noisy, and underdetermined data.

## 2 Problem Setting and Related Works

### 2.1 CT Image Reconstruction

We are interested in reconstructing 2D images, represented as functions $f(x, y)$ of tissue attenuation coefficients at pixel locations $(x, y)$ within an imaging subject cross-section. As visualized in Figure 1b, CT scanners rotate X-ray emitters and detectors around the subject, collecting measurements of the Radon transform $S_\phi(r)$ of the desired image $f$ rather than actual image values $f(x, y)$,

$$S_\phi(r) = \int \int f(x, y)\, \delta(x \cos \phi + y \sin \phi - r)\, dxdy, \tag{1}$$

for view-angles $\phi$ and X-ray detector radii $r$. $S_\phi(r)$ is also known as a *sinogram* and is not directly human-interpretable. The reconstruction of $f(x, y)$ from $S_\phi(r)$ thus constitutes an inverse problem, governed by the inverse Radon transform. Appendix A further describes the CT measurement physics and image reconstruction problem. While the Projection Slice Theorem (Bracewell, 1956) ensures that $f^*$ can be fully reconstructed from the complete sinogram, practical measurement data is finite and noisy, making the image reconstruction problem underdetermined and reconstruction UQ desireable.

We assume that image $f(x, y)$ is reconstructed on a finite set of pixels $\mathcal{X} \times \mathcal{Y}$ (assumed to be grid-spaced for comparison of INRs with classical grid-based reconstruction techniques), while sinogram measurements are observed on a finite set of view-angles $\Phi$ and X-ray detector radii $\mathcal{R}$. For notational convenience, we denote the $i$th sinogram measurement $S_i$ for $i \in \{1, \ldots, |\Phi \times \mathcal{R}|\}$ and $j$th pixel value $f_j$ for $j \in \{1, \ldots, |\mathcal{X} \times \mathcal{Y}|\}$. In a slight abuse of notation, we also denote resulting vectors of pixel values and sinogram measurements $f$ and $S$, respectively. Thus, the discretization of Eq. 1 is

$$\sum_{j=1}^{|\mathcal{X} \times \mathcal{Y}|} \mathbf{A}_{ij} f_j = S_i, \; \forall i \; ; \quad \text{equivalently,} \quad \mathbf{A}f = S, \tag{2}$$

where the $ij$th entry of the discretized Radon transform matrix, $\mathbf{A}$, represents pixel $j$'s contribution to the $i$th prediction measurement, e.g. $\mathbf{A}_{ij}$ is zero when pixel $j$ is not along the ray measured by $S_i$.

#### 2.1.1 Classical CT Reconstruction Methods

Among classical approaches for CT reconstruction, filtered backprojection (FBP) is one of the simplest analytical reconstruction techniques, providing a simple closed-form estimate of the image. By upweighting high frequencies, FBP enables reconstruction of detailed image structures, but can also emphasize high-frequency noise, resulting in poor image quality. Iterative reconstruction techniques – e.g. the algebraic reconstruction technique (ART) (Gordon et al., 1970), simultaneous iterative reconstruction technique (SIRT) (Gilbert, 1972), simultaneous algebraic reconstruction technique (SART) (Andersen & Kak, 1984), and conjugate gradient for least squares (CGLS) (Yuan & Iusem, 1996) – can mitigate these noise effects.

Another algorithm class frames the reconstruction problem as minimization of a regularized objective,

$$\min_f \|\mathbf{A}f - S\|^2 + \lambda T(f), \tag{3}$$

where $T(f)$ is a regularizer encoding a prior on $f$, with regularization strength $\lambda$. A regularizer typically used in medical imaging is the total-variation (TV),

$$T_{\text{TV}}(f) = \sum_{x,y \in \mathcal{X} \times \mathcal{Y}} \sqrt{\big(f(x+1,y) - f(x,y)\big)^2 + \big(f(x,y+1) - f(x,y)\big)^2}, \tag{4}$$

which removes unwanted image noise and artifacts, while preserving important details such as edges (Rudin et al., 1992). The minimization problem in Eq. 3 with TV regularization is solvable via proximal gradient-based techniques, e.g. the fast iterative shrinkage-thresholding algorithm (FISTA-TV) (Beck & Teboulle, 2009). Alternatively, expectation-maximization (EM) iteratively maximizes the log likelihood of the projections given the estimated image (Dong, 2007). While these regularized methods typically outperform analytic methods, iterative solutions may be computationally slow and their reconstruction quality can be further improved. Appendix B provides detailed descriptions of these classical techniques.

### 2.1.2 Deep-Learning CT Reconstruction Methods

Challenges for classical methods have motivated significant recent interest in NNs trained with large-scale datasets to reconstruct high quality CT images from low-dose acquisitions. Because reconstruction requires only a single NN forward pass, rather than a large number of optimization updates, these methods are faster than purely iterative methods. Some deep-learning (DL) approaches input analytic low-dose reconstructed CT images into an NN trained to directly produce artifact-free reconstructions from higher-dose acquisitions (Chen et al., 2017a;b; Liu & Zhang, 2018; Yang et al., 2018). Alternatively, "unrolled" network architectures (Adler & Öktem, 2018; Jin et al., 2017; Wu et al., 2019) solve Eq. 3 by chaining together NN layers such that each layer computes one optimization update. In this work, we compare our methods to the FBP-Unet (Jin et al., 2017). We also compare to GM-RED (Sun et al., 2021), which uses a deep denoiser trained on the acquired dataset to define a reconstructed image prior lying on a manifold of natural images. We note that while FBP-Unet and GM-RED require large training datasets, our method – leveraging INRs – requires only a few validation images for hyperparameter tuning.

Although DL has enabled advances in CT reconstruction (Lell & Kachelrieß, 2020), progress has been hampered by the *specific*, *expensive*, and *small* nature of CT datasets. This follows from the existence of various CT-imaging modalities (e.g. helical, spiral, electron beam, and perfusion imaging) and need for individual calibration of CT-imaging apparatuses, which compromises the *transferability* of DL models. CT data collection is also *expensive*, requiring long hours on costly machines (exposing patients to harmful radiation) and labels by expert practitioners (making large-scale annotation virtually impossible). In result, CT datasets are small relative to those of other areas, such as ImageNet (Deng et al., 2009), whose size has proven crucial in enabling state-of-the-art DL performance. In light of these challenges and the aforementioned radiation risk, there is clear need for NN methods that achieve high-quality CT image reconstruction from *small datasets* consisting of *few measurements*. Combined with our desire for calibrated UQ, this motivates the study of INRs in our UncertaINR framework.

## 2.2 Implicit Neural Representations

Implicit neural representations (INRs), implemented via NNs, are functions $f_\theta(\cdot)$ mapping coordinates $\boldsymbol{x}$ to a coordinate-wise feature $f_\theta(\boldsymbol{x})$ of interest, with parameters $\theta$ trained to match some observed signal $S$. Due to their general and scalable formulation, INRs have been applied to a wide range of data modalities including: 3D scenes (Sitzmann et al., 2019), voxel grids (Dupont et al., 2022a; Mescheder et al., 2019), video (Li et al., 2021), and audio (Sitzmann et al., 2020). In this work, we focus on UQ of INRs for CT reconstruction of a 2D image $f$, with pixel inputs $\boldsymbol{x} = (x,y) \in \mathbb{R}^2$ and observed sinogram $S$. We leave exploration of more complex data modalities, such as 3D or 4D CT reconstruction, for future work.

In recent years, INRs have grown popular both in computer graphics and unrelated fields like medical imaging (Tancik et al., 2020), arguably inspired by state-of-the-art novel view synthesis results achieved by neural radiance fields (NeRF) (Mildenhall et al., 2020). Since NeRF, INRs have been successfully applied to: high-resolution 3D scenes from unstructured collections of 2D images (Martin-Brualla et al., 2021); scalable large scene view synthesis (Tancik et al., 2022); generative modelling (Dupont et al., 2022b); meta-learning

(Tancik et al., 2021); and image segmentation of medical scans (Khan & Fang, 2022). Meanwhile, several improvements in INR architectures have accompanied these empirical gains, such as random Fourier feature (RFF) (Rahimi & Recht, 2007) encodings (Tancik et al., 2020) and periodic activation functions (Sitzmann et al., 2020), both of which have a tunable frequency hyperparameter $\Omega_0$ that enables INRs to represent high frequency functions. In UncertaINR, we adopted RFF encodings, detailed in Appendices F.3 and F.4, and similarly found $\Omega_0$ critical for decent reconstruction accuracy (Appendix F.5.1). One key challenge for INRs is their long evaluation times. However, recent literature has focused on addressing this challenge, successfully leveraging sparse voxel models (Fridovich-Keil et al., 2022) and multiresolution hash-encodings (Müller et al., 2022) to reduce evaluation times by orders of magnitude.

Recent work has also demonstrated the applicability of INRs to CT image reconstruction. Reed et al. (2021) utilize parametric motion field warped INRs to perform limited view 4D-CT reconstruction of rapidly deforming scenes. Sun et al. (2021) propose Coordinate-based Internal Learning (CoIL) and Zang et al. (2021) propose IntraTomo, which use INRs to boost the performance of classical reconstruction algorithms, such as those discussed in Section 2.1. In CoIL, an INR learns a functional form of the sinogram, receiving sinogram location $(\phi, r)$ as input and outputting projection measurement $S_\phi(r)$. This functional sinogram generates artificial measurements from view angles not included in the original measurement sinogram. The reconstruction algorithm leverages this artificially INR-enlarged measurement set to achieve improved performance reconstructing image $f$, over the same algorithm trained on the original, smaller measurement dataset.

We note that no existing INR works have addressed the aforementioned need for calibrated UQ, motivating our proposed UncertaINR framework.

## 3 UncertaINR: Uncertainty Quantification of INRs for CT

To quantify the uncertainty in reconstructing image $f$, given sinogram measurements $S$, we reformulate the CT reconstruction problem, Eq. 3, as one of Bayesian inference. We assume a Gaussian measurement model,

$$S_i \mid f \overset{\text{ind.}}{\sim} \mathcal{N}(\mathbf{A}_i f, \sigma^2) \quad \forall i \in \{1, \ldots, |\Phi \times \mathcal{R}|\}, \tag{5}$$

where $\sigma^2$ is an assumed known observation noise, and $\mathbf{A}_i$ is the $i$th row of the discretized Radon transform $\mathbf{A}$.[2] Once a prior distribution is placed over $f$, the posterior distribution over $f$ can be computed. The posterior distribution captures both plausible reconstructions of $f$ (e.g. via the posterior mean), as well as the uncertainty over reconstructions (e.g. via the posterior standard deviation).

An important practical consideration is to choose how the image $f$ is parameterized. In the following we compare two alternative parameterizations: a classical grid-based baseline and our INR-based proposal. For each of these parameterizations we will also discuss appropriate priors for $f$.

**Grid-of-Pixels Baseline**  As a baseline, we parameterize the image $f$ using a pixel-wise grid representation; specifically, we take $f \in \mathbb{R}^{|\mathcal{X} \times \mathcal{Y}|}$. A sensible prior, which prefers smoothness in images while allowing for important details such as edges, is $p(f) \propto \exp(-\lambda T_{\text{TV}}(f))$, where $T_{\text{TV}}(f)$ is the total-variation of $f$. In this case the posterior distribution is

$$p(f|S) \propto \exp\left\{-\frac{1}{2\sigma^2}\sum_{i=1}^{|\Phi \times \mathcal{R}|}\left(S_i - \mathbf{A}_i f\right)^2 - \lambda T_{\text{TV}}(f)\right\}. \tag{6}$$

This is a Bayesian extension to the grid-based methods of Section 2.1.1, with Eq. 3 corresponding to the maximum a posteriori (MAP) solution to Eq. 6. In our experiments we compare INR-based inferences to this discretized baseline, which we refer to as *Grid-of-Pixels* (GOP).

**UncertaINR**  Alternatively, in this work, we parameterize $f$ as an INR $f_\theta$ with parameters $\theta \in \mathbb{R}^p$, mapping from pixel coordinates $(x, y)$ to pixel values $f_\theta(x, y)$. A reasonable prior for $\theta$ is an independent and

---

[2]The end-to-end INR approach introduced in Tancik et al. (2020) can thus be viewed as maximum likelihood using Eq. 5's measurement model.

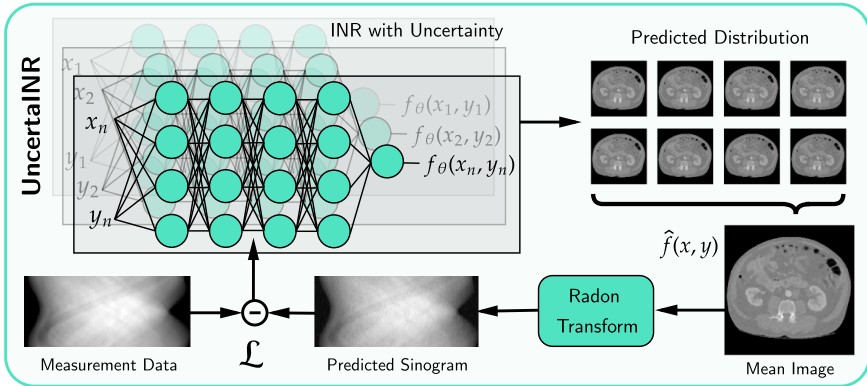

Figure 2: **UncertaINR Architecture**: An end-to-end INR with UQ (BBB, MCD, HMC, and/or DEs) is sampled across all image pixels, generating a distribution of predicted images, $\{f_{\theta_n}\}_{n=1}^{N}$. The predicted sinogram, generated by the Radon transform of predicted mean image $\hat{f}$, is compared to the true measurement data in the INR loss, $\mathcal{L}$. Once training completes, $\hat{f}$ is reported as the reconstructed image and the predictive distribution across INR samples is used to quantify uncertainty.

identically-distributed zero-mean Gaussian prior, $p_{\mathcal{N}}(\theta)$. This gives rise to an implicit prior distribution over functions $f_\theta$. Such a prior is standard in the Bayesian Deep Learning (BDL) literature (Blundell et al., 2015; Graves, 2011), and is well-known to yield a Gaussian Process (GP) limit over $f_\theta$ for wide NNs (Neal, 1996; Matthews et al., 2018; Lee et al., 2018). However, properties of implicit parameter priors are less understood for finite-width NNs, even in standard BDL applications like image classification (due to the uninformative nature of NN parameter spaces), and much less so for CT reconstruction. Thus, we choose a composite prior for UncertaINR,

$$p(\theta) \propto p_{\mathcal{N}}(\theta)p_I(\theta)$$
$$p_I(\theta) = \exp(-\lambda T(f_\theta)) \tag{7}$$

combining $p_{\mathcal{N}}(\theta)$, which constrains the NN parameter values, with $T(f_\theta)$, which imposes a smooth regularization constraint on implicit images $f_\theta$ parameterized by $\theta$. We adopt the common medical imaging practice of using TV regularization, $T = T_{TV}$ (Eq. 4). To the best of our knowledge, this constitutes the first application of TV regularization to INRs, which can be seen in Appendix G.2 Table 7 to noticeably improve UncertaINR reconstruction performance.

**Inference** Our overall framework for UncertaINR is illustrated in Figure 2. Given parameter prior $p(\theta)$, model $f_\theta$, and likelihood model $p(S|f_\theta)$ from Eq. 5, we apply Bayes' rule to $p(\theta|S) \propto p(S|f_\theta)p(\theta)$, deriving the parameter space posterior distribution $p(\theta|S)$. Given a set of sinogram measurements $S$, we ideally seek to sample from the posterior

$$p(\theta|S) \propto \exp\left\{-\frac{1}{2\sigma^2}\sum_{i=1}^{|\Phi\times\mathcal{R}|}\left(S_i - \mathbf{A}_i f_\theta\right)^2 - \frac{1}{2\tau^2}||\theta||_2^2 - \lambda T_{TV}(f_\theta)\right\}, \tag{8}$$

which assigns high probability to images that strike a balance between: 1) a small error between reconstruction $f_\theta$'s Radon transform and observed measurements $S_i$, and 2) low regularization cost under prior $p(\theta)$. $\tau^2$ is a variance hyperparameter for the zero-mean Gaussian parameter prior $p_{\mathcal{N}}(\theta)$, in Eq. 8.

Given $N$ posterior INR parameter samples, $\{\theta_n\}_{n=1}^{N}$, from Eq. 8, we can then use the induced posterior image samples $\{f_{\theta_n}\}_{n=1}^{N}$ to infer both image reconstruction and UQ over the reconstruction. For example, the reconstructed image can be estimated, e.g. by the posterior mean over locations $(x, y)$ as

$$\hat{f}(x,y) = \frac{1}{N}\sum_{n=1}^{N} f_{\theta_n}(x,y), \tag{9}$$

and the posterior predictive uncertainty can be characterized, e.g. through the predictive variance as

$$\hat{\sigma}^2(x, y) = \frac{1}{N-1} \sum_{n=1}^{N} \left( f_{\theta_n}(x, y) - \hat{f}(x, y) \right)^2. \tag{10}$$

This can be used to visualize, as in Figures 1a and 3b, regions of varying model uncertainty and calculate uncertainty metrics such as coverage and calibration (Section 4.1).

**Approximate Inference** Unfortunately, exact inference in BDL is usually intractable – due in part to the high-dimensional and complicated nature of NN posterior landscapes – meaning one must resort to approximate inference methods. Several BDL algorithms have been proposed in the literature, based on approximations with varying levels of sophistication and implementation complexity. In our experiments, we consider four approaches: *Bayes-By-Backprop* (BBB), *Monte Carlo Dropout* (MCD), *Hamiltonian Monte Carlo* (HMC), and *deep ensembles* (DEs). **BBB** is a variational evidence lower bound minimization procedure based on stochastic gradient descent (Blundell et al., 2015). **MCD** uses samples at test-time from NNs trained with dropout (Srivastava et al., 2014) (motivated as a variational approximation of a deep Gaussian process (Gal & Ghahramani, 2016)). **HMC** (Neal et al., 2011) is a gold-standard yet more computationally expensive Markov Chain Monte Carlo (MCMC) procedure leveraging Hamiltonian dynamics (via a time-reversible and volume-preserving integrator) to better explore the full distribution typical set, decreasing consecutive sample correlation and reducing the number of samples required for convergence to the posterior (Betancourt, 2017; Duane et al., 1987). Finally, **DEs** quantify predictive uncertainty by ensembling the outputs of several NN "base learners" trained for the same task with different random seeds (Lakshminarayanan et al., 2017), and is generally regarded as a state-of-the-art approach for UQ in NNs (Ovadia et al., 2019). Despite a non-Bayesian motivation in Lakshminarayanan et al. (2017), the relationship between Bayesian inference and DEs is an active area of research in the BDL community. Wilson & Izmailov (2020) argue that DEs provide a more compelling approximation to the true posterior than many standard BDL approaches, whilst others have adapted DEs to provide a Bayesian interpretation (Ciosek et al., 2019; D'Angelo & Fortuin, 2021; Pearce et al., 2020; He et al., 2020).

**Implementation Details** We note that ensembling can be combined with any of the three prior BDL methods to further improve performance and, in particular, we found that ensembling MCD base learners provides additional gains, in practice, with UncertaINR. Surprisingly, our experiments show that MCD, arguably the simplest BDL approach, produces high-quality reconstructions, outperforming DEs and comparable results to the complex HMC procedure. MCD trains a model $f_\theta$ to minimize the unnormalized version of Eq. 8 with dropout (Srivastava et al., 2014) before every weight layer. The samples $\{f_{\theta_n}\}_n$ of Eqs. 9-10 are obtained by simply performing dropout at inference time, i.e. disabling each internal network node according to a pre-set probability $p$ for different random seeds and applying a forward pass over image coordinates $(x, y)$ to obtain the pixel values $f_{\theta_n}(x, y)$. Hence, the MCD samples from the approximate posterior can be obtained very efficiently after training. For HMC, we used the No-U-Turn-Sampler (Hoffman et al., 2014) sampling scheme in NumPyro (Phan et al., 2019), both for UncertaINR, and also to obtain UQ with GOP. Further details of the approximate inference algorithms we consider are given in Appendix D.

## 4 Experimental Results

Project code is available at: `https://github.com/bobby-he/uncertainr`.

### 4.1 Experimental set up

**Datasets** Two datasets were considered. The first consists of artificial $256 \times 256$ pixel Shepp-Logan phantom (Shepp & Logan, 1974) brain images and was used for ablations. Given the data's simple nature, ablations were performed in the extremely low measurement settings of 5- and 20-view sinograms. We tuned hyperparameters on 5 validation images and evaluated performance on 5 test images. The second dataset, used for UncertaINR baseline comparisons, contains $512 \times 512$ pixel abdominal CT scan images, provided by

the Mayo Clinic for the 2016 Low-Dose CT AAPM Grand Challenge (McCollough et al., 2017). 3 validation images and 8 test images were used to generate noisy 60- and 120-view sinograms[3]

**Performance Metrics** We assessed reconstructed image samples $\{f_{\theta_n}\}_{n=1}^N$ via four metrics: *peak-signal-to-noise ratio* (PSNR), *signal-to-noise ratio* (SNR), *negative log-likelihood* (NLL), and *expected calibration error* (ECE). **PSNR** and **SNR** are common measures of predictive accuracy, whose equations we present in Appendix C. **NLL** is a common probabilistic model quality metric, assessing both predictive accuracy and uncertainty calibration. Under the assumption of an independent Gaussian model, each pixel's averaged prediction $\hat{f}(x, y)$ (Eq. 9) is sampled from a Gaussian distribution with the ground truth pixel value $f^*(x, y)$ as mean and calculated predicted variance of pixel responses, $\hat{\sigma}^2(x, y)$ from Eq. 10, as variance. The NLL is the negative log-likelihood of the pixel values under this model,

$$\text{NLL}(\{f_\theta\}_n, f^*) = \sum_{x,y} \frac{1}{2\hat{\sigma}^2(x,y)} \left(\hat{f}(x, y) - f^*(x, y)\right)^2 + \frac{1}{2} \log(2\pi\hat{\sigma}^2(x,y)).$$

Ideally, the model maximizes the likelihood, meaning NLL is minimized when $\hat{f}(x, y) = f^*(x, y)$ for all $(x, y)$ and variance $\hat{\sigma}^2(x, y)$ is small.

Finally, **ECE** assesses model prediction of its outcome probabilities, gauging reliability of the model's prediction confidence. Specifically, it describes the discrepancy between the *target coverage* (TC) and *achieved coverage* (AC). Given $N$ image samples $\{f_{\theta_n}\}_{n=1}^N$, each pixel $(x, y)$ has an empirical distribution of $N$ predicted values, $\hat{F}_N(x, y) = \{f_{\theta_n}(x, y)\}_{n=1}^N$. Ideally, the median of this distribution is the ground truth pixel value, $f^*(x, y)$. For a given TC $p$ we define,

$$C_{\hat{F}_N, f^*, p}(x, y) = \mathbb{I}\left\{Q_{50-\frac{p}{2}}\left(\hat{F}_N(x, y)\right) \le f^*(x, y) \le Q_{50+\frac{p}{2}}\left(\hat{F}_N(x, y)\right)\right\}, \tag{11}$$

where $Q_k(F)$ denotes the $k$th quantile of distribution $F$. AC is thus defined as the percentage of pixel distributions containing $f^*(x, y)$ in that quantile,

$$\text{AC}(f^*, \hat{F}_N, p) = \frac{1}{|\mathcal{X} \times \mathcal{Y}|} \sum_{x,y} C_{\hat{F}_N, f^*, p}(x, y). \tag{12}$$

If a model is perfectly calibrated, $p\%$ of the reconstructed pixel distributions will contain $f^*(x, y)$ in their $p\%$ quantile, meaning AC=TC for all quantiles. Given a finite set, $\mathcal{P}$, of percentages evenly spaced in $[0, 1]$, ECE is defined as the average difference between AC and TC,

$$\text{ECE}(f^*, \hat{F}_N) = \frac{1}{|\mathcal{P}|} \sum_{p \in \mathcal{P}} |\text{AC}(f^*, \hat{F}_N, p) - p|. \tag{13}$$

A *reliability curve*, as visualized in Figure 3b, plots AC as a function of TC. Better model calibration produces curves similar to the identity function, $\text{AC}(f^*, \hat{F}_N, p) = p$. Furthermore, akin to inverse transform sampling, the marginal distribution of the inverse quantiles of calibrated pixel predictive distributions $\hat{F}_N(x, y)$, at ground truth pixel values $f^*(x, y)$, should be uniformly distributed in $[0, 1]$. We visualize such *coverage histograms* for UncertaINR in Figure 3b. For a further discussion of calibration, coverage, and implementation details, see Appendix C.

## 4.2 Experiments on Artificial (Shepp-Logan) Data

**Ablations** Experiments were performed on the artificial Shepp-Logan dataset, described in Section 4.1, to ablate different UncertaINR hyperparameters across BDL approaches. We ablated the activation function (Tanh, SoftPlus, Sine, SiLU, and ReLU), depth, width, and random Fourier feature (RFF) embedding frequency. For MCD we also assessed sensitivity to the dropout probability $p$, while for BBB we ablated prior standard deviation and KL factor. For the sake of brevity, we only report main findings here (detailed analysis

---

[3]Gaussian noise was added to achieve a 40dB SNR relative to the original, noiseless sinogram.

Table 1: **Ablation Study**: UncertaINR accuracy, relative to classical approaches, and calibration results on the Shepp-Logan phantom dataset. Results are averaged across 5 validation and 5 test images, with the best result for each metric (PSNR, NLL, and ECE) bolded.

| Reconstruction | 5-View Validation Set | | | 5-View Test Set | | | 20-View Validation Set | | | 20-View Test Set | | |
|---|---|---|---|---|---|---|---|---|---|---|---|---|
| Type | PSNR ($\uparrow$) | NLL ($\downarrow$) | ECE ($\downarrow$) | PSNR ($\uparrow$) | NLL ($\downarrow$) | ECE ($\downarrow$) | PSNR ($\uparrow$) | NLL ($\downarrow$) | ECE ($\downarrow$) | PSNR ($\uparrow$) | NLL ($\downarrow$) | ECE ($\downarrow$) |
| FBP | 7.68 | – | – | 5.15 | – | – | 17.35 | – | – | 15.71 | – | – |
| CGLS | 16.38 | – | – | 14.62 | – | – | 21.85 | – | – | 20.82 | – | – |
| EM | 21.39 | – | – | 19.88 | – | – | 30.22 | – | – | 29.11 | – | – |
| SART | 21.12 | – | – | 19.75 | – | – | 31.97 | – | – | 30.45 | – | – |
| SIRT | 21.12 | – | – | 21.12 | – | – | 31.98 | – | – | 30.44 | – | – |
| BBB UINR | 23.26 | -1.190 | 0.152 | 22.52 | 0.138 | 0.203 | 28.25 | 1.650 | **0.121** | 28.16 | 0.562 | 0.119 |
| MCD UINR | 26.15 | -1.473 | 0.111 | 24.45 | -1.572 | 0.083 | 33.74 | 0.701 | 0.135 | 33.08 | 1.093 | 0.113 |
| DE-2 MCD UINR | 26.31 | -1.730 | 0.091 | 24.49 | -1.774 | 0.069 | 33.96 | 0.005 | 0.136 | 33.44 | -0.372 | 0.102 |
| DE-5 MCD UINR | **26.44** | -1.737 | 0.085 | **24.88** | -1.751 | **0.067** | 34.31 | -0.364 | 0.134 | **34.02** | -0.625 | 0.101 |
| DE-10 MCD UINR | 26.36 | **-2.226** | **0.075** | 24.67 | **-1.969** | 0.068 | **34.38** | **-0.529** | 0.131 | 33.86 | **-0.774** | **0.096** |

is provided in Appendix F.5). Activation function and RFF frequency were found to be the most critical hyperparameters. Sine activation produced the best-performing models, but resulting networks were sensitive to hyperparameter choice. SiLU, ReLU, and Tanh achieved slightly lower, but more consistent reconstruction accuracies. In line with recent work demonstrating RFF importance for learning high-frequency image components (Tancik et al., 2020), we found that RFF frequency significantly affected model performance. Specifically, RFF frequency must be consistent with the number of view angles – too low (high) an RFF frequency leads to blurry images (high-frequency image artifacts).

**Model comparison** We also used the Shepp-Logan dataset to compare low-complexity reconstruction and UQ methods – including UncertaINR with BBB, MCD, and DEs of (2, 5, and 10) MCD base learners, as well as classical reconstruction baselines (FBP, CGLS, EM, SART, and SIRT). For the latter, without UQ, only reconstruction accuracy is reported. Several conclusions can be drawn from the experimental results summarized in Table 1, with further analysis on variability and significance presented in Appendix F.7. First, MCD UINRs significantly outperform BBB UINRs, e.g. with PSNR gains up to 5dB (20-view test set) and ECE reductions by 2/3 (5-view test set). Due to this poor performance (more detail in Appendix F.6.1), BBB UINRs were not considered in later experiments. Second, all MCD-based methods significantly outperformed classical methods in terms of reconstruction accuracy, with gains up to ∼4dB PSNR in both test sets. Third, among UncertaINR methods, DE-5 and DE-10 MCD UINRs achieved the best performance, but the simple MCD UINR was surprisingly close (within 1dB PSNR of the best ensemble in all cases). Similar conclusions can be drawn with respect to UQ. Furthermore, despite small validation set size (5 images), UINRs generalized well to the test set. For 20-views, validation and test accuracies were comparable, with improved test set calibration. For 5-views, despite slight PSNR and NLL degradation, ECE improved in the test set. These results suggest that small validation sets are sufficient for tuning UncertaINRs.

### 4.3 Experiments on Real-World (AAPM Grand Challenge) Data

We next compared UncertaINR to state-of-the-art reconstruction approaches, on the real-world AAPM dataset, described in Section 4.1. Results are reported in Table 2, with further analysis on variability and significance in Appendix G.3. UncertaINRs were trained with MCD, DEs of INRs, DEs of MCD UINRs, and HMC – all implemented with TV regularization. More information about dataset and model hyperparameters is given in Appendix G. To understand the effect of UQ on reconstruction accuracy, we implemented our end-to-end INR without UQ ("INR" in Table 2), similar to Tancik et al. (2020)'s proposal. For 60-views, MCD UINRs, DE UINRs and DE MCD UINRs outperformed INRs, whereas HMC UINRs were competitive, but underperformed. Overall, adding UQ did not harm INR reconstruction accuracy.

The lowest block of Table 2 presents results of CNNs trained on large datasets: FBP-UNet, GM-RED[4], and these methods with CoIL. We emphasize that these methods are trained on *large* training datasets, while INRs require only a handful of images for hyperparameter tuning. Thus, the results of these methods

---

[4]Note that, while the COIL work leveraged GM-RED trained with deep image denoisers, RED can be used with denoisers that require less training data.

provide an upper bound on the reconstruction accuracy expected of UncertaINRs, but the two are not directly comparable.

Nevertheless, all (U)INRs achieve competitive performance, reaching accuracies within 1dB (60-views) or 1.5dB (120-views) of the best (large training dataset) CNN. In fact, in the low-measurement regime (60-views), all (U)INRs achieved competitive performance with the highest reconstruction accuracy UINR (DE-5 MCD UINR) – which outperformed all methods, except FBP-UNET with CoIL.

The top 3 blocks of Table 2 present results for methods that can be fairly compared to the (U)INRs: 1) classical FBP, CGLS, EM, SART, and SIRT methods, 2) Sun et al. (2021)'s results for FISTA-TV, FBP with CoIL, and FISTA-TV with CoIL, and 3) TV regularized GOP with and without HMC. Similarly to the previously presented results for synthetic Shepp-Logan data, most classical methods underperform the INRs by more than 5dB on AAPM. Only FISTA-TV is competitive, albeit still more than 1dB away from the INRs. While adding CoIL to FBP and FISTA-TV improves performance, these methods still perform worse than all 60-view (U)INRs. Furthermore, unlike CoIL-based methods, UncertaINR is end-to-end and does not require pre-processing. Finally, we found the discretized GOP instatiation significantly underperformed all (U)INR methods – highlighting the power of INRs relative to classical grid/voxel approaches.

In terms of uncertainty calibration, Table 2's results are surprising. Contrary to common BDL intuition that DEs achieve the best model calibration (Ovadia et al., 2019), we found MCD to be more effective for INR UQ calibration. Specifically, MCD UINRs and DEs of MCD UINRs achieved significantly better model calibration than DEs of INRs without uncertainty. For example, introducing MCD to a DE-10 UINR reduced ECE from 0.144 to 0.045 (60-views) and from 0.162 to 0.053 (120-views). Although increasing ensemble sizes generally improved model calibration, the performance boost was not as significant as that of using MCD. We do not currently

Table 2: **Performance Assessment**: Accuracy and calibration results of all approaches are presented for the AAPM dataset, with noise added to achieve a 40dB SNR sinogram. The table is divided into 4 method types: classical, GOP, INRs, and DL. (*) denotes results taken directly from (Sun et al., 2021). Results are averaged over all 8 test set images and the best result for each metric (SNR, NLL, and ECE) is bolded (excluding *italicized* DL methods, which rely on substantially more training data, constituting an unfair UINR comparison).

| RECONSTRUCTION | 60-VIEWS | | | 120-VIEWS | | |
|---|---|---|---|---|---|---|
| METHOD | SNR | NLL | ECE | SNR | NLL | ECE |
| FBP | 10.58 | – | – | 14.11 | – | – |
| EM | 14.47 | – | – | 15.55 | – | – |
| CGLS | 20.08 | – | – | 21.94 | – | – |
| SIRT | 20.89 | – | – | 21.36 | – | – |
| SART | 21.54 | – | – | 21.77 | – | – |
| FISTA-TV* | 26.08 | – | – | 27.59 | – | – |
| FBP CoIL* | 23.48 | – | – | 24.52 | – | – |
| FISTA-TV CoIL* | 26.95 | – | – | **28.95** | – | – |
| GOP-TV | 25.97 | – | – | 27.40 | – | – |
| HMC GOP-TV | 25.10 | -2.604 | 0.102 | 26.82 | -3.367 | 0.102 |
| INR | 27.25 | – | – | 28.81 | – | – |
| DE-2 UINR | 27.29 | 24.55 | 0.224 | 28.83 | 18.86 | 0.222 |
| DE-5 UINR | 27.30 | 8.427 | 0.176 | 28.83 | 8.804 | 0.183 |
| DE-10 UINR | 27.28 | 6.882 | 0.144 | 28.82 | 6.346 | 0.162 |
| MCD UINR | 27.38 | -3.447 | 0.078 | 28.65 | -3.759 | 0.071 |
| DE-2 MCD UINR | 27.44 | -3.573 | 0.063 | 28.68 | -3.819 | 0.056 |
| DE-5 MCD UINR | **27.48** | -3.660 | 0.051 | 28.70 | -3.876 | **0.043** |
| DE-10 MCD UINR | 27.46 | -3.689 | **0.045** | 28.74 | **-4.090** | 0.053 |
| HMC UINR | 27.10 | **-3.963** | 0.074 | 28.50 | -4.021 | 0.085 |
| *FBP-UNET** | 27.08 | – | – | 29.18 | – | – |
| *GM-RED** | 27.12 | – | – | 29.30 | – | – |
| *FBP-UNET CoIL** | 27.93 | – | – | 29.71 | – | – |
| *GM-RED CoIL** | 27.42 | – | – | 29.79 | – | – |

have a full understanding of why DEs perform poorly relative to MCD for UINRs, in contrast to standard BDL applications. However, in Appendix F.6, we hypothesize that this may be due to the model capacity of an individual ensemble member, which is dictated through the RFF encoding frequency $\Omega_0$.

Furthermore, although HMC is often considered a "golden standard" of approximate Bayesian inference (Izmailov et al., 2021), MCD UINRs achieved better performance than HMC GOP and HMC UINRs. While HMC UINRs achieved NLL competitive with the best DEs of MCD UINRs, they only achieved ECE competitive with the MCD UINR and noticeably lower SNR than all (U)INR approaches. Similar to observations in the Bayesian deep learning literature (Wenzel et al., 2020) and discussed in Appendix G, we found a related cold posterior effect in which modifying posterior temperature enables either improved SNR across HMC samples or improved UQ calibration, but not both. In all, given the large computational overhead in tuning and training HMC relative to MCD, MCD DEs appear to offer the best compromise between computational speed, reconstruction accuracy, and well-calibrated uncertainty.

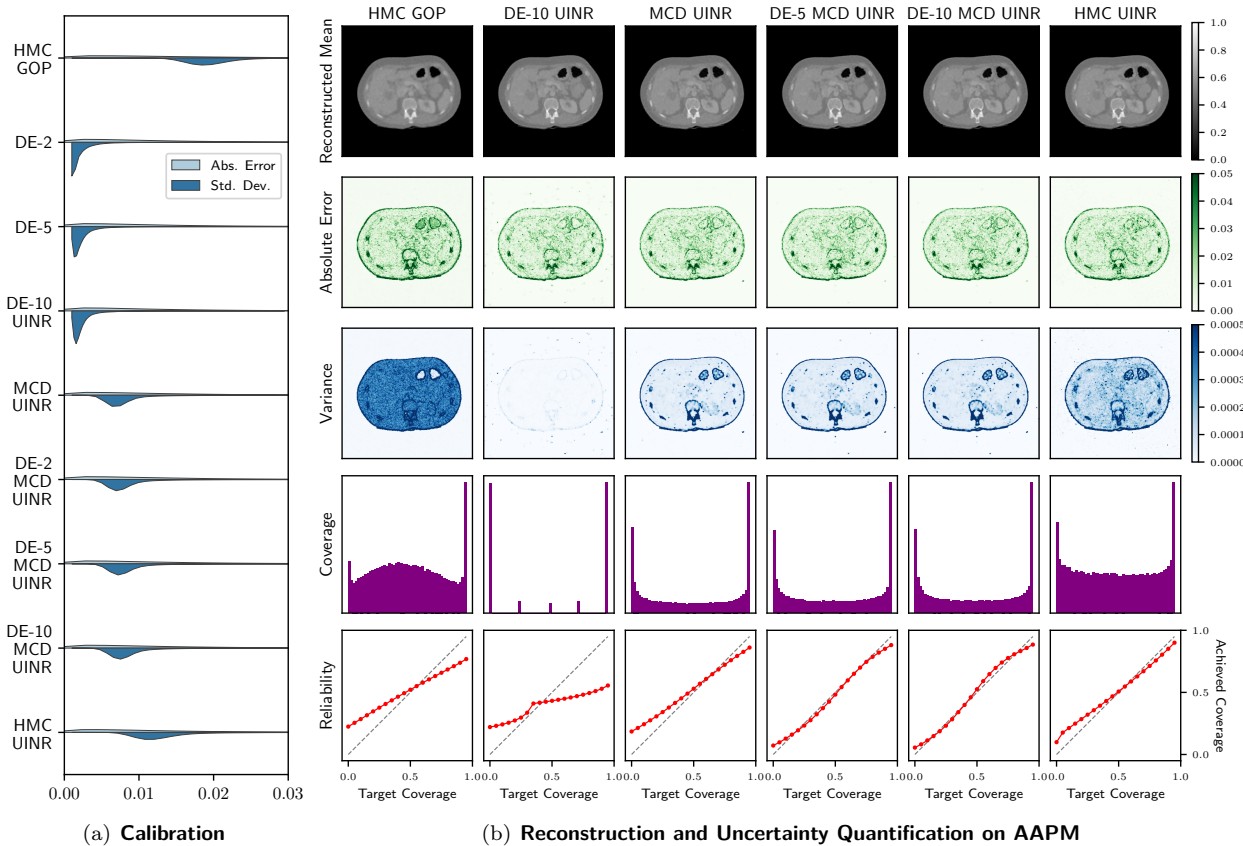

(a) **Calibration**

(b) **Reconstruction and Uncertainty Quantification on AAPM**

Figure 3: For an AAPM test image and the UQ approaches of this work: **(a)** Violin plots of pixel-wise absolute error and predicted standard deviation distributions indicate model calibration. **(b)** Visualizations of the reconstructed mean, absolute error, predicted variance, coverage, and reliability.

The benefits of MCD for INR calibration are further illustrated by the violin plots of Figure 3a, comparing the pixel-wise absolute error versus predicted standard deviation distributions across GOP and UINR models. For non-MCD DEs, the standard deviation distribution skews towards smaller values than the absolute error distribution, indicating that the model is overconfident. The opposite is true for HMC GOP, which is underconfident. Meanwhile, HMC and MCD UINRs predicted standard deviation distributions most closely resembling those of the absolute error, indicating decent calibration.

Finally, Figure 3b visualizes the model output, calibration diagnostics, and uncertainty on a single test image for GOP and the different UINRs proposed in this work. Similar figures are presented for the remaining test images in Appendix G. As reflected in the absolute error images, the GOP reconstructed mean is blurry relative to those of the UINRs. Meanwhile, the reliability curves and coverage plots of both HMC GOP and the DE-10 UINR are quite poor relative to the nearly uniform coverage plots and ideal reliability curves of the (DE) MCD UINRs and HMC UINR. However, these absolute error, coverage, and reliability metrics require the ground truth image to compute and thus would not be available to a doctor.

In real-world scenarios, the only visualizations available are the reconstructed mean and variance images. Given that a well-calibrated model reports larger variance in regions of larger absolute error, variance can be used as a proxy for reconstruction error. For example, the Figure 3b absolute error images show that the models underperform at predicting boundaries between different tissues, which is reflected in corresponding higher uncertainties in the variance images. When presented to a doctor, our UncertaINR variance images could inform more cautious diagnoses based on perceived issues in those regions.

## 5 Limitations, Broader Impact, and Future Work

Despite its decent performance, UncertaINR should not be considered as a replacement for professional medical diagnosis but simply as a tool to aid it. In this work, UncertaINR was thoroughly studied on two datasets, with AAPM data representing a retrospectively-simulated low-dose acquisition. While these experiments offer a promising proof-of-concept of uncertainty quantification for INR-based low-dose CT reconstruction, additional evaluation should be performed on larger, real acquired low-dose CT data before this strategy is deployed in medical settings.

Possible future work includes evaluating UncertaINR in other modalities, such as MRI or 3D/4D imaging settings. In such extensions it would be beneficial to further improve training and memory efficiency of UncertaINR, and to do so UncertaINR could be combined with recent INR-related advances, such as meta-learning (Tancik et al., 2021) and compression (Dupont et al., 2021). Another possibility is to leverage the predictive uncertainty achieved by UINR in an active learning (Cohn et al., 1996) setting, enabling efficient measurement procedures and reducing harmful patient radiation exposure. Finally, our findings raise some fundamental questions about UQ in the INR setting, such as the poor performance of deep ensembles and the effectiveness of MCD, which counter common beliefs in, and should be of interest to, the Bayesian deep learning community.

## 6 Conclusion

In this work, we proposed UncertaINR: a Bayesian reformulation of INR-based image reconstruction. In the high-stakes and well-motivated application of CT image reconstruction, UncertaINR attained calibrated uncertainty estimates without sacrificing reconstruction quality relative to other classical, INR-based, and CNN-based reconstruction techniques on realistic, noisy, and underdetermined data. In the context of INR UQ, contrary to common intuition, we found that simple and efficient MC Dropout rivaled (even outperformed) the popular deep ensembles and the sophisticated, yet computationally-expensive Hamiltonian Monte Carlo methods. UncertaINR's strategic use of INRs outperformed classical reconstruction approaches, while alleviating key challenges faced by state-of-the-art DL methods – namely generalizability and small-scale medical datasets. In addition to informing doctor diagnoses, UncertaINR's well-calibrated uncertainty estimates could pave the way for reduced healthcare costs, via methods like automated triage, and reduced patient radiation exposure, via methods like active learning.

## 7 Acknowledgements

Francisca Vasconcelos primarily carried out this work at the University of Oxford, while supported by the Rhodes Trust via a Rhodes Scholarship. She is now at UC Berkeley, supported by an NSF Graduate Research Fellowship under grant number DGE 2146752. Bobby He is supported by the EPSRC and MRC through the OxWaSP CDT programme (EP/L016710/1). Nalini Singh is supported by a Google PhD Fellowship.

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

# 8 Appendices

## A Computed Tomography

Diagnostic X-rays constitute the largest man-made source of radiation exposure to the general population (de Gonzalez & Darby, 2004; Picano, 2004). In 2010, 5 billion medical imaging studies were performed worldwide, two-thirds of which employed ionizing radiation (Roobottom et al., 2010), and the use of radiology has only grown since (Lin, 2010). Computed tomography (CT), also known as computed axial/assisted tomography (CAT), is a noninvasive medical imaging technique frequently used in radiology to generate detailed images of the body and comprises the majority of this radiation exposure (Brenner & Hall, 2007), with an estimated 29,000 current or future cancer cases linked to CT scans performed in the United States of America in 2007 alone (De González et al., 2009). Since its original development in the 1970s, CT has become widespread in medical imaging – with over 70 million CT scans taken and reported annually in the United States, since 2007 (Smith-Bindman et al., 2009). There are multiple types of CT scanners, such as spiral CT, electron beam CT, and CT perfusion imaging. In this work, we focus on spiral, also know as spinning tube or helical, CT.

In spiral CT, illustrated in Fig. 4, the patient lies along the central axis of a cylindrical measurement tube. As the scan is performed, an X-ray generator rotates around the patient while moving along the axis of measurement. X-rays are emitted, which pass through the patient and are attenuated at various rates by the different types of tissues in the body, as described in Appendix A.1. After exiting the body, the attenuated X-rays are measured by X-ray detectors positioned and moving opposite to the X-ray source. These measurements are used to create a sinogram, as described in Section A.2, which is not understandable by doctors. This sinogram is then input to a reconstruction algorithm, which solves an under-determined inverse problem, described in Section A.3, to generate a human-understandable 2D or 3D image of the organ of interest. This image can then be used by the doctor for medical diagnosis.

### A.1 X-Ray Attenuation

X-rays produced by CT scanners can interact with matter via the photoelectric effect, the Compton effect, and coherent scattering. Through these interactions, some of the emitted X-ray photons are absorbed or scattered when passing through different tissues in the body. The attenuation is described by the Beer-

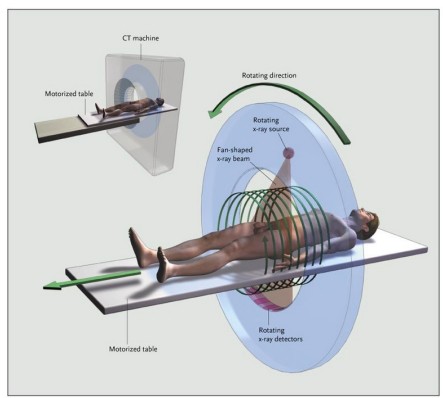

Figure 4: Illustration of a CT scanning device. Reproduced with permission from (Brenner & Hall, 2007), Copyright Massachusetts Medical Society.

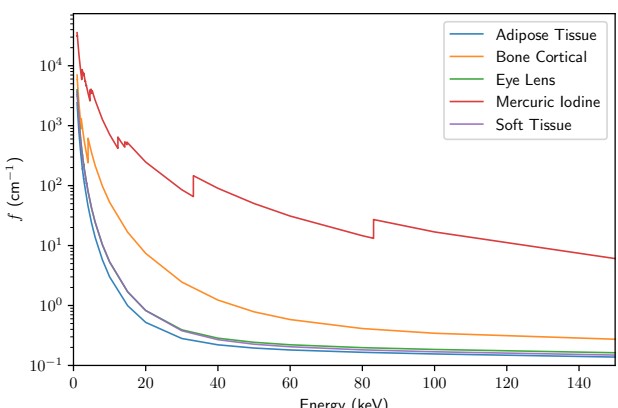

Figure 5: The attenuation coefficient ($f$) of different tissues found in the body, as a function of energy (keV). The plot was generated using X-ray mass attenuation coefficients ($f/\rho$) and densities ($\rho$) from the NIST Standard Reference Database (Hubbell & Seltzer, 2004; Berger et al., 2004).

Lambert Law,

$$J = J_0 e^{-fL}, \tag{14}$$

where $J$ and $J_0$ are the incident and transmitted X-ray intensities; $L$ is the material thickness; and $f$ is the linear attenuation coefficient of the material,

$$f = \tau_1 + \tau_2 + \tau_3, \tag{15}$$

with photoelectric ($\tau_1$), Compton ($\tau_2$), and coherent scattering ($\tau_3$) attenuation coefficients. Attenuation coefficients for common materials in the body – iodine, bone, water, and soft-tissue – are plotted, in Fig. 5, over the range of incident X-ray energies used in CT imaging. It is clear that the attenuation of X-ray photons can be used to distinguish and, thus, image various tissues in the body (Hsieh, 2003).

### A.2 CT Projection Measurements

As previously described, the CT scan relies on an X-ray generator which rotates around the patient, emitting X-ray photons. In this work, we only consider a restricted case of the spiral CT setup, in which there is no motion along the patient axis. Instead, we focus on reconstructing singular 2D image cross-sections and assume a parallel-beam geometry, in which photons are emitted and detected with the linear geometry of Figure 1b.

We begin by considering the measurements of a single detector, measuring at angle $\phi$. Assume that the X-ray generator outputs monoenergetic X-rays of intensity $J_0$. If the patient were simply a homogeneous block of tissue, with length $\Delta\ell$ and attenuation coefficient $f^*$, we could directly apply the Beer-Lambert law (Eq. 14),

$$J = J_0 e^{-f^* \Delta\ell}, \tag{16}$$

to solve for the output attenuated X-ray intensity $J$. In reality, several blocks of tissue will be present in the patient, each with its own attenuation coefficient. However, since the exit X-ray flux from one block of tissue is the entrance X-ray flux to its neighboring block, we can simply apply the Beer-Lambert law in a cascading fashion over intervals of length $\Delta\ell$ and attenuation coefficients $(f_1^*, f_2^*, ..., f_n^*)$,

$$J = J_0 e^{-f_1^* \Delta\ell} e^{-f_2^* \Delta\ell} ... e^{-f_n^* \Delta\ell} = J_0 e^{-\sum_{i=1}^{n} f_i^* \Delta\ell}. \tag{17}$$

As $\Delta\ell \to 0$, the summation term becomes an integration over the length, $L$, of the patient,

$$J = J_0 e^{-\int_L f^*(\ell) d\ell}. \tag{18}$$

Finally, dividing both sides of the expression by $J_0$ and taking a negative logarithm, we define the projection measurement term,

$$S_\phi = -\ln\left(\frac{J}{J_0}\right) = \int_L f^*(\ell) d\ell. \tag{19}$$

As illustrated in Figure 1b, a CT scanner with a parallel beam geometry contains several detectors side-by-side, collectively measuring a plane of attenuated X-ray photons. In this 2D imaging space, the projection measurement becomes a function of detector position, $r$. Thus, Eq. 19 is re-expressed as the line integral,

$$S_\phi(r) = -\ln\left(\frac{J}{J_0}\right) = \int f^*(\phi, r) ds, \tag{20}$$

known as a forward-projection (FP). It follows from the coordinate system of Figure 1b that, for measurements at angle $\phi$, point $(x, y)$ within the patient cross-section is projected onto detector position

$$r' = x \cos\phi + y \sin\phi. \tag{21}$$

Combining this with Eq. 20, we derive the Radon transform (Deans, 2007) of the patient cross-section,

$$S_\phi(r) = -\ln\left(\frac{J}{J_0}\right) \tag{22}$$

$$= \int_{\mathcal{Y}} \int_{\mathcal{X}} f(x, y) \, \delta(x \cos\phi + y \sin\phi - r) \, dx dy,$$

where $\delta$ is the Dirac delta function and $\mathcal{X} \times \mathcal{Y}$ is the set of image pixels $(x, y)$. Sweeping over all the angles, these projective measurements are stacked to form a Radon transform image. As depicted in Figure 1b, the projection measurements of the blue point across several angles produces a sinusoidal curve. The representation of all the CT scan measurements is thus known as a sinogram.

### A.3  CT Image Reconstruction Problem

Sinograms are not human-interpretable. They depict the integrated attenuation coefficients, or projection measurements $(S_\phi(r))$, of the patient cross-section from several angles $(\phi)$ over all detector positions $(r)$. Instead, the desired outcome of a CT scan is a reconstructed image of the cross-section itself. This corresponds to the attenuation coefficient function, $f(x, y)$, which is the inverse of the Radon transform of Eq. 22, or

$$f(x, y) = \frac{1}{2\pi} \int_0^\pi u_\phi(x \cos \phi + y \sin \phi) \, d\phi, \tag{23}$$

where $u_\phi$ is the derivative of the Hilbert transform of $S_\phi(r)$ (Helgason, 1984). The Projection-Slice theorem (Bracewell, 1956) ensures that $f^*$ can be fully reconstructed with infinite measurement angles, $\phi$. In practice, however, it is not possible to acquire infinite measurements. Typically, reconstruction quality improves with number of measurements, but this increases radiation exposure. In practice, hundreds of measurements are performed in a CT scan, but there is interest in reducing this number. In this work, we study algorithm performance in the very low measurement data regime, where uncertainty quantification over the value of $f(x, y)$ becomes especially important. The combination of limited and noisy real-world data renders the reconstruction of the desired image much more complex than simply evaluating the integral of Eq. 23. Leveraging assumptions or data-driven insights about the measurements and physics at play, several statistical models have been developed and used to derive various image reconstruction algorithms, as discussed in the next section.

# B  Classical Reconstruction Algorithms

In this appendix, we provide brief descriptions of the classical approaches implemented in this work via the TomoPy Astra (Pelt et al., 2016) software package. These algorithms are clinically approved and widely used in medical imaging, serving as a basis of comparison for the methods developed in this work. Note that although our approach used NNs, it was not a data-driven approach, but rather learned image functions from small amounts of data. Thus, we do not discuss data-driven techniques. Further, note that we did not implement the FISTA-TV algorithm, which was also used as a classical baseline, but cited results from the CoIL work (Sun et al., 2021), where a description of the algorithm can be found.

## B.1  Filtered Back-Projection (FBP)

Filtered back-projection (FBP) (Pan et al., 2009) is an analytic algorithm which calculates a stable, discretized version of the inverse Radon transform. As the name implies, there are two key steps: filtering and back-projection.

The forward-projection of Eq. 20 describes how X-rays passing through the object domain create a measurement. In back-projection (BP), this measurement is integrated back along the X-ray path across the object domain. This is done over all projection angles $\phi$, using

$$f_{\text{BP}}(x, y) = \int S_\phi(x \cos \phi + y \sin \phi) \, d\phi \tag{24}$$

to reconstruct the object attenuation coefficient image. As the number of projection angles increases, the image reconstruction improves. However, as shown in Figure 6, this back-projection is insufficient to guarantee a clear image. While information about the low frequencies of the object are captured in measurements at several view angles, that of high frequencies may only be captured in a few view-angles. Thus, the low frequencies are sampled far more densely than the higher frequencies, resulting in a blurry image. This can be corrected by suppressing the lower frequencies with filtering, by applying to each projective measurement, $S_\phi(r)$, the sequence of a Fourier transform (FFT), high-pass filter, and an inverse Fourier transform (iFFT). While several high-pass filters can be used, a popular choice is the Ram-Lak filter, which generates the filtered projective measurement

$$\tilde{S}_\phi(r) = \int \mathcal{F}[S_\phi](\omega)|\omega|e^{i2\pi\omega r} d\omega, \tag{25}$$

where $\mathcal{F}[S_\phi](\omega)$ is the Fourier transform of $S_\phi(r)$ and $|\omega|$ the frequency response of the filter. Performing back-projections of all the filtered projective measurements,

$$f_{\text{FBP}}(x, y) = \int \tilde{S}_\phi(x \cos \phi + y \sin \phi) \, d\phi, \tag{26}$$

results in a sharper object attenuation coefficient image. Figure 6 visualizes the difference in reconstruction performance of BP and FBP for increasing measurement view angles, $\phi$. Although the analytic FBP algorithm is fast and numerically stable, it suffers from poor resolution-noise trade-off.

## B.2  Algebraic and Iterative Reconstruction

The reconstruction problem can be formulated as a system of linear equations

$$\mathbf{W}\vec{f} = \vec{S}, \tag{27}$$

where $\vec{S}$ is an $m \times 1$ vector of the $m$ projective measurement values in the sinogram; $\vec{f}$ is an $n \times 1$ vector of the $n$ attenuation coefficient pixel values in the reconstruction image; and $\mathbf{W}$ is a, typically sparse, $m \times n$ weight matrix representing the contribution of each of the $m$ sinogram values to each of the $n$ image pixel values. Given the vector $\vec{S}$, the goal is to solve for $\vec{f}$. If $\mathbf{W}$ were invertible, $\vec{f}$ would simply be $\mathbf{W}^{-1}\vec{S}$. However, because $n$ is usually much larger than $m$, the system of equations of Eq. 27 is underconstrained. In algebraic

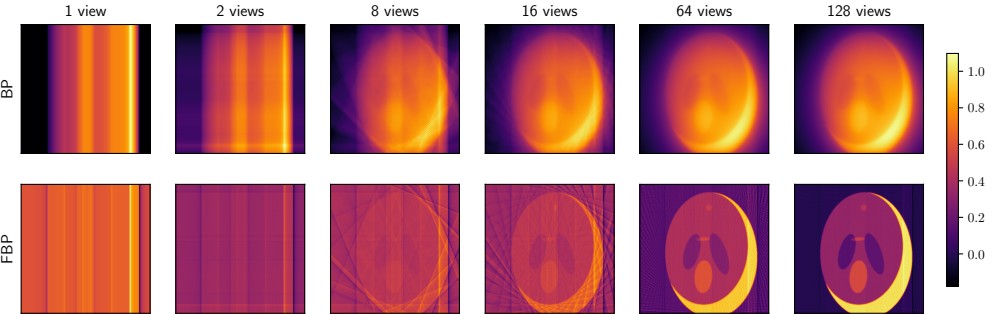

Figure 6: Comparison of the reconstruction quality, as a function of the number of views, of the BP (top) and FBP (bottom) algorithms.

reconstruction, this problem is addressed by using iterative algorithms that pose the reconstruction of $\vec{f}$ as the solution of a constrained optimization problem,

$$\vec{f}^* = \arg\min_{\vec{f}} || \vec{S} - \mathbf{W}\vec{f}||, \text{ subject to } f_i \geq 0 \ \forall i. \tag{28}$$

Several families of iterative solvers can be used to solve this optimization, such as Landweber, Krylov subspaces, and expectation maximization (EM). The key benefit of iterative methods is that prior system knowledge can be integrated, via the cost function and initialization of $\mathbf{W}$. Their down-side is that they are not necessarily stable, may not converge, and are much slower than analytic techniques, such as FBP.

### B.3  Simultaneous Iterative Reconstruction Technique (SIRT)

The simultaneous iterative reconstruction technique (SIRT) (Pryse et al., 1993; Bust & Mitchell, 2008) is a Landweber iterative method that updates the image reconstruction using all available sinogram projection data, $\vec{S}$, simultaneously. The optimization update at step $k$ is defined as

$$\vec{f}^{(k+1)} = \vec{f}^{(k)} + \mathbf{B}\mathbf{W}^T\mathbf{D}\big( \vec{S} - \mathbf{W}\vec{f}^{(k)} \big), \tag{29}$$

where $\mathbf{D} \in \mathbb{D}^{m \times m}$ is a diagonal matrix containing the inverse row sums, $d_{ii} = (\sum_{j=0}^{n-1} w_{ij})^{-1}$, and $\mathbf{B} \in \mathbb{R}^{n \times n}$ is a diagonal matrix containing the inverse column sums, $b_{ii} = (\sum_{i=0}^{m-1} w_{ij})^{-1}$. The weighted projection difference, $\mathbf{D}\big( \vec{S} - \mathbf{W}\vec{f}^{(k)} \big)$, corresponds to the inverse of the length each X-rays passes through the volume. Shorter rays have a higher contribution, with the weighting required to guarantee convergence. This difference is forward-passed back to the image domain, using the weighted back-projection term, $\mathbf{B}\mathbf{W}^T$, where it can be used to update the reconstruction. These updates iteratively solve the problem

$$\vec{f}^* = \arg\min_{\vec{f}} || \vec{S} - \mathbf{W}\vec{f}||_{\mathbf{D}} = \arg\min_{\vec{f}} \big( \vec{S} - \mathbf{W}\vec{f} \big)^T \mathbf{D} \big( \vec{S} - \mathbf{W}\vec{f} \big), \tag{30}$$

converging to a weighted least-squares solution, with weights given by the inverse row sums of $\mathbf{W}$.

### B.4  Simultaneous Algebraic Reconstruction Technique (SART)

The algebraic reconstruction technique (ART) (Gordon et al., 1970) was one of the first proposed algebraic iterative algorithms for CT image reconstruction. It is a Landweber technique almost identical to the SIRT algorithm. However, a single projective measurement is used to update the reconstruction image per update step. Generally, the ART algorithm reaches a solution much faster than SIRT, but does not have stable convergence if the system of equations is inconsistent, for example due to measurement noise.

The simultaneous algebraic reconstruction technique (SART) (Andersen & Kak, 1984) was proposed in 1984, as an improvement to ART, and is also a Landweber algebraic iterative algorithm. It combines the reduced

runtimes of ART with the improved convergence of SIRT, by using all the projective measurements from a single view angle in each optimization iteration. The update of image vector index $i$ at step $n$ is defined as

$$f_i^{(n+1)} = f_i^{(n)} + \frac{\lambda_n}{\sum_{j=0}^{n-1} w_{ij}} \sum_{j=\phi_n L+1}^{\phi_n L+L} w_{ij} \frac{S_j - \hat{S}_j}{\sum_{g=0}^{m-1} w_{gj}}, \tag{31}$$

where $\lambda_n << 1$ is a, potentially dynamic, relaxation parameter; $\phi_n$ is the $(n \mod N)^{\text{th}}$ measurement angle of the sinogram, assuming $N$ total measurement angles; and $L$ is the number of projective measurements taken at each angle. SART typically converges to a good reconstruction within a few iterations.

## B.5    Conjugate Gradient Least Squares (CGLS)

The conjugate gradient least squares (CGLS) (Yuan & Iusem, 1996) algorithm is a Krylov subspace iterative method. Since it requires a positive-definite system matrix, the CT image reconstruction problem is reformulated in terms of the set of normal equations

$$\mathbf{W}^T \mathbf{W} \vec{f} = \mathbf{W}^T \vec{S}. \tag{32}$$

Due to the positive-definiteness of $\mathbf{W}^T \mathbf{W}$, there exists a set of conjugate normal vectors $\mathcal{Q} = \{\vec{q}_1, ..., \vec{q}_n\}$, where $\vec{q}_i^T \mathbf{W}^T \mathbf{W} \vec{q}_j = 0, \forall i \neq j \in (1, n)$. Since $\mathcal{Q}$ forms a basis for $\mathbb{R}^n$, the image vector $\vec{f}$ can be rexpressed as a linear combination of these conjugate normal vectors,

$$\vec{f} = \sum_{i=1}^{n} \alpha_i \vec{q}_i. \tag{33}$$

Thus, solving for $\vec{f}$ becomes a problem of solving for the conjugate normal basis vector directions, $\vec{q}_i$, and their corresponding weights, $\alpha_i$. This can be achieved iteratively by expressing the problem as a quadratic least-squares minimization of the function

$$L(\vec{f}) = \frac{1}{2} \vec{f}^T \mathbf{W}^T \mathbf{W} \vec{f} - \vec{f}^T \mathbf{W}^T \vec{S}, \tag{34}$$

which has gradient $\nabla L(\vec{f}) = \mathbf{W}^T \mathbf{W} \vec{f} - \mathbf{W}^T \vec{S}$ and a guaranteed unique minimizer because the Hessian $\nabla^2 L(\vec{f}) = \mathbf{W}^T \mathbf{W}$ is symmetric positive-definite. The name conjugate gradient least squares comes from the fact that, in each iteration, a conjugate basis vector and its weight are found by taking a gradient step in the direction that minimizes the least-squares function, $L(\vec{f})$, as

$$\vec{f}^{(k+1)} = \vec{f}^{(k)} + \alpha_k \vec{q}_k \tag{35}$$

$$\vec{e}_k = \mathbf{W}^T \vec{S} - \mathbf{W}^T W \vec{f}^{(k)} \tag{36}$$

$$\vec{q}_k = \vec{e}_k - \sum_{i<k} \frac{\vec{q}_i^T \mathbf{W}^T \mathbf{W} \vec{e}_k}{\vec{q}_i^T \mathbf{W}^T \mathbf{W} \vec{q}_i} \vec{q}_i \tag{37}$$

$$\alpha_k = \frac{\vec{q}_k^T \vec{e}_k}{\vec{q}_k^T \mathbf{W}^T \mathbf{W} \vec{q}_k} \tag{38}$$

where $\vec{e}_k$ is the residual at step $k$. Thus, the main difference between SIRT/SART and CGLS is that the search direction in SIRT/SART is determined only by the projection difference at that point, while in CGLS the search directions of all the previous iterations are also taken into account. CGLS typically converges much faster than SIRT, but has a large memory footprint.

## B.6    Expectation Maximization (EM)

The final classical approach to CT image reconstruction that we consider is a statistical iterative method known as expectation maximization (EM) (Dong, 2007). This technique explicitly encodes prior knowledge

about the X-ray physics at hand. Each projective measurement, $S_j$ is modeled as a Poisson distribution,

$$S_j \sim \mathcal{S}_j = \text{Poisson}(\lambda_j) = \frac{\lambda_j^{S_j} e^{-\lambda_j}}{p_j!}, \tag{39}$$

where the distribution mean $\lambda_j = \mathbb{E}[\mathcal{S}_j]$ is the function

$$\lambda_j = \sum_i w_{ij} f_i \tag{40}$$

of the probability $w_{ij}$ that an X-ray photon penetrating image pixel $i$ was measured at detector location $j$; and the underlying attenuation coefficient function $f$ to reconstruct.

The measurement sinogram is modeled as the likelihood

$$p(\vec{S}|\vec{f}) = \prod_j \frac{\lambda_j^{S_j} e^{-\lambda_j}}{S_j!} = \prod_j \frac{(\sum_i w_{ij} f_i)^{S_j} e^{-(\sum_i w_{ij} f_i)}}{S_j!}. \tag{41}$$

The EM algorithm computes the maximum likelihood estimate of $f$,

$$\hat{f}_{\text{MLE}} = \underset{f}{\text{argmax}}\left\{ \log\left(p(S|f)\right) \right\}, \tag{42}$$

by alternating between expectation and maximization steps. These can be combined into the update-step

$$\hat{f}_i^{(k+1)} = \frac{\hat{f}_i^{(k)}}{\sum_j w_{ij}} \sum_j \frac{w_{ij} S_j}{\sum_i w_{ij} \hat{f}_i^{(k)}}. \tag{43}$$

The EM algorithm is computationally intensive, but guaranteed to converge to a local optimum of the likelihood. Further, although a Poisson distribution was assumed for $S_j$ in this discussion, further knowledge of the detector noise can be easily incorporated into the model.

### B.7 Performance of Classical Reconstruction Techniques on Shepp-Logan

Finally, we evaluate the performance of these classical reconstruction algorithms on the Shepp-Logan validation set, depicted in Figure 12. Figure 7 shows average PSNR (for the 5 validation images) as a function of view angle, for each of the 5 reconstruction methods. Also shown are the values below which PSNR is usually considered unacceptable (20dB), at which reconstruction is considered lossy (30dB), and above which has high quality (40dB). FBP has the worst performance, which is particularly poor in the low-view regime ($< 30$ views). The iterative reconstruction algorithms perform better. CGLS has slow convergence to high quality image reconstruction. EM, SIRT, and SART all converge with far fewer views, achieving lossy compression with only $\sim$20 views. EM levels out and requires around 180 views to achieve high-quality reconstruction. SIRT and SART have near identical performance, passing the high-quality reconstruction threshold with only $\sim$75 views and SIRT performing slightly better for larger view numbers.

Table 3: Classical reconstruction PSNR, averaged across the five validation set images, in the 5-, 20-, and 180-view cases. The best achieved PSNR for each view-# is bolded.

| # Views | FBP | CGLS | EM | SART | SIRT |
|---:|---:|---:|---:|---:|---:|
| 5 | 7.68 | 16.38 | **21.39** | 21.12 | 21.12 |
| 20 | 17.35 | 21.85 | 30.22 | **31.98** | 31.97 |
| 180 | 36.74 | 38.6 | 40.46 | 42.51 | **42.76** |

Table 3 reports the PSNR values obtained for 5, 20, and 180 views. EM is the best performer for 5 views, SART for 20, and SIRT for 180. However, while the performance of the three algorithms is similar for 5,

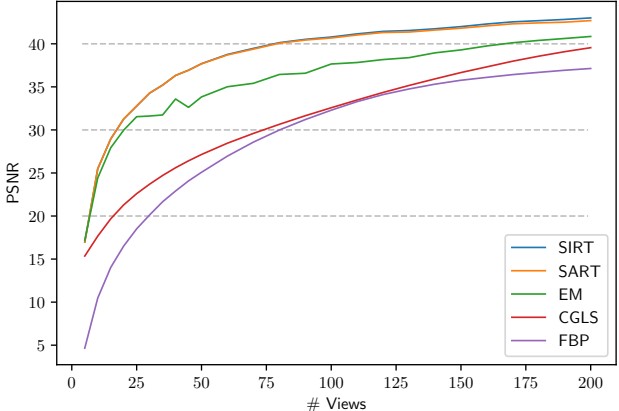

Figure 7: PSNR as a function of number of views for the classical reconstruction algorithms.

SART and SIRT perform better than EM for larger number of views. In the low-view regime, roughly an order of magnitude is required to achieve a 10dB improvement in average PSNR. Figure 8 shows a reconstructed image for each of these algorithm-view combinations. With 5 views the reconstruction algorithms are able to capture low-frequency object structure, but the image would not be useful for medical diagnosis. With 20 views it is clear that the algorithms are already capturing high-frequency components of the object, but the images have many artifacts. By 180 views, the reconstructed images are nearly identical to the ground truth images of Figure 12, with any discrepancies in PSNR due mostly to minor image reconstruction artifacts or imprecisions.

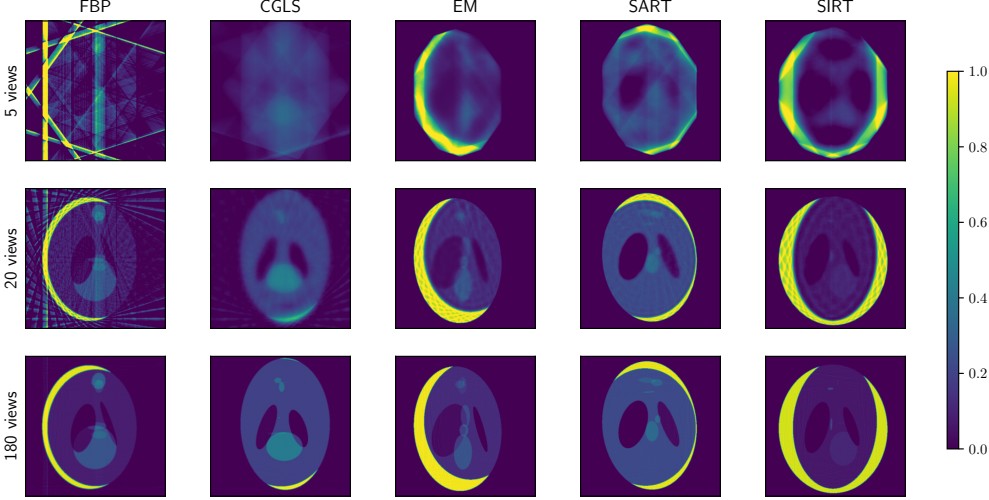

Figure 8: Reconstructions, generated with varying view angles, by the 5 classical reconstruction algorithms on the Shepp-Logan validation set.

## C   Metrics and Uncertainty

### C.1   PSNR & SNR

PSNR is defined as

$$\mathrm{PSNR}\big(f^*, f\big) = 10 \log_{10}\left(\frac{\max(f^*)^2}{\mathrm{MSE}(f^*, f)}\right), \tag{44}$$

$$\text{where}\quad \mathrm{MSE}\big(f^*, f\big) = \frac{1}{|\mathcal{X} \times \mathcal{Y}|} \sum_{x,y} \big(f^*(x,y) - f(x,y)\big)^2$$

while SNR is defined as

$$\mathrm{SNR}\big(f^*, f\big) = 20 \log_{10}\left(\frac{||f^*||_2}{||f^* - f||_2}\right), \tag{45}$$

where $f^*$ denotes the ground truth image, $f$ the noisy/reconstructed image, and $||\cdot||_2$ the $\ell^2$-norm. SNR is strictly less than PSNR, with higher SNR and PSNR corresponding to better image reconstruction. In the absence of any noise, $f^*$ and $f$ are identical, making SNR and PSNR infinite. For lossy images, PSNR is typically between 30-50dB, with values over 40dB considered very good, and values below 20dB considered unacceptable (Bull, 2014).

### C.2   Types of Uncertainty

Bayesian modeling can address two distinct types of uncertainty: aleatoric and epistemic (Der Kiureghian & Ditlevsen, 2009; Kendall & Gal, 2017). Aleatoric uncertainty is due to *measurement noise*, such as X-ray detector noise. This type of uncertainty cannot be reduced, even if more measurements are taken, since it is inherent to the measurement. To see why, think of rolling an unbiased die. Irrespective of how many times you roll the die, you will always be uncertain of the outcome of the next roll, since each outcome has a $\frac{1}{6}^{\text{th}}$ probability. On the other hand, epistemic or model uncertainty accounts for *uncertainty in the model parameters*. This type of uncertainty can be reduced with more measurement data. To see why, imagine a model that aims to predict the outcome of a biased die roll, with no prior information about the bias. As more data is taken, the variance in the model parameters decreases, and the model output distribution better approximates the true biased die distribution. Even in the presence of aleatoric uncertainty, this work primarily focuses on quantifying epistemic/model uncertainty. Namely, we consider how well the model reconstructs the ground truth attenuation coefficient image from the sinogram data.

### C.3   Calibration and Coverage

Calibration is a metric that assesses a model's ability to predict the probabilities of its outcomes, gauging the reliability of the model's confidence in its predictions. For example, a model performing class predictions is considered calibrated if it assigns a class 50% probability and that class actually appears 50% of the time in prediction. For further information on calibration of class prediction models, we refer the reader to (Guo et al., 2017; Nixon et al., 2019). Since this work focuses on regression models, the remaining discussion is centered on calibrated regression (Kuleshov et al., 2018).

Let $F$ be the cumulative distribution function (CDF) of model predictions $f(x, y)$, that seek to approximate ground truth image $f^* \in \mathcal{F}$, where $\mathcal{F}$ denotes the functional space of possible images. Letting $f_x := f(x, y)$ We denote the corresponding quantile function as

$$F_x^{-1}(\tilde{p}) = \inf\{f_x : \tilde{p} \le F(f_x)\}, \tag{46}$$

where $F^{-1}$ performs the mapping $F^{-1} : [0, 1] \to \mathcal{F}$ and $\tilde{p}$ is a confidence interval. For calibrated regression, ground truth pixel $f(x, y)$ should fall in a, say, 90% confidence interval 90% of the time. Thus, the regression model is calibrated for confidence interval $\tilde{p}$ if

$$\lim_{X \to \infty} \frac{1}{X} \sum_{x=1}^{T} \mathbb{I}\big\{f_x \le F_x^{-1}(\tilde{p})\big\} = \tilde{p}, \;\; \forall \tilde{p} \in [0, 1], \tag{47}$$

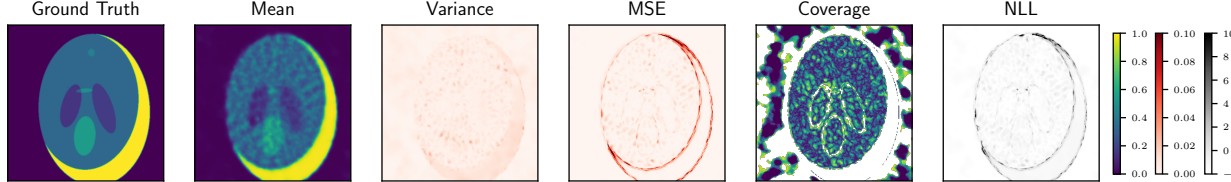

Figure 9: The ground truth image is plotted alongside different metrics for assessing the reconstructed predicted distribution generated by an UINR model. From left to right, we show the ground truth, predicted mean image, predicted variance image, mean squared error, coverage quantile of each pixel, and negative log-likelihood of each pixel. Note that PSNR is calculated using the ground truth and predicted mean image. In a real medical setting, the ground truth is unknown, the doctor would be given the predicted mean image and the predicted variance image could be provided as supplementary information to help the doctor reach a diagnosis. Further note that white regions in the coverage image denote that the ground truth pixel value did not fall in the range of the predicted distribution.

as the number of pixel samples approaches infinity, $X \to \infty$. If $f_X^*$ denotes the ground truth value for i.i.d. random pixel $(x, y) \in X$, a sufficient condition for calibrated regression is

$$p\left(f_X^* \leq F_X^{-1}(\tilde{p})\right) = \tilde{p} \, , \, \forall \, \tilde{p} \in [0, 1]. \tag{48}$$

Since practical dataset sizes are finite, preventing perfect calibration, different metrics have been developed to assess empirical model calibration.

Reliability diagrams serve as a visual representation of model calibration, plotting expected sample accuracy as a function of average model confidence. Ideally these would be continuous plots, but, in practice, samples are binned into $M$ bins according to their prediction confidence. Let $B_m$ be the set of indices, $i$, of samples with prediction confidence, $\hat{p}_i$ in the interval $I_m = (\frac{m-1}{M}, \frac{m}{M}]$. The expected accuracy (acc) and confidence (conf) are approximations to the terms of Eq. 48, namely

$$p\left(f_X^* \leq F_X^{-1}(\tilde{p})\right) \approx \text{acc}(B_m) = \frac{1}{|B_m|} \sum_{i \in B_m} \mathbb{I}\left\{f_i^* \leq F_i^{-1}(\tilde{p})\right\} \tag{49}$$

$$\tilde{p} \approx \text{conf}(B_m) = \frac{1}{|B_m|} \sum_{i \in B_m} \hat{p}_i. \tag{50}$$

The calibration error (CE) is the discrepancy

$$\text{CE}(\tilde{p}) = \mid p\left(f_X^* \leq F_X^{-1}(\tilde{p})\right) - \tilde{p} \mid \approx \mid \text{acc}(B_m) - \text{conf}(B_m) \mid = \text{CE}(B_m). \tag{51}$$

It can be measured on a reliability diagram as the difference between the expected accuracy curve and the ideal $\text{acc}(B_m) = \text{conf}(B_m)$ line. The expected calibration error (ECE) quantifies the calibration error of the full distribution as

$$\text{ECE}(f^*, F_X^{-1}, \tilde{p}) = \frac{1}{M} \sum_{m=1}^{M} \mid \text{acc}(B_m) - \text{conf}(B_m) \mid. \tag{52}$$

The model is considered calibrated if $\text{ECE}(x, f) = 0$.

In practice, modifications were made to the previously described theory of reliability curves and ECE. You may notice in Figure 10 that, instead of plotting *accuracy* and *confidence*, we instead plot analogous *target coverage* and *achieved coverage*. Typically, a coverage value, $\bar{p}$, refers to a quantile of data points lying within $\pm \frac{\bar{q}}{2}\%$ of the median (50% quantile). Specifically, in our setup, we use different uncertainty quantification methods (BBB, MCD, and DE) to sample $N$ different model weights for our INR, each set of weights corresponding to a different model output. Given that each output corresponds to an image, for each pixel, $(x, y)$, we have a distribution of $N$ predicted values, $F_N(x, y)$. Ideally, the median of the pixel distribution would be equivalent to the ground truth pixel value, $f^*(x, y)$. However, this is unrealistic to expect in

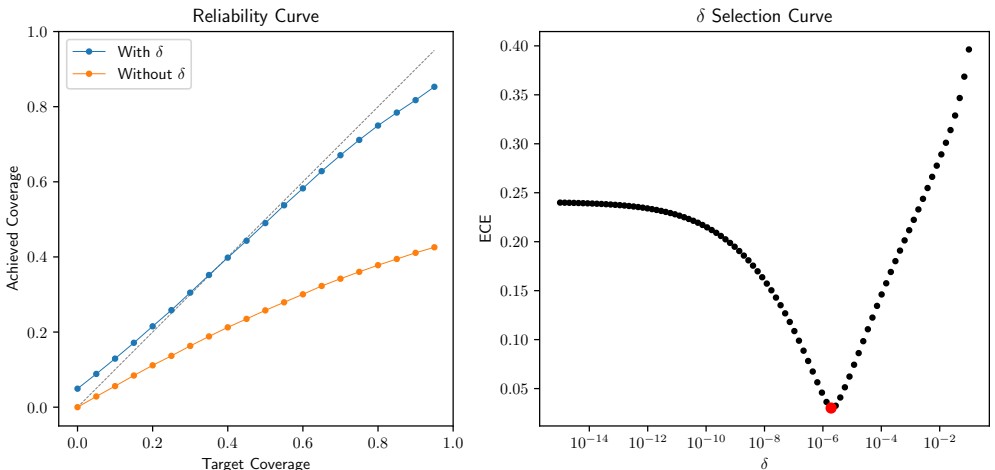

Figure 10: Both plots were made with an MCD UINR model trained on 20 views, achieving a PSNR of 19. **Left)** A plot of the model reliability curves, with the grey dashed line indicating a perfectly calibrated model. The blue curve is the empirical reliability curve of the model when a small $\delta$ term is added symmetrically to the target coverage, in order to slightly widen the quantile ranges. Although this $\delta$ term has nearly negligible magnitude, it significantly improves the model reliability curve, as illustrated by the orange curve of reliability without the added $\delta$ term. **Right)** The added $\delta$ term was not chosen arbitrarily, but selected to minimize ECE.

practice. Thus, we check whether the ground truth pixel lies within the predicted pixel distribution quantile, $Q_n$, specified by coverage value, $\bar{p}$,

$$Q_{50-\frac{\bar{p}}{2}}\big(F_N(x,y)\big) \leq f^*(x,y) \leq Q_{50+\frac{\bar{p}}{2}}\big(F_N(x,y)\big). \tag{53}$$

If a model is perfectly calibrated, $\bar{p}\%$ of reconstructed pixels distributions will contain the ground truth pixel in their $\bar{p}\%$-th quantile, corresponding to the grey dashed line in Figure 10. Thus, model confidence can be seen as a pre-selected quantile for each pixel (target coverage), $p$, while accuracy is the percentage of pixel distributions containing the ground truth that quantile (achieved coverage),

$$\text{AC}(f^*, F_N, \bar{q}) = \frac{1}{|\mathcal{X}|}\frac{1}{|\mathcal{Y}|}\sum_{x\in\mathcal{X}}\sum_{y\in\mathcal{Y}}\mathbb{I}\bigg\{Q_{50-\frac{\bar{q}}{2}}\big(F_N(x,y)\big) \leq f^*(x,y) \leq Q_{50+\frac{\bar{q}}{2}}\big(F_N(x,y)\big)\bigg\}. \tag{54}$$

The ECE is thus implemented as,

$$\text{ECE}(f^*, F_N) = \frac{1}{|\mathcal{P}|}\sum_{\bar{p}\in\mathcal{P}}|\text{AC}(f^*, F_N, \bar{p}) - \bar{p}|, \tag{55}$$

where $\mathcal{P}$ is a finite set of percentages evenly spaced in $[0, 1]$, separated by percentage interval $i << 1$. The fifth image of Figure 9, plots the smallest quantile of each pixel containing the corresponding ground truth pixel, for an example UINR reconstruction with $N = 50$. Note that white regions indicate that the ground truth value does not fall within the minimum and maximum predicted pixel values.

There is one final modification made in implementing the reliability curves, in order to effectively assess model calibration. Since the final layer of all the NNs used for the INR have a sigmoid activation, ensuring that the model output is in the range $(0, 1)$. However, the sigmoid function only approaches 0 and 1 in the infinite limit, meaning that in practice our model will never output 0 or 1 exactly. However, our images contain a large percentage of pixels with exactly 0 value, especially for noiseless artificial data, which in the context of medical imaging is regions containing air and no tissue. This is problematic for calibration, since all of predicted pixel values will be near-zero, but will not actually contain the ground truth value of 0. This is illustrated by the orange reliability curve in Figure 10, for which only 40% of pixels contain the ground

truth in their full range of predicted pixel values, for an MCD model trained on 20 views with $N = 50$. Our proposed solution to this issue is slightly widening the quantile range by adding a negligible $\delta$ term. Thus, for coverage value $\bar{p}$, we now check if the ground truth pixel lies in,

$$Q_{50-\frac{\bar{q}}{2}}\big(F_N(x,y)\big) - \delta \le f^*(x,y) \le Q_{50+\frac{\bar{q}}{2}}\big(F_N(x,y)\big) + \delta, \tag{56}$$

where $0 < \delta << 1$. In this case, if our predicted pixel values are slightly larger than 0, the $\delta$ offset can widen the quantile range to include 0, enabling these pixels to contribute to the achieved calibration (this also applies to pixels with exact value of 1). The improvement in using a delta offset is illustrated by the blue reliability curve in Figure 10, which is much closer to the ideal grey dashed line than the orange curve with $\delta$. It should be noted that the value of $\delta$ is not assigned arbitrarily, but instead optimized to minimize overall ECE. For too small a $\delta$, the quantiles will not be widened sufficiently to capture ground truth 0 pixels. However, for too large of $\delta$, the quantiles will be widened too much, reducing overall calibration, as achieved coverage is much higher than target coverage for low coverage values. Thus, ECE as a function of $\delta$ is expected to have a unique minima, as illustrated by the example in Figure 10.

### C.4 Assessing Model Quality

Section 4.1 describes how PSNR and SNR quantify image reconstruction quality, coverage metrics (such as ECE) gauge the uncertainty calibration, and NLL encapsulates both. In this work, we aim to optimize both reconstruction and calibration quality, meaning the best metric would, naively, be NLL. However, there is often a trade-off between calibration and prediction quality. Specfically, NN overfitting to NLL manifests in probabilistic error rather than prediction error (Guo et al., 2017). Furthermore, while this work focuses on uncertainty quantification of INRs for medical imaging, little prior work has addressed this problem. Most existing techniques only quantify reconstruction PSNR and SNR. Hence, for fair comparison, we assess and optimize our models primarily according to PSNR and SNR. However, for similarly performing models, we use NLL and ECE as secondary selectors for the best model. Note that initial attempts at optimizing models according to coverage metrics resulted in preference for the lowest capacity models possible. This indicates that optimal performance according to coverage favors blurry image reconstruction, with as little certainty as possible in the final image.

# D   Bayesian Deep-Learning Methods and Implementations

Building off of the high-level descriptions of the different BDL methods discussed in Section 3, we provide more detailed algorithms descriptions and implementation specifics.

## D.1   Bayes-by-Backprop

A popular method for variational approximation to exact Bayesian updates is Bayes-by-Backprop (BBB) (Blundell et al., 2015). BBB aims to find the optimal parameters $\psi$ of an approximate distribution on the NN weights, $q(\theta|\psi)$, also known as the variational posterior. This is achieved by maximizing the variational free energy/evidence lower bound (ELBO),

$$\mathcal{L}(S, \psi) = \mathbb{E}_{q(\theta|\psi)}[\log p(S|\theta)] - \mathbb{KL}\left[\, q(\theta|\psi) \,||\, p(\theta)\,\right], \tag{57}$$

with respect to the variational parameters $\psi$. The ELBO can be optimized with respect to $\psi$ using stochastic gradients estimated by Monte Carlo,

$$\nabla_\psi \mathcal{L}(S, \psi) \approx \nabla_\psi \left(\, \log p(S|\theta) + \log p(\theta) - \log q(\theta|\psi)\right), \tag{58}$$

where $\theta \sim q(\cdot|\psi)$ is a sample drawn from the variational posterior, and the gradient is taken through $\theta$ using the reparameterization trick (Rezende et al., 2014).

In this work, the BBB variational posterior is treated as a Gaussian distribution, $\mathcal{N}(\mu_\psi, \sigma_\psi)$. The elements of $\sigma_\psi$ comprise a diagonal covariance matrix, meaning weights are assumed to be uncorrelated. A Gaussian prior, $p(\theta) = \mathcal{N}(\theta|\sigma^2)$, with tunable $\sigma$ is used to initialize the network. Training the network requires computing a forward-pass and backward-pass. Although the network is parameterized by a distribution of weights, in each forward pass a single sample is drawn from the variational posterior and propagated through the network to perform updates. A re-parameterization trick (Kingma & Welling, 2014), in which the sample $\epsilon$ is transformed by the function $\mu_\psi + \sigma_\psi \odot \epsilon$, is used to ensure a gradient can be calculated for backpropagation. Finally, to aid learning, it is common to modify the ELBO as

$$\tilde{\mathcal{L}}(y, \psi) = \mathbb{E}_{q(\theta|\psi)}[\log p(S|\theta)] - \xi \cdot \mathbb{KL}\left[\, q(\theta|\psi) \,||\, p(\theta)\,\right], \tag{59}$$

where $\xi > 0$ is an added hyperparameter, known as the Kullback-Leibler (KL) factor. This is beneficial for training because it puts greater emphasis on the training data in the loss, through the $\mathbb{E}_{q(\theta|\psi)}[\log p(S|\theta)]$ In the context of medical imaging with INRs, this re-weights the importance of obtaining a Radon transform of network outputs close to the sinogram measurement data.

## D.2   Monte Carlo Dropout

Another popular approach is Monte Carlo dropout (MCD) (Gal & Ghahramani, 2016). There, the authors argue that optimizing a NN regularized with dropout (Srivastava et al., 2014) applied to every layer can be interpreted as variational approximation for a deep GP. The first two moments of the corresponding variational posterior can be approximated using Monte Carlo, with $N$ samples from NNs sampled with dropout. The predictive mean is thus calculated as,

$$\mathbb{E}_{q(\hat{f}_\theta|y)}(\hat{f}_\theta) \approx \frac{1}{N} \sum_{n=1}^{N} \hat{f}(\theta_{(n)}), \tag{60}$$

which is equivalent to averaging the results of $N$ stochastic forward passes through the network. Details of the MCD implementation are provided in the "Implementation Details" of main paper Section 3.

## D.3   Hamiltonian Monte Carlo

A final means of BNN inference which we explore in this work, is Hamiltonian Monte Carlo (HMC) (Duane et al., 1987; Neal et al., 2011; Betancourt, 2017). Originally proposed for calculations in lattice quantum

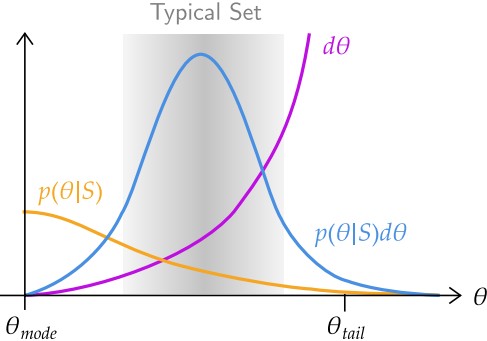

Figure 11: An illustration of concentration of measure, in which the expectation, $p(\theta|S)d\theta$, of the BNN's high-dimensional weight posterior is concentrated in the typical set. This can be attributed to the fact that, while most of the distribution density $p(\theta|S)$ is concentrated about the distribution modes, the volume $d\theta$ is concentrated at the tails. Unlike other MCMC methods, HMC efficiently explores and samples from the entire distribution typical set. (Figure inspired by (Betancourt, 2017).)

chromodynamics (Duane et al., 1987), HMC is an instance of the Metropolis-Hastings algorithm (Chib & Greenberg, 1995) for Markov Chain Monte Carlo (MCMC) (Gilks et al., 1995). In the context of quantifying NN uncertainty, HMC can be used to obtain a sequence of random samples that converge in distribution to samples from the BNN posterior distribution, $p(\theta|S)$ (Izmailov et al., 2021). For the high-dimensional distributions associated with BNNs, the density $p(\theta|S)$ is concentrated at the distribution mode, while the volume $d\theta$ is concentrated at the distribution tails. The resulting distribution expectation – a product $p(\theta|S)d\theta$ of distribution volume and density – concentrates in a nearly-singular neighborhood, known as the typical set, as illustrated in Figure 11. Traditional Metropolis-Hastings MCMC, using a Gaussian random walk proposal distribution, typically fails to explore the full typical set of these distributions. HMC, however, leverages the physics of Hamiltonian dynamics via a time-reversible and volume-preserving integrator, to simulate and propose points scattered around the typical set. This significantly improves exploration of the full distribution and decreases the correlation between consecutive samples, reducing the total number of required MCMC samples. In this work, HMC was implemented via the NumPyro Python package (Phan et al., 2019), using a leapfrog integrator and No-U-Turn sampler (Hoffman et al., 2014). For a more detailed description of the HMC algorithm, in the UncertaINR context, we refer the reader to Appendix E.

### D.4 Deep Ensembles

Alternatively, predictive uncertainty can be quantified by aggregating the outputs of several NN "base learners" trained for the same task from different initializations, a method known as deep ensembles (DEs) (Lakshminarayanan et al., 2017). Averaging predictions over multiple NNs consistent with the training data leads to better predictive performance, enables uncertainty quantification, makes DEs robust to model misspecification, and presents a strong baseline in out-of-distribution detection (Ovadia et al., 2019). Ensembling can also be applied on top of other methods to further improve performance. In principle, more base learners results in better performance, but, in practice, training large ensembles is computationally expensive and diminishing returns are observed after $\sim$10 base learners. In this work, UncertaINR leverages ensembles of $M$ Monte Carlo dropout base learners, such that Eqs. 9 and 10 are updated to,

$$\hat{f}_{\text{DE}}(x,y) = \frac{1}{N}\sum_{m=1}^{M}\sum_{n=1}^{N/M} f_{\theta_{m,n}}(x,y) \tag{61}$$

$$V_{\text{DE}}(x,y) = \frac{1}{N-1}\sum_{m=1}^{M}\sum_{n=1}^{N/M} \left(f_{\theta_{m,n}}(x,y) - \hat{f}_{\text{DE}}(x,y)\right)^2.$$

The relationship between Bayesian inference and DEs is an active area of research in the BDL community. Wilson & Izmailov (2020) argue that DEs provide a more compelling approximation to the true posterior than many standard BDL approaches, whilst others have adapted DEs to provide a Bayesian interpretation (Ciosek et al., 2019; D'Angelo & Fortuin, 2021; Pearce et al., 2020). He et al. (2020) characterize how DEs relate to posterior inference in the limit of infinite NN width.

Typically, DEs induce randomness by training the same network several times with randomized initializations and data order. However, recent work (Zaidi et al., 2021) has shown that ensembling over architectures can outperform the more common single-architecture DEs for uncertainty estimation. For our large-scale hyperparameter study (on Shepp-Logan phantom data), in which thousands of BBB and MCD UncertaINR base learners were trained, we were able to easily implement architecture-ensembled DEs, by selecting the best-performing UncertaINRs as base learners. However, for the high-resolution AAPM-Mayo data, in which UncertaINR training times were extended significantly, we found it too computationally demanding to hypertune multiple architectures to ensemble. Thus, for the final results presented, DEs are created by ensembling the same network trained with randomized weight initialization.

### D.5 Hardware and Software Notes

The experiments presented in this paper were computationally intensive, requiring hundreds of compute hours on parallelized GPUs. Runtimes for each of the methods developed and implemented in this work on the AAPM dataset are presented in Table 4. Non-HMC methods were run on a cluster of 4 GPU nodes consisting of 8 GPUs each, containing a mixture of GTX 1080, GTX 1080Ti, and GeForce RTX 2080 Ti cards. HMC methods (*) were run on two Titan RTX cards, which were faster and had double the memory. Note that the HMC models were further intialized to a pre-trained check-point, which helped the models converge much faster. The time required to obtain this pre-trained checkpoint is not reflected in the HMC runtimes. Furthermore, while overall runtimes of the ensemble methods ($^\dagger$) appear long, each of the base models could be trained in parallel.

Table 4: **Computation Times**: Most experiments were run on a GPU cluster consisting of four GPU nodes with 8 GPUs each, consisting of a mixture of GTX 1080, GTX 1080Ti, and GeForce RTX 2080 Ti cards. Experiments denoted with a (*) were run on faster Titan RTX cards, with increased memory. ($^\dagger$) is used to denote ensemble experiments, which in practice were run in parallel to achieve faster runtimes.

| RECONSTRUCTION METHOD | 60-VIEW RUNTIME | 120-VIEW RUNTIME | COMPUTE TYPE |
|---|---|---|---|
| GOP-TV | < 10 MIN | ∼ 10 MIN | GTX/GEFORCE |
| HMC GOP-TV | 25+ MIN* | 45+ MIN* | TITAN RTX* |
| INR | ∼2 HRS | ∼2.75 HRS | GTX/GEFORCE |
| DE-2 UINR | ∼4 HRS$^\dagger$ | ∼5.5 HRS$^\dagger$ | GTX/GEFORCE |
| DE-5 UINR | ∼10 HRS$^\dagger$ | ∼11 HRS$^\dagger$ | GTX/GEFORCE |
| DE-10 UINR | ∼20 HRS$^\dagger$ | ∼27.5 HRS$^\dagger$ | GTX/GEFORCE |
| MCD UINR | ∼2.25 HRS | ∼2.8 HRS | GTX/GEFORCE |
| DE-2 MCD UINR | ∼5 HRS$^\dagger$ | ∼5.6 HRS$^\dagger$ | GTX/GEFORCE |
| DE-5 MCD UINR | ∼11.25 HRS$^\dagger$ | ∼14 HRS$^\dagger$ | GTX/GEFORCE |
| DE-10 MCD UINR | ∼22.5 HRS$^\dagger$ | ∼28 HRS$^\dagger$ | GTX/GEFORCE |
| HMC UINR | 8+ HRS* | 11.5+ HRS* | TITAN RTX* |

Overall, we do not forsee computation as a limitation of UncertaINR. In fact, this work demonstrates that MCD (one of the least computationally intensive means of UQ) is highly effective for INR UQ, while adding minimal overhead to the INR evaluation time. Furthermore, as mentioned in Section 2.2 recent work on INR optimization has found ways to reduce INR evaluation times. These approaches could be implemented directly within the UncertaINR framework to reduce training time, without affecting the network results or uncertainty quantification. This, however, would neither add any novelty or change the fundamental claims of this paper, which are about UQ and not computational speed.

The project codebase was developed in Python, mostly using Pytorch (Paszke et al., 2019), Hydra (Yadan, 2019), and Weights & Biases (Biewald, 2020) to implement the NN functionality and Blitz (Esposito, 2020) for BNN functionality. For HMC, we used the No-U-Turn-Sampler (Hoffman et al., 2014) sampling scheme in NumPyro (Phan et al., 2019), which is based in JAX (Bradbury et al., 2018).

# E HMC Sampling Algorithm for UncertaINR

Our goal in using HMC with UncertaINR is to sample weight parameters, $\{\theta^{(1)}, ..., \theta^{(N)}\}$, from the BNN weight posterior, $p(\theta|S)$. This problem is reformulated in terms of physics-inspired dynamics. These dynamics are governed by Hamiltonian

$$\mathcal{H}(\theta, P_\theta) = U(\theta) + \frac{1}{2}P_\theta^T M^{-1} P_\theta, \tag{62}$$

where $\theta$ are the 'position' terms, $P_\theta$ are the 'momentum' terms, $U(\theta) = -\ln p(\theta|S)$ is the system potential energy, $\frac{1}{2}P_\theta^T M^{-1} P_\theta$ is the system kinetic energy, and $M$ is a symmetric positive definite mass matrix. HMC starts by initializing parameters, $\theta^{(0)} \sim \mathcal{N}(0, \frac{1}{\tau})$, by sampling from the Gaussian distribution of prior precision $\tau$. At HMC iteration $n$, the parameters are initialized to $\theta^{(n)}(0) = \theta^{(n)}$ while the momentum is initialized by sampling from the normal distribution $P_\theta^{(n)}(0) \sim \mathcal{N}(0, M)$ of variance defined by the mass matrix. The leapfrog iterative algorithm is then used to simulate system dynamics for time $L\Delta t$, where $L$ is the number of leapfrog steps and $\Delta t$ the step size. In each step, the leapfrog algorithm alternates between momentum and position updates, using

$$P_\theta^{(n)}\left(t + \frac{\Delta t}{2}\right) = P_\theta^{(n)}(t) - \frac{\Delta t}{2}\nabla U(\theta)|_{\theta=\theta^{(n)}(t)} \tag{63}$$

$$\theta^{(n)}(t + \Delta t) = \theta^{(n)}(t) + \Delta t M^{-1} P_\theta^{(n)}\left(t + \frac{\Delta t}{2}\right) \tag{64}$$

$$P_\theta^{(n)}(t + \Delta t) = P_\theta^{(n)}\left(t + \frac{\Delta t}{2}\right) - \frac{\Delta t}{2}\nabla U(\theta)|_{\theta=\theta^{(n)}(t+\Delta t)}, \tag{65}$$

so as to solve Hamilton's equations

$$\frac{d\theta}{dt} = \frac{\partial \mathcal{H}}{\partial P_\theta} \tag{66}$$

$$\frac{dP_\theta}{dt} = \frac{\partial \mathcal{H}}{\partial \theta}. \tag{67}$$

Since the leapfrog iterative algorithm is a discretized numerical approximation to the true integral, the limit $\Delta t \to 0$ would be needed to solve Hamilton's equations exactly. To correct for bad proposals from the leapfrog method, an HMC Metropolis-Hastings acceptance ratio is defined as

$$\alpha_{\text{HMC}}(\theta^{(n)}(0), \theta^{(n)}(L\Delta t)) = \min\left\{1, \frac{\exp[-\mathcal{H}(\theta^{(n)}(L\Delta t), P_\theta^{(n)}(L\Delta t))]}{\exp[-\mathcal{H}(\theta^{(n)}(0), P_\theta^{(n)}(0))]}\right\}, \tag{68}$$

where $\mathcal{H}$ is the Hamiltonian defined in Eq. 62. In result, the parameter sample returned by iteration $n$, which is also the parameter initialization of iteration $n + 1$, is

$$\theta^{(n+1)}(0) \,|\, \theta^{(n)}(0), \theta^{(n)}(L\Delta t) = \begin{cases} \theta^{(n)}(L\Delta t), & \text{with probability } \alpha_{\text{HMC}}(\theta^{(n)}(0), \theta^{(n)}(L\Delta t)) \\ \theta^{(n)}(0), & \text{otherwise} \end{cases}. \tag{69}$$

This process is repeated for $T'$ HMC iterations. Since the prior weight initialization does not usually lie within the typical set of the distribution, the samples initially output by HMC are poor indicators of the typical set region. To address this problem, a burn-in period of $B$ initial iterations is defined and all burn-in samples are discarded after the algorithm is complete.

## F  Preliminary Ablation Study

Our preliminary ablation study presents a large-scale study of uncertainty quantification for INRs. Specifically, we test the relative ability of UncertaINR (using MCD, BBB, and/or DEs) to reconstruct artificial noiseless CT brain images. From this study, we present guiding principles for effective UncertaINR hyperparameter selection and compare to traditional CT reconstruction techniques.

### F.1  Dataset

Given the long INR training times required to reconstruct large, high-frequency images (Fridovich-Keil et al., 2022), we opted for a dataset of simple images, enabling large-scale hyperparameter sweeps. Specifically, the Shepp-Logan phantom (Shepp & Logan, 1974) approach was used to generate ($256 \times 256$ pixel) artificial brain images, with corresponding measurement sinograms generated via the Radon transform. No noise was added to the measurement sinograms. In all, 10 ground truth images, depicted in Figure 12, and 20 corresponding sinograms were generated: 5 validation and 5 test set sinograms each for the 5- and 20-view ($\phi$) cases.

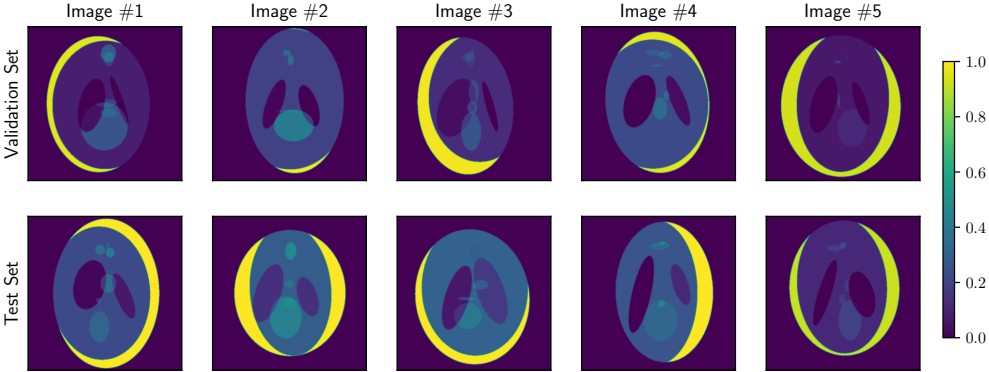

Figure 12: The ground truth images used in tuning and assessing our preliminary Shepp-Logan hyperparamter exploration. The five validation set images were used to optimized model hyperparameters, while test set images were used to assess the finalized models.

### F.2  Baselines

In this study, we compared UncertaINR to the classical medical image reconstruction algorithms – FBP, CGLS, EM, SART, and SIRT – described in Section 2.1.1 and Appendix B, implemented using the TomoPy Astra wrapper (Pelt et al., 2016). A detailed analysis of the classical reconstruction methods on the Shepp-Logan validation set is presented in Appendix B.7.

### F.3  Tuneable INR Model Parameters

There are several degrees of freedom in designing an INR, including its size, embeddings, activation functions, and optimizer. These design choices are critical in determining model performance, but there is little theoretical understanding of how to best select most of these model parameters. In this section, we lay out the different parameters we considered in designing our INRs and provide any known insights as to how they affect model performance. These insights informed the hyperparameter sweeps described in the following section.

#### F.3.1  Width and Depth

The size of an NN is determined by both its width (number of nodes per layer) and depth (number of layers). Universal approximation theorems (Hornik et al., 1989) have been derived in both the arbitrary-width (Cybenko, 1989; Hornik, 1991) and arbitrary-depth (Hanin, 2019; Kidger & Lyons, 2020; Lu et al.,

2017) cases, demonstrating that NNs are theoretically guaranteed universal function approximators in the infinite limit. In practice, however, neural networks have finite width and depth. Recent work has empirically demonstrated and theoretically suggested that, in this regime, increased-depth networks generally perform better than increased-width networks (Lu et al., 2017; Raghu et al., 2017). It is also known that, while neural networks are overparametrized relative to the amount of training data, this overparametrization is key for their generalization ability (Jacot et al., 2021; Neyshabur et al., 2019). However, for INRs specifically, it has been shown that relatively small networks can typically be used to learn decent functional image encodings (Dupont et al., 2021). Thus, we tend to sweep over smaller widths and depths than standard deep-learning networks.

### F.3.2 Fourier Feature Mappings

Random Fourier features (RFF) were first introduced in 2007 by (Rahimi & Recht, 2007) as a means of accelerating kernel methods. The key idea is to map the input data to a randomized low-dimensional feature space, while maintaining the kernel of the original data. Given input $\vec{x} \in \mathbb{R}^n$, the RFF mapping takes the form

$$\gamma_{\text{RFF}}(\vec{x}) = [\cos(2\pi B\vec{x}), \sin(2\pi B\vec{x})]^T, \tag{70}$$

where $B$ is an $m \times n$ matrix, with each entry sampled from $\mathcal{N}(0, \Omega_0^2)$. The standard deviation, $\Omega_0$, is a tuneable hyperparameter, but remains static after initialization – i.e. it is not modified with NN weights during the MLP training. There exist other types of Fourier feature mappings, such as positional encodings, in which

$$\gamma_{\text{PE}}(\vec{x}) = [..., \cos(2\pi\Omega_0^{j/m}\vec{x}), \sin(2\pi\Omega_0^{j/m}\vec{x}), ...]^T, \tag{71}$$

for $j = 0, ..., m-1$.

In 2018, it was theoretically demonstrated that NNs can be approximated by kernel regression via the neural tangent kernel (NTK) (Jacot et al., 2021). Using this intuition, in 2020, it was argued that applying a simple Fourier feature mapping to input data enables MLPs to learn high-dimensional functions rapidly, even in low-dimensional problem domains (Tancik et al., 2020), making the technique particularly well-suited for INRs. In fact, positional encodings have been shown to have key importance in the success of NeRF (Mildenhall et al., 2020) and Fourier feature mappings have been shown to boost the performance of the CoIL network for medical image reconstruction (Sun et al., 2021). In this work, all networks apply an RFF mapping, $\gamma_{\text{RFF}}$, to the input data. The standard deviation, $\Omega_0$, is tuned among other hyperparameters.

### F.3.3 Activation Functions

Activation functions are key to the success of neural networks, transforming what would otherwise be simple linear systems into complex, non-linear universal function representers. We performed hyperparameter sweeps with five activations widely used in the MLP and INR literature – ReLU, SiLU, Sine, SoftPlus, and Tanh. We now briefly review these activation functions, as well as their use in deep learning.

The rectified linear unit (ReLU), plotted in blue in Figure 13, was introduced as early as the 1960s for visual feature extraction in hierarchical NNs (Hara et al., 2015). The ReLU is defined as

$$\text{ReLU}(x) = \max\{0, x\}, \tag{72}$$

returning its input if greater than zero and otherwise returning zero. Despite its hard non-linearity at zero, non-differentiability at zero, and vanishing gradient challenge (Lu et al., 2020), the ReLU was shown in 2011 to enable better training than previously used activation functions, such as Sigmoid and Tanh, by inducing sparse representations (Glorot et al., 2011). As of 2017, the ReLU was the most popular activation function for deep NNs (Ramachandran et al., 2017).

The sigmoid-weighted linear unit (SiLU), plotted in orange in Figure 13, is a specific instance of the Swish activation function family and was proposed in 2017 as a continuous, 'undershooting' version of the ReLU (Ramachandran et al., 2017). The Swish family, parameterized by $\beta$, is defined as

$$\text{Swish}_\beta(x) = x \cdot \sigma(\beta x) = \frac{x}{1 + e^{-\beta x}}, \tag{73}$$

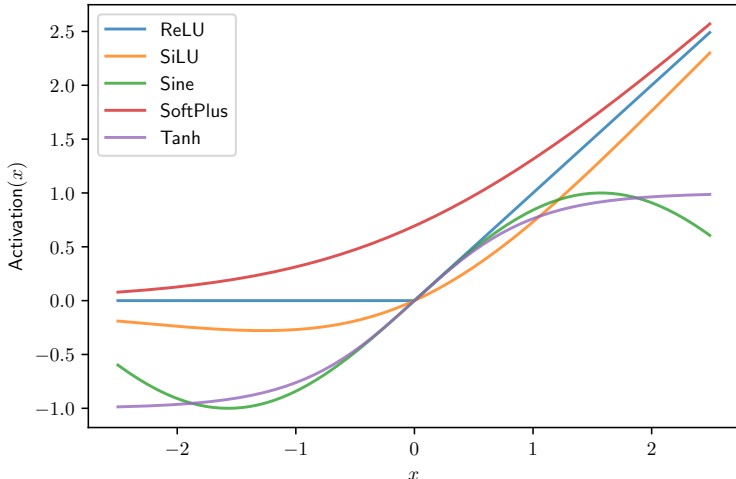

Figure 13: The five different activation functions – ReLU, SiLU, Sine, SoftPlus, and Tanh – tested in our preliminary hypertuning experiment.

where $\sigma(x)$ is the sigmoid function. By setting $\beta$ to different values in $[0, \infty)$, Swish$_\beta$ non-linearly interpolates smooth functions between the linear function and ReLU. In 2017, Swish was empirically shown to outperform ReLU, a result theoretically attributed to its bounded, smooth, and non-monotonic nature (Ramachandran et al., 2017). More recently, Swish has been shown to outperform both ReLU and Sine in the context of CT image reconstruction via Automatic Integration (AutoInt) (Lindell et al., 2021). The SiLU is the specific instance of Swish where $\beta = 1$,

$$\text{SiLU}(x) = x \cdot \sigma(x) = \frac{x}{1 + e^{-x}} \ . \tag{74}$$

The Sine activation function, plotted in green in Figure 13, is the sinusoid

$$\text{Sine}_{\omega_0} = \sin(\omega_0 \cdot x). \tag{75}$$

In the 2020 SIREN paper (Sitzmann et al., 2020), INRs with sinusoidal activation functions and random Fourier features were empirically demonstrated to outperform ReLU-based INRs. Theoretically, it was argued that these periodic activations are better suited to capturing naturally complex signals and their derivatives. However, the performance of these activations depends strongly on the choice of frequency, $\omega_0$, which needs to be tuned.

The SoftPlus activation function, plotted in red in Figure 13, has continuous and differentiable form

$$\text{Softplus}(x) = \ln(1 + e^x). \tag{76}$$

It was introduced in 2001 (Dugas et al., 2001) as the primitive of the sigmoid function. It is primarily used as a smooth approximation to the ReLU activation and to constrain to positive outputs, since Softplus$(x) \in (0, \infty)$.

The hyperbolic tangent Tanh, plotted in purple in Figure 13, has form

$$\text{Tanh}(x) = \frac{e^x - e^{-x}}{e^x + e^{-x}}, \tag{77}$$

and is both differentiable and monotonic. It has a form similar to the sigmoid function, with $\text{Tanh}(x) = 2\sigma(2x) - 1$, but lies in the range (-1,1) instead of (0,1), meaning it does not constrain to positive values. Before the ReLU became popular, the sigmoid and Tanh were two of the most common activation functions. Tanh was easier to train and typically outperformed the sigmoid as an activation function. However, because these sigmoidal activation functions saturate for large inputs, their derivatives vanish for these inputs, leading to slow convergence of learning algorithms. This has motivated the increased use of ReLU-like activation functions (Goodfellow et al., 2016), which ameliorate the vanishing derivative problem.

### F.4 Experimental Design

The goal of this hyperparameter study was to understand the relative performance of UncertaINR with BBB, MCD, and DEs. In all these cases, well-tuned hyperparameters were needed to achieve decent model reconstruction accuracy and uncertainty calibration. Large hyperparameter sweeps were used to find the optimal parameters for BBB and MCD. The best performing MCD UncertaINRs were used as base learners for the DEs.

#### F.4.1 BBB & MCD

For both BBB and MCD, MLP design choices had a large effect on INR reconstruction performance. Carefully designed hyperparameter sweeps were thus run to strategically search the MLP parameter space for four different settings: (1) MCD 5-view, (2) MCD 20-view, (3) BBB 5-view, and (4) BBB 20-view.

The model selection process began with a coarse grid search to efficiently prune across the wide range of possible parameter combinations. Specifically, we considered model activation type, depth, width, RFF embedding frequency $\Omega_0$, and dropout probability. Among these, activation type was the only categorical parameter, using the five activation types plotted in Figure 13: Tanh, SoftPlus, Sine, SiLU, and ReLU. For the remaining parameters, this initial coarse grid search was used to get a sense of orders of magnitude, for the sake of computational feasibility. We swept over model depths of 3, 6, and 9; widths of 16, 64, 256, and 1024; and RFF $\Omega_0$'s of 1, 5, 10, and 15. Three values - 0.2, 0.5, and 0.8 - were considered for the final parameter, dropout probability, which is specific to MCD. For BBB, we instead swept over the Gaussian prior standard deviation (values 10, 100, and 1000)[5] and KL factor (values 1e-10, 1e-5, and 1e-1)[6]. For these coarse grid searches, all networks were trained using the Adam optimizer with no weight decay and the default learning rate of 3e-4. For each set of parameters, three individual INRs were trained, one for each of the three validation images, Image #1-#3, shown in Figure 12[7]. All reported metrics are averaged across the three test images, in an effort to ensure model generalization and prevent overfitting to a particular image. For both the 5- and 20-view experiments, 2,160 (2,512) models were trained and tested for the MCD (BBB) coarse grid sweeps. This resulted in a total of 9,344 models trained and tested during these initial grid searches. Since the performance metrics are calculated from a distribution of predictions, sampled according to the uncertainty method, all hyperparameter sweeps used 50 prediction samples to enable uncertainty quantification, mean output prediction, and metric calculation.

For now, we conclude our discussion of methodology, with the second hyperparameter sweep – a fine Bayesian search used to generate the final models. This Bayesian sweep leveraged a reduced search space, informed by the first coarse grid search. It was found that Bayesian sweeps do not perform well with categorical variables, so independent sweeps of $\sim$200 runs were performed for each of the three best performing activation functions from the grid search. Since only 600 models were trained in total in each uncertainty-view setting, all five validation images of Figure 12 were used as the validation set, to improve model generalization ability. Further, AdamW was used to optimize the models, with a weight decay hyperparameter added to the sweep. In the case of MCD with 20-views, three sweeps were performed, one for each of the three activation functions: Sine, SiLU, and Tanh. A log uniform distribution, in the range 1e-16 to 1e-1, was swept for the weight decay, while uniform distributions $U(\min, \max, q)$, where $q$ is the discrete interval, were generated and swept over for the remaining numerical parameters: depth $\in U(2, 12, 1)$, width $\in U(200, 1000, 100)$, $\Omega_0 \in U(3, 15, 1)$, and $p(\text{dropout}) \in U(0.1, 0.6, 0.1)$. The top performing model according to PSNR, across all three Bayesian sweeps, was selected as the final model.

#### F.4.2 Deep Ensembles

Given the robust and computationally intensive nature of DEs, we did not perform large-scale hyperparameter sweeps, as was the case for BBB and MCD. DEs combine the outputs of multiple base learner models

---

[5]We originally swept over BBB standard deviation (values 0.2, 0.5, and 0.8), which are more on par with theoretical expectations. However, we found that increasing prior standard deviation significantly improved final model performance.

[6]Given the added BBB uncertainty hyperparameter, we reduced relative number of search values for the remaining sweep parameters.

[7]Again, for the sake of computationally efficiency, only validation Images #1 and #2 were used for BBB.

to improve uncertainty calibration. Assuming each base learner makes a reasonable prediction, even if not optimized, adding more base learners should only maintain or improve uncertainty calibration. In order to create a DE of size $M$ (DE-$M$), the top-$M$ performing models identified by the MCD hyperparameter sweeps were used as base learners. If $N$ total samples were desired for uncertainty quantification, each of the MCD baselearners was sampled repeatedly to generate $\frac{N}{M}$ predictions. These predictions were pooled together to create a sample of size $N$, from which uncertainty was quantified, the mean prediction generated, and model metrics calculated. In order to remain consistent with the BBB and MCD experiments, we used $N = 50$.

### F.5 Uncertainty Quantification Method Performance Analysis

### F.5.1 MCD Hyperparameter Analysis

For MCD hyperparameter sweeps, metrics were averaged across the INR model reconstruction of the first three validation set images of Figure 12. However, we also recorded the PSNR of the INRs trained on each individual image. Box plots for the individual distributions of PSNR for validation Images #1, #2, and #3 are shown in Figure 14. Ideally, PSNR should be consistent across the three images. This is roughly the case (barring a few outlier points) for models using the Tanh, SoftPlus, Sine, and Silu activation functions. Notice, however, that in all cases the distribution is broadest for Image #3. This effect is exacerbated for models using the ReLU activation function, for which PSNR of Image #3 ranges all the way from approximately 0dB to nearly the maximum achieved PSNR, in both the 5- and 20-view cases. This indicates that ReLU in particular, but all the all activation functions to some extent, struggle to capture features of Image #3.

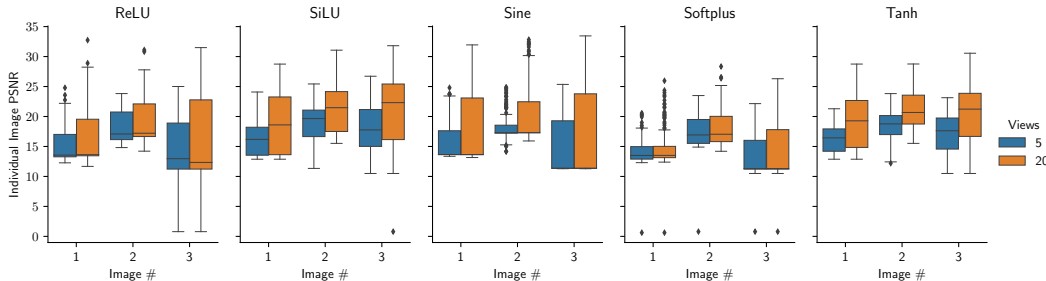

Figure 14: Reconstruction PSNR for Images #1-3 of Figure 14, for MCD INRs with different activations and different numbers of views.

To understand what is unique about Image #3, the image reconstructions of the best overall performing model are shown, for each activation function and 20-views, in Fig. 15. It is apparent that Sine and SiLU produce smoother images, while Tanh, Softplus, and ReLU produce spottier images. Except for SoftPlus, all activation functions manage to capture low-frequency image information and strong edges. However, all activations struggle to capture the small, low-intensity ellipses in the center of Image #3. Overall, it is clear that, irrespective of activation function, 20-views are insufficient to robustly capture fine CT image details. This individual image analysis also suggests that the Sine activation performs the best for MCD, achieving the highest overall PSNR.

Figure 16 shows boxplots of the average PSNR distributions for MCD models trained in the 5- and 20-view cases. Each column contains all the models trained with one of the five activation functions, while each row shows how the PSNR distribution changes as a function of hyperparameter – depth, width, probability of dropout, or RFF frequency $\Omega_0$. Ideally, PSNR would be consistently large across hyperparameter values, indicating that the architecture is robust and does not require much tuning. In practice, however, we find that the activation functions are either consistent or high-performing, but not both. As previously mentioned, Sine achieves the best overall PSNR. However, it is also the least consistent activation function, with its highest performing models typically being outliers (indicated by diamonds in Figure 16). Softplus, on the other extreme, is very consistent across hyperparameter values, but performs consistently poorly. Tanh, Silu, and ReLU have less extreme variations. Their top models perform slightly worse than the best Sine models, but they perform much more consistently across hyperparameter values (ReLU is a bit inconsistent in width and probability of dropout). This suggests that the Sine network is potentially the best reconstruction

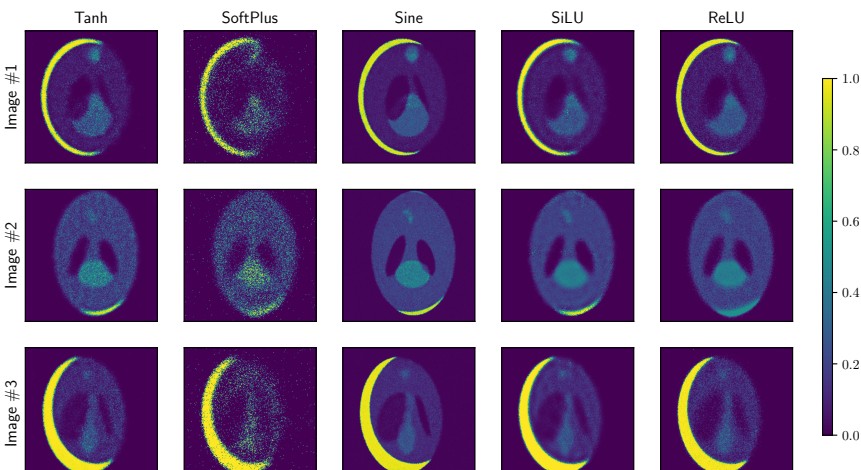

Figure 15: Best image reconstructions obtained with each activation function, for 20-view MCD.

network, but significant tuning (in terms of hyperparameter search) effort may be needed to achieve that solution. For practitioners inclined to perform less tuning, the Tanh and SiLU networks may be a preferred solution, due to their robustness and competitive top performance.

For ReLU, Tanh, SiLU, and Sine, it should also be noted that the PSNR distributions behave similarly in the 5- and 20-view cases (with 5 views performing consistently worse than 20 views, as expected) for all hyperparameters except RFF $\Omega_0$. This is consistent with recent results (Tancik et al., 2020) suggesting that RFF embeddings enable NNs to learn higher frequency image information. In the 5-view case, where the available data is insufficient for the INR to confidently learn high-frequency image features, performance is poor for increasing $\Omega_0$. In the 20-view case, the increased data enables the network to learn higher frequency image components. However, because a higher-frequency $\Omega_0$ is required to ensure the network can actually learn those frequencies, best performance tends to occur for larger $\Omega_0$. This effect can also be observed in Figure 17. For 5 views, the reconstruction is blurry for $\Omega_0 = 1$ but the network starts to learn smaller ellipses for $\Omega_0 = 5$. By $\Omega_0 = 10$, however, the network is trying to learn higher-frequencies than the data cannot specify, resulting in artifacts, which are exacerbated for $\Omega_0 = 30$. In the 20-view case, a similar trend of reduced blurriness and increasingly sharper images can be seen between $\Omega_0 = 1$ an $\Omega_0 = 10$. However, the reconstructed images have much more recognizable details and less artifacts than those obtained with 5 views. It is only for $\Omega_0 = 30$ that artifacts start to appear.

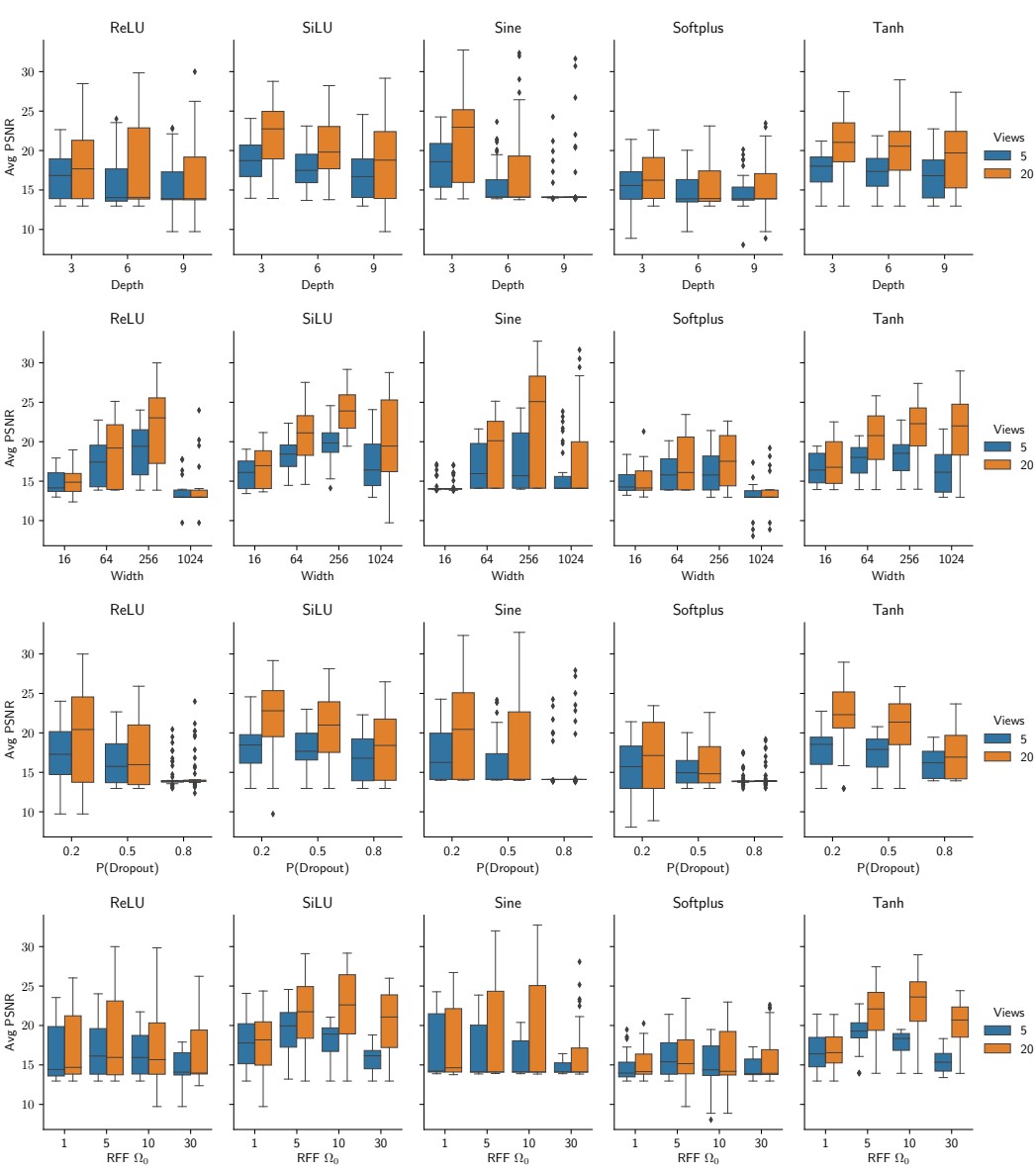

Figure 16: Boxplots of the average PSNR of MCD models trained in the coarse grid search hyperparameter sweep, for both 5 and 20 views. Each column corresponds to a different activation function and each row to a sweep over one of the remaining hyperparameters - depth, width, probability of dropout, and RFF frequency $\Omega_0$. Individual diamond points are outliers.

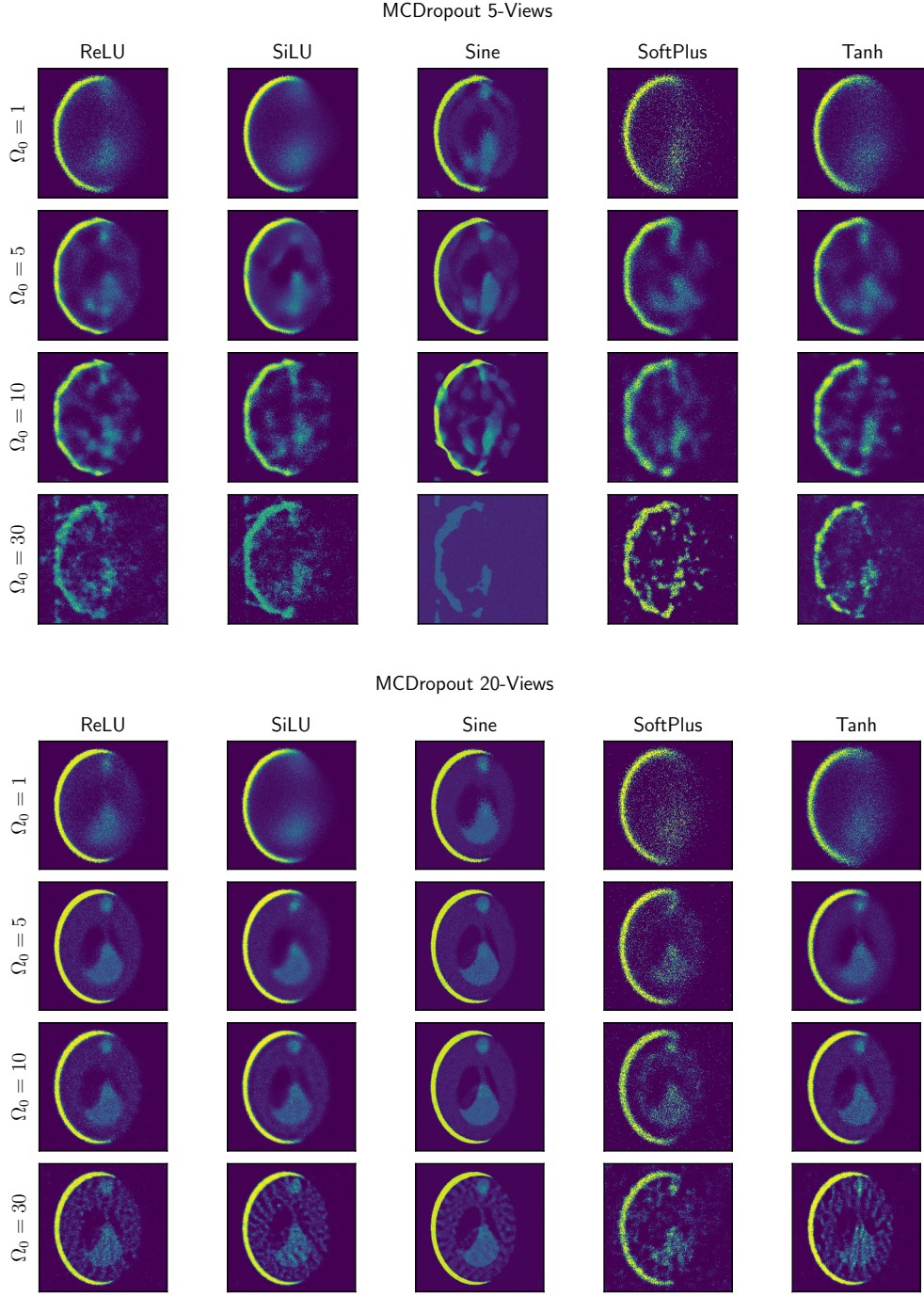

Figure 17: MCD image reconstruction, in both the 5- and 20-view cases, for each activation function and RFF frequency $\Omega_0$ value. Note that in the 5-view case, $\Omega_0 = 5$ enables the network to learn low-frequency image features, without many artifacts. Smaller $\Omega_0$ causes the network to produce overly simple output, whereas larger $\Omega_0$ induces high-frequency artifacts. In the 20-view case, similar observations are made, but the optimal $\Omega_0 = 10$. In this case, the reconstruction has higher frequencies without significant artifacts, for most activation functions.

### F.6 Comparing the effects of RFF frequency for DEs and MCD

In Section 4, we saw that MCD is an effective approach to obtain calibrated UQ in INRs, compared to DEs. This is a surprising observation, particularly in relation to the typical belief, in BDL settings (e.g. image classification), that deep ensembles are a very strong or state-of-the-art baseline (Ovadia et al., 2019).

While we do not have a full justification for this observation, we suspect that the underperformance of DEs (relative to MCD), in the context of UncertaINR, is related to the capacity that each individual ensemble member has. In particular, we observed that the RFF frequency $\Omega_0$ is a very important hyperparameter. This frequency is related to the 'capacity' of the model, in the sense that higher $\Omega_0$ implies ability to learn higher frequency details of the image, as shown in Fig. 10 of Tancik et al. (2020).

In Fig. 18, we plot the effect of RFF frequency $\Omega_0$ for different numbers of sinogram views (5 and 20), for both MCD and DEs. For 20-view Shepp-Logan data, we observe that an ensemble (blue) achieves roughly the same PSNR at both $\Omega_0 = 8$ and $\Omega_0 = 40$ (top right). However, the NLL at $\Omega_0 = 40$ is significantly better (lower) than the lower capacity $\Omega_0 = 8$ (bottom right). This indicates that the models with higher RFF frequencies generate more diverse, but still accurate, predictions and hence obtain better uncertainty estimates. On the other hand, MCD introduces an alternative effective way of encouraging diverse yet accurate predictions through different dropout seeds at a set dropout rate $p$. Thus, dropout combined with RFF encodings allows the individual models to have sufficient capacity to produce diverse yet accurate predictions. In contrast, deep ensembles rely solely on the RFF encodings to regulate the model capacity (and thus, diversity in predictions). This is also corroborated in Fig. 18 by the fact that MCD achieves better NLL (on Shepp-Logan phantoms) at lower $\Omega_0$ (i.e. lower capacity) than deep ensembles.

A more formal understanding of the settings in which deep ensembles are effective, as well as interactions with MCD, is important future work both for UncertaINR and BDL generally.

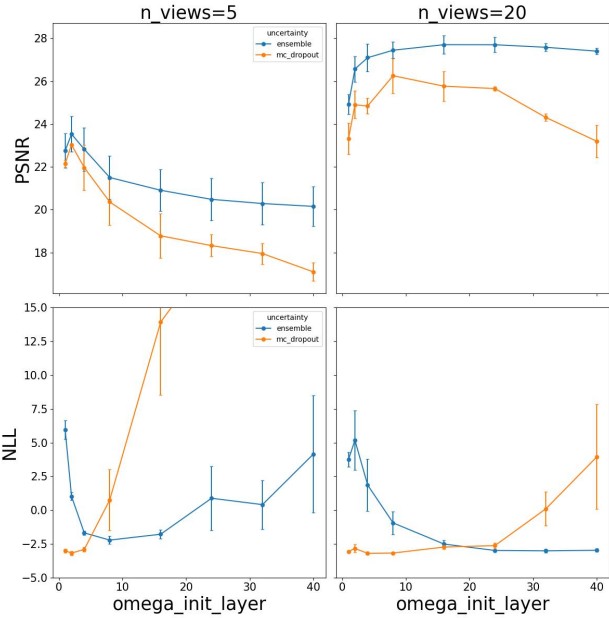

Figure 18: The effect of changing RFF frequency $\Omega_0$ for PSNR and NLL with MC dropout and Deep Ensembles.

### F.6.1 BBB Hyperparameter Analysis

The BBB model selection analysis is presented in a similar fashion to that of MCD. The main differences to MCD are that, in the BBB grid search, metrics are averaged over only the first two validation set images; the width and RFF search spaces are reduced; and the uncertainty parameters are the KL factor and Gaussian prior standard deviation (not probability of dropout). Figure 19 shows boxplots of average PSNR as a function of different hyperparameter values. Unlike MCD, model performance is extremely consistent across activation functions for every hyperparameter, except for width. We note that Tanh has some variation for RFF $\Omega_0$, following trends similar to those discussed in the case of MCD. Overall, SiLU performs the best, with ReLU and Tanh following closely behind. Interestingly, Sine performs the worst, failing to reach even 20db. However, greater performance consistency across hyperparameter configurations comes at the price of weaker top-performing models, which achieve lower PSNR values than those of MCD. Because the image sets are not identical, these comparisons across approaches should be taken with some reservation.

To understand why there is so much variation in BBB performance as a function of width, consider Figure 20, where the average PSNR obtained for each width is plotted as a function of prior standard deviation. It can be seen that the PSNR values are extremely consistent for each network width, across prior standard deviations. From a Bayesian perspective, the fact that the prior does not affect inference suggests that the latter is dominated by the model likelihood. However, mean performance decreases as a function of model width, indicating that the model becomes increasingly misspecified for larger widths. Given the Gaussian assumptions made in the variational inference specifications of BBB, this suggests that the true posterior distribution becomes less Gaussian as network width increases, and the variational approximation deteriorates. When combined with the low performance of BBB relative to other uncertainty quantification methods, reported in Table 1, this indicates that BBB may not be well suited for uncertainty quantification of INRs in the medical imaging context.

Furthermore, the overall BBB ablation results appear to mimic Coker et al. (2022)'s observation of mean-field BNN performance degradation with increased width. In all, BBB's poor performance is unsurprising and well-established in the BDL literature (Ovadia et al., 2019) (here BBB=SVI).

Table 5: Top 10 performing MCD models and their performances, for the 5-view case (**Top**) and 20-view case (**Bottom**). Models are ranked by PSNR, but NLL and ECE are also reported.

| Rank | Activation | Depth | Width | RFF $\Omega_0$ | PD | W. Decay | PSNR | NLL | ECE |
|---|---|---|---|---|---|---|---|---|---|
| 1 | Sine | 4 | 800 | 2 | 0.4 | 0.001 | 26.15 | -1.437 | 0.122 |
| 2 | Sine | 3 | 600 | 4 | 0.4 | 0.157 | 25.79 | -1.799 | 0.008 |
| 3 | Sine | 3 | 600 | 8 | 0.7 | 0.366 | 25.76 | -1.579 | 0.009 |
| 4 | Sine | 3 | 700 | 2 | 0.4 | 4.46E-4 | 25.75 | -1.533 | 0.117 |
| 5 | Sine | 4 | 700 | 3 | 0.5 | 9.36E-4 | 25.72 | -1.390 | 0.011 |
| 6 | Sine | 3 | 500 | 5 | 0.7 | 0.077 | 25.63 | 2.622 | 0.161 |
| 7 | Sine | 5 | 700 | 3 | 0.6 | 0.012 | 25.62 | -1.149 | 0.123 |
| 8 | Sine | 3 | 500 | 5 | 0.7 | 0.068 | 25.62 | -1.643 | 0.007 |
| 9 | SiLU | 8 | 300 | 3 | 0.2 | 2.68E-7 | 25.62 | 0.219 | 0.389 |
| 10 | Sine | 3 | 500 | 4 | 0.7 | 0.170 | 25.59 | -1.79 | 0.083 |

| Rank | Activation | Depth | Width | RFF $\Omega_0$ | PD | W. Decay | PSNR | NLL | ECE |
|---|---|---|---|---|---|---|---|---|---|
| 1 | Sine | 3 | 400 | 9 | 0.4 | 2.06E-5 | 33.74 | 0.701 | 0.134 |
| 2 | Sine | 4 | 300 | 12 | 0.4 | 2.63E-7 | 33.41 | 1.076 | 0.149 |
| 3 | Sine | 3 | 500 | 8 | 0.5 | 0.006 | 33.37 | -0.502 | 0.137 |
| 4 | Sine | 5 | 500 | 11 | 0.4 | 3.74E-6 | 33.29 | -0.288 | 0.137 |
| 5 | Sine | 6 | 400 | 10 | 0.2 | 5.85E-6 | 33.28 | 4.654 | 0.152 |
| 6 | Sine | 6 | 500 | 8 | 0.2 | 1.117E-4 | 33.24 | 2.622 | 0.161 |
| 7 | Sine | 4 | 400 | 9 | 0.5 | 0.001 | 33.21 | -0.798 | 0.143 |
| 8 | Sine | 3 | 500 | 10 | 0.5 | 2.53E-4 | 33.20 | -0.716 | 0.129 |
| 9 | Sine | 5 | 500 | 11 | 0.2 | 7.87E-7 | 33.18 | 1.973 | 0.163 |
| 10 | Sine | 4 | 500 | 8 | 0.5 | 4.56E-5 | 33.17 | -1.257 | 0.114 |

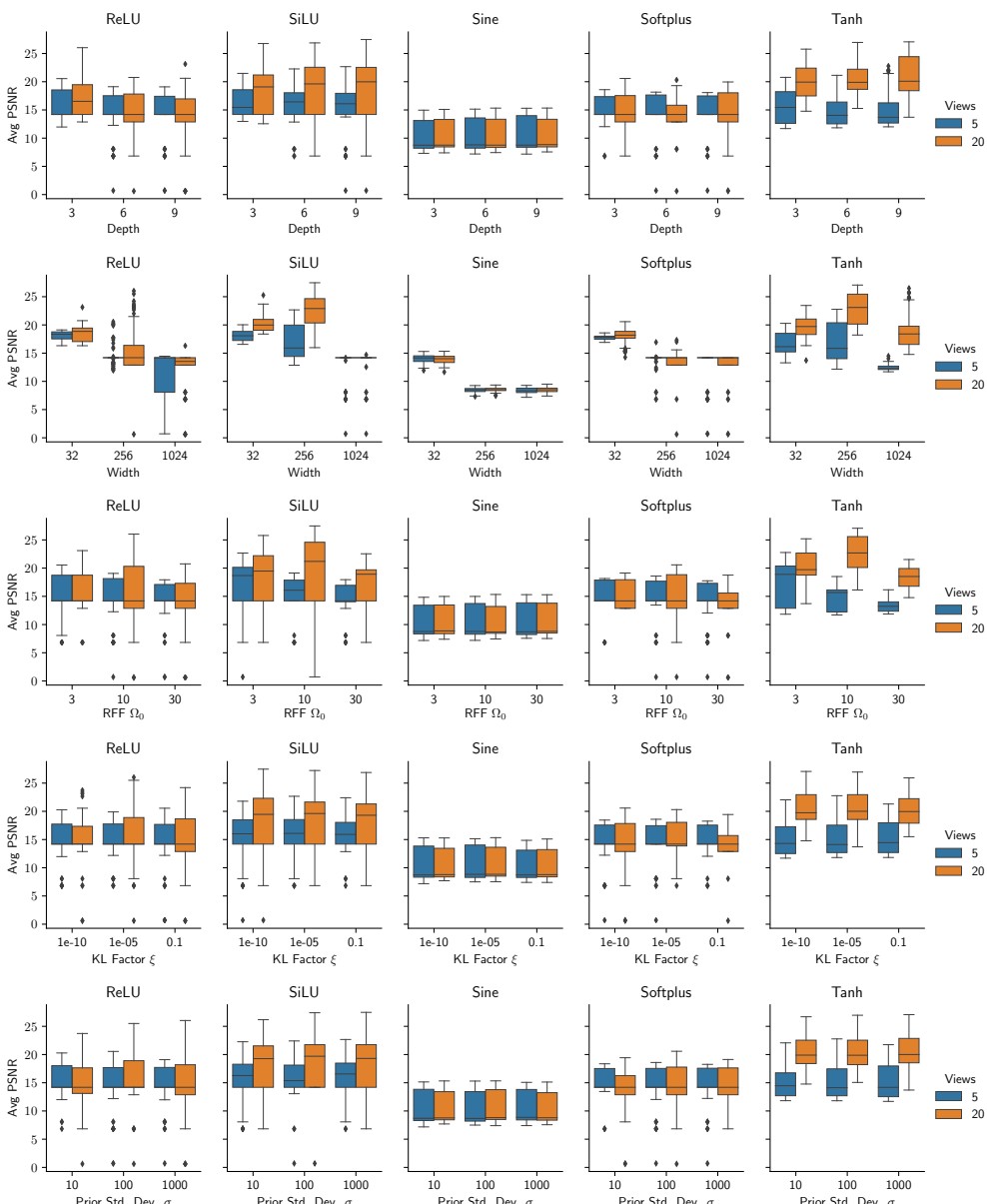

Figure 19: Boxplots of the average PSNR of BBB models trained in the coarse grid search hyperparameter sweep, for both 5 and 20 views. Each column corresponds to a different activation function and each row to a sweep over one of the remaining hyperparameters - depth, width, RFF $\Omega_0$, KL factor $\xi$, and prior standard deviation $\sigma$.

### F.6.2 DE Performance Analysis

As described in Appendix F.4.2, DEs of $M$ base learners were created by combining the top-$M$ performing NNs, according to average PSNR. As reported in Table 1, the best MCD model outperforms the best BBB model significantly, with at least a 2dB increase in PSNR, as well as reduced NLL and ECE, for both the 5- and 20-view cases. Thus, we ensembled the top MCD models produced by the second fine Bayesian hyperparameter sweeps of Section F.6.1. The parameterizations and performance of the 10 best performing models for both the 5- and 20-view cases are listed in Table 5. Note that, in both cases, the model architectures vary greatly across models. While the Sine activation function is fairly consistent, the remaining parameters vary greatly. For example, in the 5-view case, model depths range from 3 to 8, widths

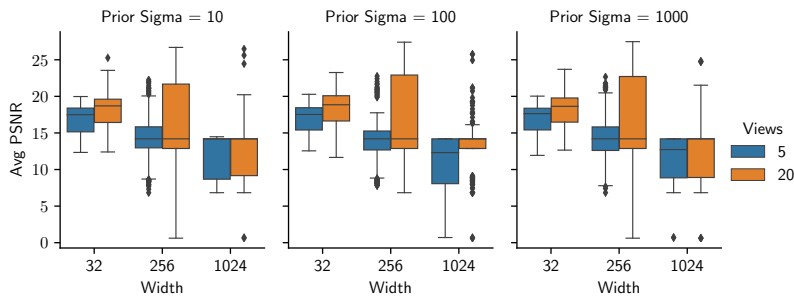

Figure 20: Boxplots of average PSNR of BBB models of varying width for each value of prior standard deviation $\sigma$, for both 5- and 20-view cases.

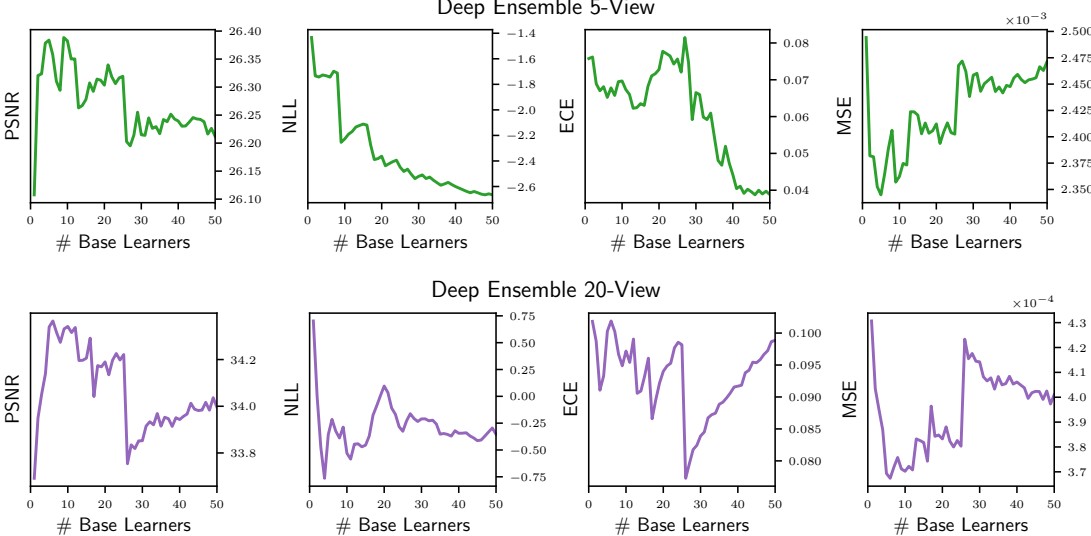

Figure 21: PSNR, NLL, ECE, and MSE plotted as a function of the number of base-learners used in DEs, for both the 5- and 20-view cases. Note that each added base learner performs slightly worse in terms of image reconstruction quality than the network preceding it.

from 300 to 800, RFF $\Omega_0$ from 2 to 8, and probability of dropout (PD) from 0.2 to 0.7, all with a variety of weight decays. These variations increase base learner diversity well beyond different weight values.

DE combines varied base learners to improve uncertainty calibration. In principle, if all base learners achieved the same PSNR, model performance should only increase (or plateau) and variance should only decrease (or plateau) as more base-learners are added. In our case, however, each new added base learner had a slightly lower PSNR on the validation set. To verify how this affected model performance and calibration, we generated plots of each metric as a function of # of DE base learners. As shown in Figure 21, both PSNR and MSE improve significantly as the first base learners are added to the ensemble, but begin to worsen for larger ensembles. Considering that models of worse PSNR are being added with each increase in # of base learners, it is unsurprising that performance eventually degrades. However, ensembling never reduces performance below that of using the single best model. NLL and ECE are more sensitive to the number of base learners. NLL demonstrates the overall best performance gain as a function of baselearners. ECE, however, does not change consistently with DE size, with large ensembles sometimes even harming performance relative to the single best model. In all, it is clear that ensembling improves model performance and calibration. However, larger ensembles have no gains over smaller ensembles and are far more computationally expensive. Thus, for the final results, presented in Table 1, only ensembles of sizes 2, 5, and 10 are considered.

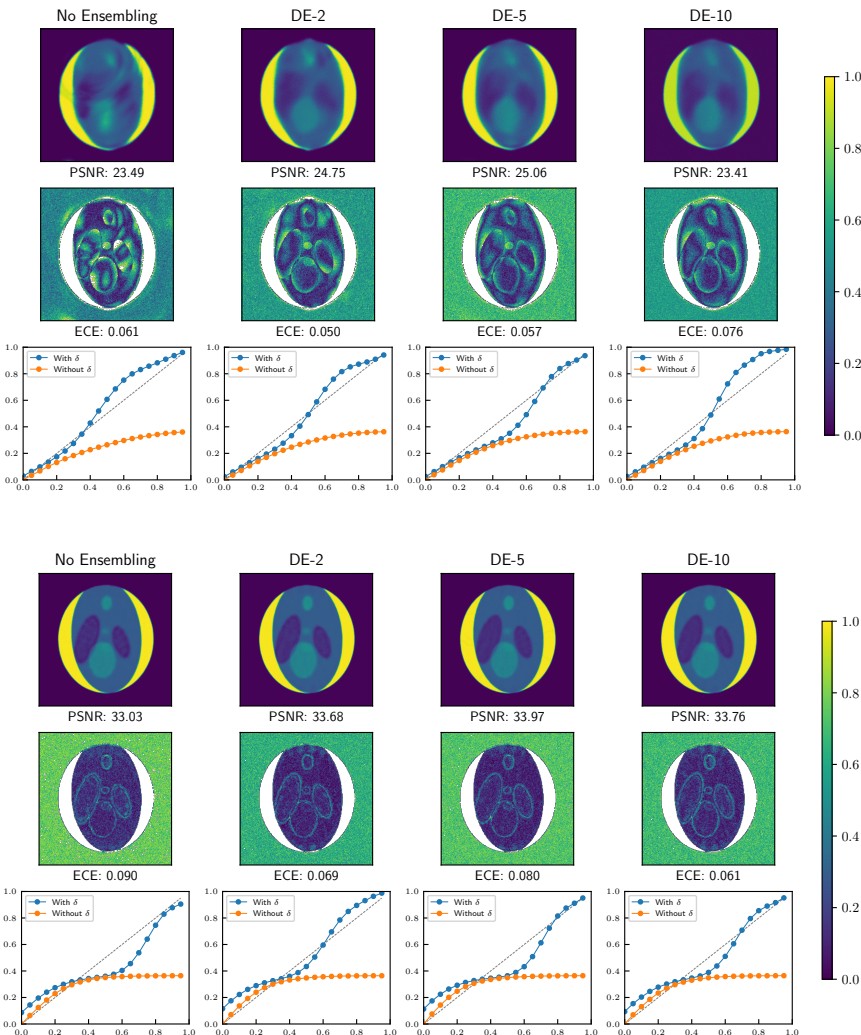

Figure 22: Image reconstruction and calibration performance of the final DE models on test set Image #3 for 5-views (top) and 20-views (bottom). Different columns show the results of different ensemble sizes, ranging from 1 to 10. Top row shows the reconstructed image, middle row the pixelwise coverage, and bottom row the reliability curve.

Figure 22 illustrates image reconstruction performance for the different ensemble types on test set Image #3. In the 5-view case, DEs achieve impressive performance improvements, with a PSNR increase of over 1.5dB for DE-5 and an ECE reduction of 0.011 for DE-2. In the 20-view case the baseline is much better performing. Hence, although there are gains in PSNR, these are not very noticeable in the reconstructed image. However, the improvements in calibration are larger, with an ECE drop of 0.2 for DE-2 and a clear improvement in the image reliability curve.

## F.7 Final Results Conclusions

The primary goal of this study was to understand the relative performance of different uncertainty quantification techniques for UncertaINR, on the Shepp-Logan dataset. Table 1, in the main text, presents metrics assessing the best-performing MCD, DE-2 MCD, DE-5 MCD, and DE-10 MCD UncertaINRs. These methods consistently outperformed the classical reconstruction techniques in terms of image quality, while producing reasonably well calibrated uncertainty estimates. BBB was found to be the worst performing uncertainty quantification approach, generally producing the poorest calibrated uncertainty estimates and worse image

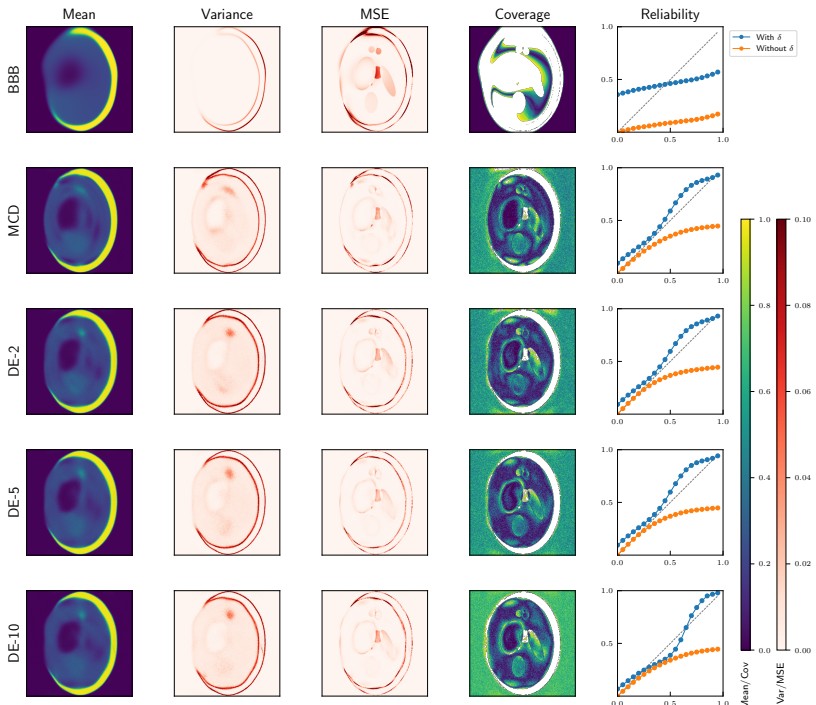

Figure 23: Validation results of all approaches for 5-view Shepp-Logan data. From left to right: mean image reconstruction, variance, MSE, coverage, and reliability diagram.

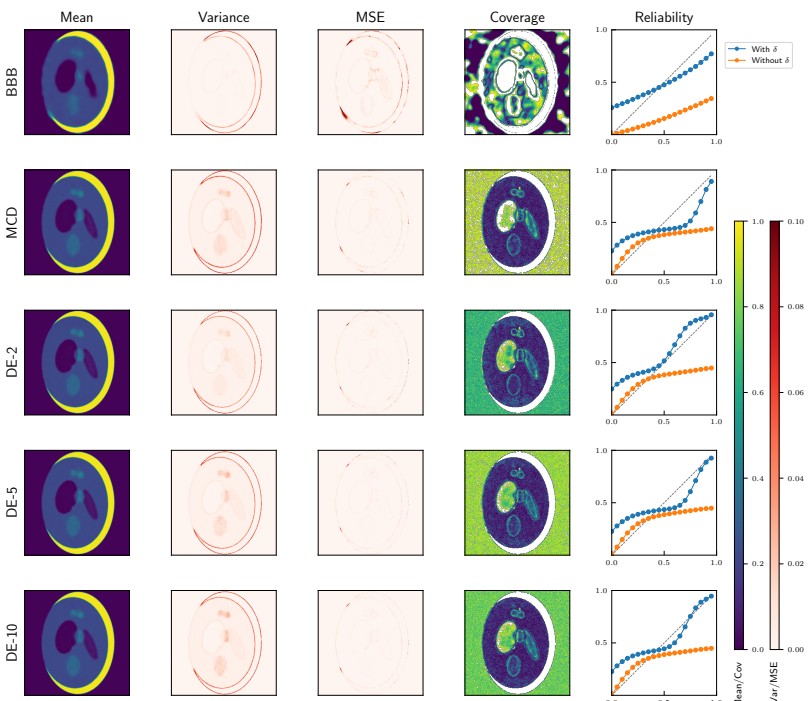

Figure 24: Validation results of all approaches for 20-view Shepp-Logan data. From left to right: mean image reconstruction, variance, MSE, coverage, and reliability diagram.

Table 6: **Ablation Study Differences**: Test set results from Table 1 are presented again, in terms of differences from the average metric value for each image. Specifically, the table reports the average and standard error of these differences across the 5 test images for each metric.

| Reconstruction Type | 5-View Test Set Differences | | | 20-View Test Set Differences | | |
|---|---|---|---|---|---|---|
| | PSNR (↑) | NLL (↓) | ECE (↓) | PSNR (↑) | NLL (↓) | ECE (↓) |
| FBP | $-14.862 \pm 0.239$ | – | – | $-13.203 \pm 0.168$ | – | – |
| CGLS | $-5.397 \pm 0.201$ | – | – | $-8.086 \pm 0.272$ | – | – |
| EM | $-0.136 \pm 0.093$ | – | – | $0.199 \pm 0.364$ | – | – |
| SART | $-0.270 \pm 0.167$ | – | – | $1.536 \pm 0.260$ | – | – |
| SIRT | $-0.270 \pm 0.167$ | – | – | $1.533 \pm 0.261$ | – | – |
| BBB UINR | $2.504 \pm 0.378$ | $1.524 \pm 0.383$ | $0.105 \pm 0.003$ | $-0.749 \pm 0.152$ | $0.782 \pm 2.137$ | $0.013 \pm 0.008$ |
| MCD UINR | $4.437 \pm 0.244$ | $-0.186 \pm 0.094$ | $-0.015 \pm 0.005$ | $4.175 \pm 0.218$ | $0.329 \pm 0.532$ | $0.007 \pm 0.012$ |
| DE-2 MCD UINR | $4.473 \pm 0.267$ | $-0.389 \pm 0.063$ | $-0.028 \pm 0.006$ | $4.529 \pm 0.276$ | $-0.153 \pm 0.580$ | $-0.005 \pm 0.011$ |
| DE-5 MCD UINR | $4.866 \pm 0.402$ | $-0.366 \pm 0.070$ | $-0.032 \pm 0.006$ | $5.109 \pm 0.329$ | $-0.405 \pm 0.511$ | $-0.005 \pm 0.006$ |
| DE-10 MCD UINR | $4.655 \pm 0.303$ | $-0.583 \pm 0.262$ | $-0.030 \pm 0.006$ | $4.957 \pm 0.144$ | $-0.554 \pm 0.519$ | $-0.010 \pm 0.011$ |

reconstruction than classical techniques in the 20-view regime. MCD consistently outperformed classical approaches and was generally better calibrated than BBB. Ensembling over MCD base learners, however, was the most successful approach, outperforming the best classical approach by ∼4dB in the 5-view case and ∼3dB in the 20-view case, as well as achieving the lowest overall NLL and ECE values.

Given the varying difficulty of images in the Shepp-Logan test set, we did not believe that reporting standard deviations in Table 1 would accurately reflect reconstruction variability. Instead, Table 6 presents the average and standard error over the difference from the mean metric value for each image. Calculating the difference from mean for each image reduces the effect of variability in image difficulty. Notice that the differences are significant with respect to the standard error across distinct methods, although not as significant across ensemble sizes for the MCD UINRs.

One further remark regarding Table 1 is that, although the classical reconstruction procedures did not use the validation set images as a validation set (since no hyperparameters were tuned), image reconstruction quality still deteriorated in the test set. This indicates that the test set data is actually more challenging than the validation set. Given that UncertaINR performance did not significantly decline on the test set, we have strong reason to believe that none of the UncertaINR approaches over-fitted to the validation set.

Figures 23 and 24 visually illustrate the difference in the uncertainty quantification of the different methods, for the 5- and 20-view cases respectively. Beginning with the 5-view case, the mean predicted image was fairly blurry for all methods, only capturing low-frequency image components and exhibiting high uncertainty surrounding edges. Furthermore, note that all reliability curves are very well calibrated in this 5-view case (after a $\delta$-selection adjustment), except for BBB. Similar trends are present in the 20-view case, but it is visually harder to distinguish differences in the image output due to the higher quality of all reconstructions.

## G   UncertaINR AAPM Performance Assessment

### G.1   Dataset

In order to assess the performance of UncertaINR in a realistic setting, we used data from the American Association of Physicists in Medicine (AAPM) and Mayo Clinic 2016 Low-Dose CT grand challenge (McCollough et al., 2017). Specifically, we use the 8 reconstructed images used as a test set in the CoIL work (Sun et al., 2021), shown in Figure 25, so as to directly compare to their results. These ground truth images were reconstructed from real CT scan measurement data.

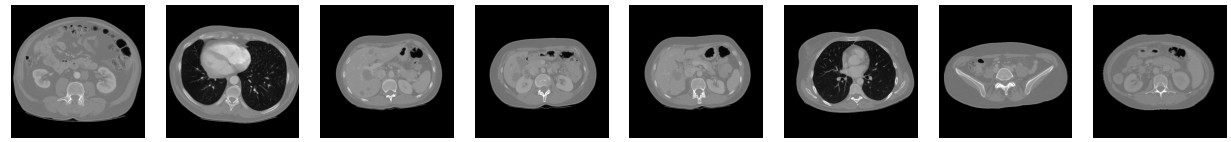

Figure 25: The 8 AAPM-Mayo test set images used to assess UncertaINR performance.

In order to make the dataset more realistic and compare to the results presented in CoIL, noise was added to the image sinograms before being input to the models. This artificial data generation pipeline is illustrated in Figure 26. More specifically, uniformly distributed Gaussian white noise was added to the sinogram, so as to achieve a desired SNR relative to the original, noise-less sinogram. In this work, we chose to present results for a noise level achieving sinograms of 40dB SNR.

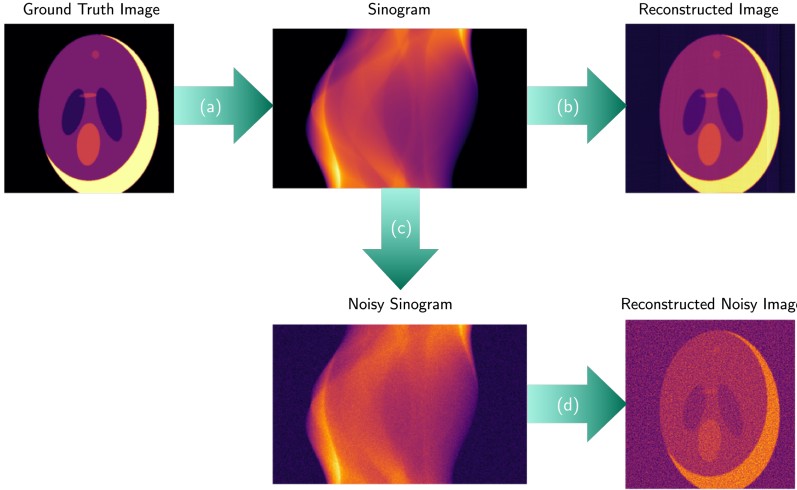

Figure 26: A flowchart of the artificial data generation pipeline used in this work. **(a)** The Radon transform is used to generate a sinogram, corresponding to measurement data, from the ground truth data. **(b)** In the noiseless case, this sinogram would be directly used to train our model. Note that even though the sinogram is noiseless, the reconstruction is not perfect. **(c)** In a more realistic scenario, Gaussian white noise is added to the sinogram. **(d)** This noisy sinogram data is then fed into the model, which produces a reconstructed image of the ground truth, of lower quality that produced in the noiseless case.

### G.2   Final Model Hyperparameters

Given the increased computational costs in running UncertaINR on the AAPM-Mayo dataset, it was not feasible to perform large-scale hyperparameter sweeps, like those performed in the ablation study presented in Appendix F. However, coarse searches were performed for each hyperparameter and new training approaches were used in order to achieve the competitively performing models presented in Table 2. For reproducibility, we provide the final hyperparameters and training procedures.

Beyond training insights learned from the hyperparameter study, further improvements were made to UncertaINR training in order to achieve competitive reconstruction accuracy on the higher-resolution, higher-frequency, and noisy Mayo-AAPM data. Specifically, images were zero-padded to ensure that all image content was contained in the projections when calculating the Radon transform. All UncertaINR were optimized using Adam (not AdamW, to eliminate the computational costs of tuning the weight decay parameter) and we found that longer training times were needed (on the order of 15,000 epochs). If that many training epochs cannot be used for some reason, we found that stochastic weight averaging (Izmailov et al., 2018) can be used to improve reconstruction accuracy by a few decibel for UncertaINRs trained with few epochs. With the increased number of training iterations, the Sine activation proved too unstable and we found SiLU to be the best performing activation function, with Tanh a close second. Noting the significance of RFFs from the ablation study, we found it crucial that the width of the RFF layer be at least as large as the padded image width. Finally, before adding a regularization term to the UncertaINR loss, our models were not competitive with state-of-the-art reconstruction techniques. While adding an isotropic TV regularization,

$$\tilde{T}_{\mathrm{ISO}}(f) = \sum_{x,y \in \mathcal{X} \times \mathcal{Y}} (f(x+1, y) - f(x,y))^2 + (f(x, y+1) - f(x,y))^2,$$

boosted performance by a few decibel, the anisotropic approximation of the TV regularizer,

$$\tilde{T}_{\mathrm{ANISO}}(f) = \sum_{x,y \in \mathcal{X} \times \mathcal{Y}} |f(x+1, y) - f(x,y)| + |f(x, y+1) - f(x,y)|,$$

achieved better reconstruction accuracy, especially in the low-measurement (60-view) regime. (at the slight cost of uncertainty calibration). The relative performance of these two regularizers relative to no TV regularizer are presented for both GOP and MCD UINR in Table 7. While we do not have a justification for why the anisotropic approximation performs better than isotropic for reconstruction accuracy, we believe that regularization/prior exploration would be interesting future work.

Table 7: The affect of different regularizers on image reconstruction performance.

| Reconstruction Method | Regularization Type | 60-View SNR | 120-View SNR |
|---|---|---|---|
| GOP | None | 18.33 | 18.85 |
| | Isotropic | 23.62 | 25.60 |
| | Anisotropic | 25.97 | 27.40 |
| MCD Uinr | None | 25.75 | 27.35 |
| | Isotropic | 25.78 | 28.06 |
| | Anisotropic | 27.38 | 28.65 |

Given all these training insights, the hyperparameters of the top-performing (MCD and HMC) UINRs, reported in Table 2, are presented in Table 8.

Table 8: Hyperparameters of the top-performing MCD UINRs, HMC UINRs, and INRs reported in Table 2. Additional HMC-specific hyperparameters, such as burn-in periods, are provided in Appendix G.2.1.

| Model | # Views | Act. Func. | Depth | Width | RFF $\Omega_0$ | Reg Type | Reg Coeff | $p$(Dropout) | # Epochs |
|---|---|---|---|---|---|---|---|---|---|
| INR | 60 | SiLU | 5 | 280 | 48 | Aniso | 0.05 | 0 | 15,000 |
| MCD UINR | 60 | SiLU | 5 | 280 | 35 | Aniso | 0.03 | 0.2 | 15,000 |
| HMC UINR | 60 | SiLU | 4 | 200 | 48 | Aniso | 0.03 | N/A | N/A |
| INR | 120 | SiLU | 5 | 280 | 60 | Aniso | 0.05 | 0 | 15,000 |
| MCD UINR | 120 | SiLU | 5 | 280 | 40 | Aniso | 0.008 | 0.2 | 15,000 |
| HMC UINR | 120 | SiLU | 4 | 200 | 60 | Aniso | 0.03 | N/A | N/A |

### G.2.1 HMC-specific hyperparameters

As mentioned in Section 3, all our HMC experiments used the NUTS (Hoffman et al., 2014) implementation in NumPyro (Phan et al., 2019). All HMC runs ran for 1000 samples and had a burn-in period of 500

samples, with a thinning factor of 5, so that every fifth post burn-in sample was used at evaluation time. We initialized the model parameters from a trained model for both GOP and UINRs (corresponding to models in the *GOP* and *INR* rows of Table 2 respectively) to enable faster mixing. For NUTS specific hyperparameters, we set the *max_tree_depth* to 8 and target a Metropolis-Hastings acceptance rate of 0.8. During the burn-in period, NUTS adaptively sets the step-size and (diagonal) mass matrix hyperparameters which are crucial to the effectiveness of HMC; more details can be found in Hoffman et al. (2014).

**HMC-GOP hyperparameters** For 60 and 120 views we set the TV regularisation strength to 0.5 and 0.3 respectively. Both values were tuned on a small grid of hyperparameter values.

**HMC-UNIR hyperparameters** We set the parameter variance, $\tau^2$ in Eq. 8, to $\frac{1}{\sqrt{\gamma \times \text{width}}}$, with $\gamma = 0.2$ and 0.15 respectively for 60 and 120 views. $\gamma$, and the regularisation coefficients in Table 8, were tuned on a small grid of values due to the high computational cost of HMC. Due to the non-convex nature of NN parameter space, we aggregated the samples from two independent chain runs (from two independently trained MAP initializations) for HMC-UINR, the maximum number of chains that fit into memory on a 24GB VRAM Titan RTX GPU.

### G.3 Results Analysis

Given the varying difficulty of images in the AAPM test set, we did not believe that reporting standard deviations in Table 2 would accurately reflect variability. Instead, Table 9 presents the average and standard error over the difference from the mean metric value for each image. Calculating the difference from mean for each image reduces the effect of variability in image difficulty. Notice that, generally speaking, our conclusions from Table 2 regarding the different methods (e.g. MCD provides more calibrated UQ than ensembling and our INR-based methods outperform their classical counterparts) are statistically significant relative to standard error across different images.

Table 9: **AAPM Differences**: AAPM results from Table 2 are presented again, in terms of differences from the average metric value for each image. Specifically, the table reports the average and standard error of these differences across the 8 test images for each metric.

| RECONSTRUCTION | 60-VIEWS | | | 120-VIEWS | | |
|---|---|---|---|---|---|---|
| METHOD | SNR (↑) | NLL (↓) | ECE (↓) | SNR (↑) | NLL (↓) | ECE (↓) |
| FBP | $-13.46 \pm 0.46$ | – | – | $-11.38 \pm 0.49$ | – | – |
| EM | $-9.57 \pm 0.82$ | – | – | $-9.94 \pm 0.37$ | – | – |
| CGLS | $-3.96 \pm 0.22$ | – | – | $-3.55 \pm 0.26$ | – | – |
| SIRT | $-3.15 \pm 0.22$ | – | – | $-3.81 \pm 0.40$ | – | – |
| SART | $-2.50 \pm 0.21$ | – | – | $-3.72 \pm 0.19$ | – | – |
| GOP-TV | $1.93 \pm 0.10$ | – | – | $1.91 \pm 0.07$ | – | – |
| HMC GOP-TV | $1.06 \pm 0.13$ | $-4.70 \pm 0.93$ | $-0.004 \pm 0.011$ | $1.33 \pm 0.12$ | $-4.60 \pm 0.81$ | $-0.007 \pm 0.010$ |
| INR | $3.21 \pm 0.09$ | – | – | $3.32 \pm 0.10$ | – | – |
| DE-2 UINR | $3.25 \pm 0.09$ | $22.45 \pm 3.41$ | $0.118 \pm 0.004$ | $3.34 \pm 0.10$ | $17.62 \pm 2.30$ | $0.114 \pm 0.004$ |
| DE-5 UINR | $3.26 \pm 0.10$ | $6.32 \pm 0.93$ | $0.070 \pm 0.006$ | $3.34 \pm 0.09$ | $7.57 \pm 1.29$ | $0.075 \pm 0.007$ |
| DE-10 UINR | $3.25 \pm 0.10$ | $4.78 \pm 0.93$ | $0.038 \pm 0.010$ | $3.33 \pm 0.09$ | $5.11 \pm 1.20$ | $0.054 \pm 0.008$ |
| MCD UINR | $3.34 \pm 0.11$ | $-5.55 \pm 0.72$ | $-0.029 \pm 0.005$ | $3.14 \pm 0.13$ | $-4.98 \pm 0.61$ | $-0.040 \pm 0.006$ |
| DE-2 MCD UINR | $3.41 \pm 0.11$ | $-5.68 \pm 0.77$ | $-0.044 \pm 0.004$ | $3.20 \pm 0.13$ | $-5.05 \pm 0.64$ | $-0.052 \pm 0.006$ |
| DE-5 MCD UINR | $3.44 \pm 0.11$ | $-5.76 \pm 0.80$ | $-0.055 \pm 0.005$ | $3.22 \pm 0.13$ | $-5.11 \pm 0.66$ | $-0.065 \pm 0.007$ |
| DE-10 MCD UINR | $3.42 \pm 0.10$ | $-5.79 \pm 0.81$ | $-0.061 \pm 0.004$ | $3.25 \pm 0.13$ | $-5.32 \pm 0.77$ | $-0.055 \pm 0.005$ |
| HMC UINR | $3.06 \pm 0.09$ | $-6.07 \pm 0.93$ | $-0.033 \pm 0.010$ | $3.01 \pm 0.11$ | $-5.25 \pm 0.75$ | $-0.023 \pm 0.008$ |

### G.4 Observed Cold Posterior Type Effect

We observe in Table 2 that there is a decrease in SNR performance of HMC GOP-TV relative to its deterministic counterpart GOP-TV (25.10 vs 25.97 for 60 views), the latter corresponding to the maximum-a-posterior (MAP) point estimate for the posterior, Eq. 6, from which we seek to sample.

This evokes similarities with the recently observed "cold-posterior" effect (CPE) in BDL (Wenzel et al., 2020). In this setting, one considers a tempered posterior $p_{\mathcal{T}}(f|S) \propto \exp\left(\frac{-U(f)}{\mathcal{T}}\right)$, where the Bayes' posterior corresponds to temperature $\mathcal{T} = 1$ and "cold" posteriors ($\mathcal{T} < 1$) place more weight on parameter regions

of high posterior probability. In the cold limit $\mathcal{T} \to 0_+$, the tempered posterior becomes exactly the MAP point estimate.

For context, the tempered version of Eq. 6 for GOP yields:

$$p_\mathcal{T}(f|S) \propto \exp\left\{-\frac{1}{2\sigma^2\mathcal{T}} \sum_{i=1}^{|\Phi \times \mathcal{R}|} \left(S_i - \mathbf{A}_i f\right)^2 - \frac{\lambda}{\mathcal{T}} T(f)\right\}, \tag{78}$$

The CPE is the observation that colder posteriors ($\mathcal{T} < 1$) outperforms the standard Bayes' posterior ($\mathcal{T} = 1$) in terms of generalization, and understanding its causes has been a topic of recent study in BDL (Wenzel et al., 2020; Adlam et al., 2020; Aitchison, 2020; Fortuin et al., 2022; Wilson & Izmailov, 2020; Izmailov et al., 2021; Noci et al., 2021; Nabarro et al., 2022).

Inspired by this line of work, as well as our observation that HMC GOP-TV underperforms MAP GOP-TV (point estimate) in terms of SNR in Table 2, we studied the effect of tempering posteriors in GOP with HMC in 27. We find a similar effect to the cold-posterior whereby SNR performance is improved at cold temperatures, and even allows tempered GOP-HMC to outperform its deterministic MAP counterpart. However, we also observe that the colder temperatures are detrimental to the metrics which consider UQ: ECE (which is purely about UQ) and NLL (which considers both UQ and reconstruction accuracy), presumably because the tempered sample only samples around the MAP estimate with little diversity in samples.

There are three commonly presumed causes for the cold-posterior effect (Noci et al., 2021): data-curation; data-augmentation; and prior mispecification. Given that our observation model (Eq. 5) is constructed by design, and that we use noisy data exactly matching our likelihood model, we rule out data-curation (proposed by Aitchison (2020) who argue that the CPE is caused by the fact that standard DL datasets, e.g. CIFAR-10, are well-curated and that standard likelihoods like cross-entropy do not account for this) as the cause for our observed effect in 27. Likewise, our CT reconstruction setting is very different to standard DL supervised learning and there is no natural analogue for data-augmentation. This leaves prior mispecification, in the form of the TV-regularization term in Eq. 78, as a potential cause for our observed cold-posterior like effect in CT reconstruction.

We note that the grid-of-pixels model uses a simple grid-based pixel-by-pixel dimension array of parameters without any mention of NNs, and enjoys a convex log-posterior compared to the more complex non-convex parameter loss landscapes that exist in (B)DL. In this sense, to the best of our knowledge 27 represents the first suggestion that a cold-posterior effect can occur outside of NN-based models.

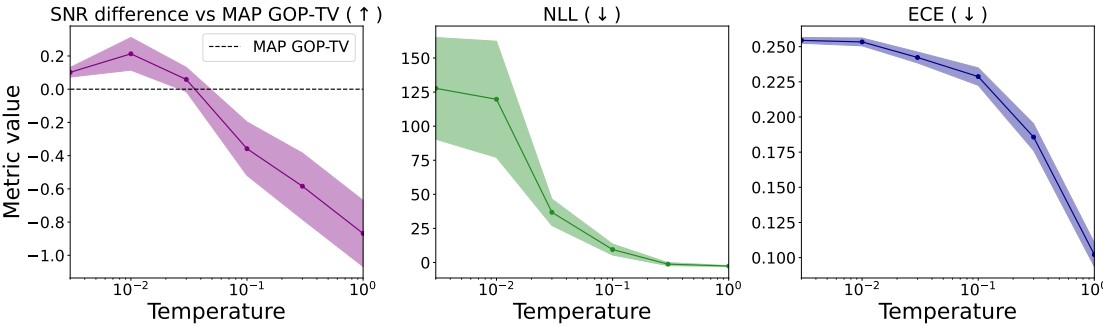

Figure 27: A cold-posterior like effect observed for GOP-TV with HMC for 60-views AAPM.

## G.5 UncertaINR AAPM Test Set Outputs

In our main figure of results, Figure 3b, we present the model outputs and metrics for only 1 out of the 8 AAPM test set images, as illustrated in Figure 25. We conclude the paper by presenting the figures for all

test set images. Note that the observed trends and analysis, as presented in main paper Section 4.3, hold for the remaining, following test set image results.

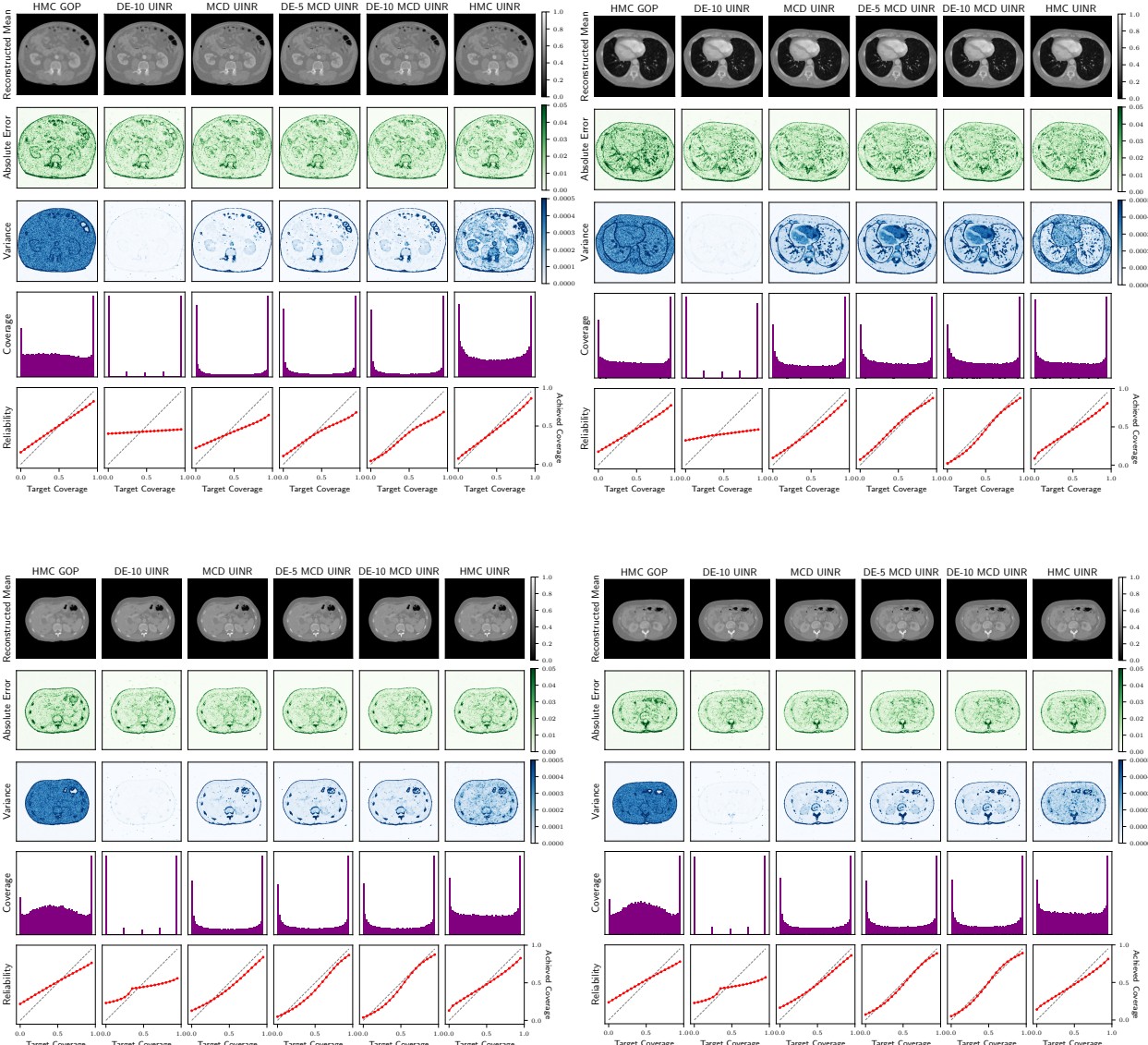

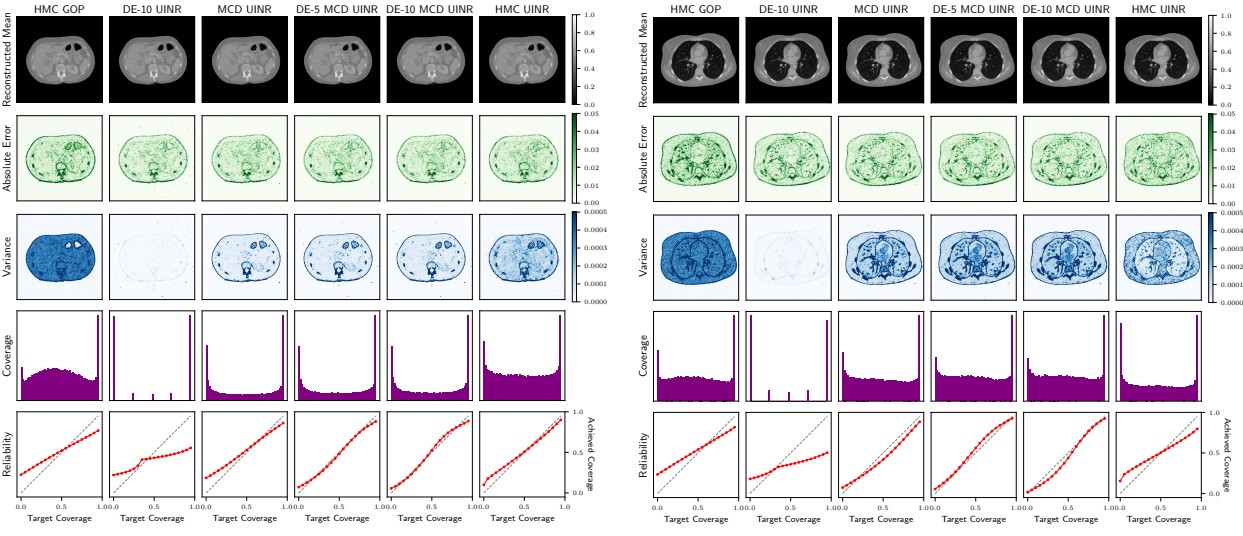

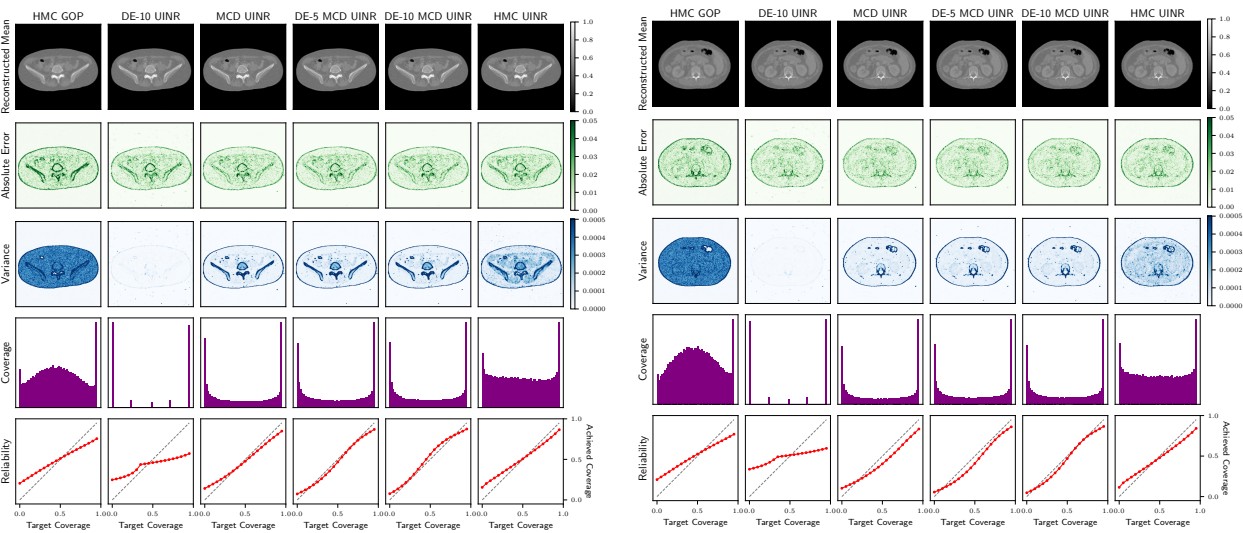

