# OpenReview forum: "UncertaINR: Uncertainty Quantification of End-to-End Implicit Neural Representations for Computed Tomography"
_TMLR — Accepted by TMLR_

### Review · Reviewer_98YR · 2022-10-30

**Summary Of Contributions:**

The paper performs an extensive empirical evaluation of uncertainty quantification  (UQ) methods for implicit neural representations for CT inverse problems. The paper compares the following UQ methods

-  monte carlo dropout
- Hamiltonian monte carl
- deep ensembles
- bayes by backpropr


which are evaluated on 1 artificial and 1 real world dataset. The paper perform extensive ablation studies on the artificial dataset and compares against, classical and deep learning baselines, evaluating reconstruction quality as well as calibration of the UQ and finding that their method compares favourably against the baselines, finding in particular that DE and MCD are pareto-strong methods in terms of computational effort,reconstruction quality  and UQ calibration

**Audience:**

Yes

**Broader Impact Concerns:**

The use of UQ methods in medical imaging could  increase the trust medical personal places in these methods, which given that while MCD and DE are impressively correlated with error, we still lack (to my knowledge) theory and experience of the interplay between neural method, MCD/DE, data properties and the final UQ. While the work is an important step towards creating this experience, I think the limitations section I raised as missing in weaknesses also warrants a statement about the context in which this work should be placed w.r.t use in real world usage (i.e., make an assessment and official recommendation on whether these results and these methods should be applied in practice and what precautions would be recommended/what future work should be done to avoid over-optimism and potential harm caused by using these methods to speed up CT scans).

**Claims And Evidence:**

Yes

**Requested Changes:**

Please address the  weaknesses I outlined above, also I believe you meant to write theta and not W in appendix E, right below eq. 62? Otherwise there might be something missing somewhere since I do not see W reappear in that section.



**Strengths And Weaknesses:**

Strengths:

- very well written, I was unfamiliar with CT inverse problems before but the paper gave enough context to follow along and to confirm things with google searches
- extensive and rigorous ablation study and overall evaluation in main body and appendix
- touches almost all relevant points one might be interesting in in such a study ....
- code and data are provided and readable

Weaknesses

- with the exception of an easy runtime and energy comparison (figure 7 shows view dependency, but I would like seeing convergence plots or at least approximate times to convergence of each method, total computational overhead as well as *ideally* energy consumption/carbon impact quantified in the paper. Carbon and energy are optional, but I think adding the other elements to the paper would improve the evaluation)
- include the standard deviations across results tables please? especially since all activation functions studies appear to have significant std
- there is no limitations section in main body or appendix or discussion of limitations in the conclusions, which despite the papers thorough investigation appears odd to me (small dataset size, only a single natural dataset, a collection of unanswered questions raised throughout the work would all benefit the consumers of this work in my opinion)

---

> ### Author Response · Authors · 2022-12-30
> **Response to Reviewer 98YR**
>
> We are glad to hear that the reviewer found the work to be “very well written”, “extensive”, “rigorous”, and “interesting”. We further thank the reviewer for their helpful suggestions and feedback. In particular, we will now describe how we have addressed each of the listed weaknesses and requested changes, to further improve the manuscript and study. (Note: we have uploaded a new version of the manuscript, with modified/added text in red.)
>
> #### **Convergence Times:**
> A more detailed comparison of runtimes is now provided in Appendix Section D.5. In particular, as recommended by the reviewer, we added a table of the convergence times of the different methods (Table 4).
>
> #### **Standard Deviations:**
> To address concerns about variability in the reported Shepp-Logan and AAPM results, we have added Tables 6 and 9 (Appendices F.7 and G.3) res. For each metric and each method these tables present the relative difference between: said method on an individual image versus the mean over different methods for that same image. Tables 6 and 9 present the mean and standard errors of these relative differences, where the mean and standard errors are taken over the different images in our test set. We see that, generally speaking, our conclusions from Table 2 regarding the different methods (e.g. MCD provides more calibrated UQ than ensembling and our INR-based methods outperform their classical counterparts) are statistically significant across different images. The reason we choose to display standard errors in *relative differences* is that there is a large amount of variability between different images in our test set, and studying the relative differences allows us to account for this.
>
> #### **Limitations Section:**
> In light of the reviewer feedback, we have added a ``*Limitations, Broader Impact, and Future Work*'' section to the main paper (Section 5).
>
> #### **Typo Correction:**
> The reviewer was correct in noting a typo below Equation 62, which we have corrected to say $\Theta$ instead of $W$. Thank you for the correction.

---

### Review · Reviewer_yF4H · 2022-12-06

**Summary Of Contributions:**

The paper presents a novel method, UncertaINR, based on implicit neural representations (INR), which achieves well-calibrated uncertainty estimates without sacrificing reconstruction quality relative to other classical, INR-based, and CNN-based reconstruction techniques on realistic, noisy, and underdetermined data.

**Audience:**

Yes

**Claims And Evidence:**

Yes

**Requested Changes:**

More discussion is required on the priors. E.g., GOP vs. zero-mean Gaussian prior or any other potential priors.

Can you also clarify on why TV is commonly used in medical imaging. It be helpful for readers not familiar with medical-imagining.

It is quite interesting to see MC-based approach ourperform deep ensembles. It would be great to have a separate discussion on this aspect.

Computational cost of each of the compared approaches is also missing. Can authors provide them?

**Strengths And Weaknesses:**

Strengths:
- The proposed approach is simple and could benefit medical imaging community working in inverse problem.
- The presentation and flow of the paper is great.
- The experimental design is clear, easy to follow and demonstrate the strength of the proposed UncertaINR.

Weakness:
- The proposed approach is based on INRs and the extension looks minimal. There could be different ways to extend INRs (choice of priors, choices in architectures, etc.). The paper doesn't seem to acknowledge or discuss these aspects.
- The paper is missing some critical discussions.

---

> ### Author Response · Authors · 2022-12-30
> **Response to Reviewer yF4H**
>
> We are glad that the reviewer found the paper “presentation and flow great” and “the experimental design clear, easy to follow, and demonstrative of the strength of the proposed UncertaINR”. We agree with the reviewer that UncertaINR “could benefit the medical imaging community”. We further thank the reviewer for their helpful suggestions and feedback. In particular, we will now describe how we have addressed each of the listed weaknesses and requested changes, to further improve the manuscript and study. (Note: we have uploaded a new version of the manuscript, with modified/added text in red.)
>
> #### **Extension Looks Minimal:**
> We respectfully disagree that our work is a minimal extension to the existing INR literature or that we fail to acknowledge/discuss these extensions in the paper. As mentioned in the introduction (Section 1 paragraph 2), all existing INR reconstruction approaches are focused on reconstruction accuracy, whereas our focus is on the low-measurement CT reconstruction setting, where uncertainty quantification (UQ) is crucial. This leads to several considerations in our work that differ from and extend the existing INR literature. First, it is not clear how to define the problem of UQ in INRs, as existing INRs produce a single point estimate for the NN parameters. We pose the INR training problem as one of Bayesian inference (as described in Section 3), which enables UQ by inferring the posterior distribution over INR parameters. The Bayesian formulation also enables a richer treatment of the problem than those of the existing INR literature. For example, the adoption of TV regularization (discussed in Section 3) as an extended function space prior leads to improved reconstruction accuracy and uncertainty. We have added Table 5 to Appendix G.2 to concretely demonstrate this.
>
> In terms of extending INRs themselves, we considered various architectural modifications in terms of model depth, width, and activation function in our ablations, as detailed in Appendix F. As just mentioned, our formulation led to the adoption of TV regularization as a function space prior for INRs, an approach not previously considered in INRs.
>
> #### **More Discussion on Priors:**
> We have updated Section 3 in an attempt to clarify our discussion of the priors and Grid-of-Pixels baseline. To reiterate the key ideas, the Grid-of-Pixels (GOP) prior is a probabilistic extension to the classical grid-based methods (described in Section 2.1.1), where each pixel in the image is treated as a ‘parameter’ with value inferred from the observed sinogram data. On the other hand, the zero-mean Gaussian prior (described in Section 3)  treats both the popular weight decay regularizer and also INR NN initialization (e.g. the He initialization - https://arxiv.org/abs/1502.01852) as inducing a Gaussian prior distribution over NN parameters, implicitly defining a prior over the INR functional output. Zero-mean Gaussian priors are commonly used in the field of Bayesian Deep Learning (see main paper citation: Fortuin et al. 2022 ``*Bayesian neural network priors revisited.*''), to constrain the values of parameters to sensible values and also lead to well-known connections to Gaussian Processes priors in wide NN function outputs (Neal, 1996; Lee et al. 2018; Matthews et al. 2018).
>
> #### **Clarification on TV:**
> The TV regularizer removes unwanted image noise and artifacts, while preserving important details such as edges. This is particularly important for medical images because edges typically correspond to structures that are important for medical diagnosis, such as the boundaries of tissues or bones, and must be rendered sharply. In the paper, we reference Rudin et al. 1992, which is the original proposal for TV regularization. We have updated Appendix G.2, with Table 5, to demonstrate how TV regularization substantially improves image reconstruction performance for both GOP and MCD UINR. We have also updated the main text  (Section 3) to reference this table.
>
> #### **MCD vs DE Discussion:**
> Like the reviewer, we were surprised to find Monte Carlo Dropout experimentally outperformed deep ensembles. At this point, we do not claim to have a theoretical justification for this result. Nevertheless, we offer a brief discussion in the main paper (Section 4.3 paragraph 5) and Appendix F.6. To summarize, we hypothesize that the relatively poor performance of DE relative to MCD may be due to the model capacity of individual ensemble members, which is dictated through the RFF encoding frequency. However, a deeper understanding of this finding remains an interesting topic for future work, as noted in our newly added “*Limitations, Broader Impact, and Future Work*” (Section 5).
>
> #### **Convergence Times:**
> We thank the reviewer for this helpful suggestion and have added a more thorough discussion of convergence times in Appendix Section D.5, including a table of convergence times for the different methods (Table 4).

---

### Review · Reviewer_RRpV · 2022-12-13

**Summary Of Contributions:**

This work presents a first study on the calibration performance of of implicit neural representations for image reconstruction. To this end, authors reformulate the INRs from a bayesian perspective, which allows to quantify the uncertainty on the predictions of the proposed model. For evaluation purposes, two CT datasets are employed, and several existing methods are included in their comparison.

Please find below my detailed comments for this work.


**Audience:**

Yes

**Broader Impact Concerns:**

No concerns.

**Claims And Evidence:**

Yes

**Requested Changes:**

Please see the list of weaknesses for the potential changes to improve the manuscript.

**Strengths And Weaknesses:**


Strengths:

- Interesting task. Despite the technical contribution is limited, assessing the calibration performance of image reconstruction is an interesting and important topic. Calibration of discriminative models is gaining popularity in discriminative models. Nevertheless, to my knowledge, improving the uncertainty estimation on image reconstruction remains unexplored.

- Literature review is properly conducted. Existing works in relevant topics are properly discussed, which better motivates the goal of this work, i.e., addressing the calibration performance of uncertainty quantification of implicit neural representation models in image reconstruction.

- The empirical results obtained by the different proposed methods outperform classical reconstruction methods. As stated by the authors, deep learning based methods (Table 2) outperform the proposed approach but they require more data. Nevertheless, I have some concerns on these results that I will discuss in the weaknesses section.


Weaknesses:

- I think that extending the proposed method to other modalities would better highlight its generability. Furthermore, having only one artificial and one real benchmark could be suboptimal to showcase the superiority of the presented approach.

- The work of Reed et al (Dynamic reconstruction from limited views with implicit neural presentation and parametric motion models) should be included in the experimental section, as to my understanding they also reconstruct CT images from a limited number of samples. Indeed, authors in that work also resort to the artificial Shepp-Logan dataset. Thus, I strongly recommend the authors to include this work in their empirical validation. As a note, please correct the reference, as this is an ICCV’21 paper, not arxiv anymore.

- I stress that image reconstruction is not my application domain, but I wonder how useful/reliable 5 and 8 testing images are to draw conclusions about any method.

- Table 2 reports the performance of different methods when 60-views and 120-views are used for measurement regime. I wonder whether this means that the methods are trained with these number of images (i.e,. 60 or 120). If yes, I would argue that the deep learning methods achieve better reconstruction performance on the real dataset. Please clarify whether this is the case (i.e., the number of views is the number of training images across all the methods)

- In both Table 1 and 2, the different versions of the proposed method outperforms classical approaches. Nevertheless, it seems that there is no a clear trend among all these methods (particularly for the real dataset, Table 2). For example, HMC UINR obtains the best NLL performance, while its ECE values largely degrade compared to the ensemble with 10 models (or very similar with MCD UINR). If authors would have to suggest which method to promote for researches wanting to continue in this topic, what would be their suggestion?

- I feel that the supplemental material can be shortened substantially. There are many sections that are well-known or can be found in existing publications/books. Thus, I do not see how they help to the manuscript (beyond avoiding the reader having to search for this content).

---

> ### Author Response · Authors · 2022-12-30
> **Response to Reviewer RRpV (Part I)**
>
> We thank the reviewer for their helpful suggestions and feedback. We agree with the reviewer that the task at hand is “interesting”, “important”, and “well-motivated”. As the reviewer notes, there has been little prior work on uncertainty quantification in image reconstruction. We are glad that the reviewer finds the task “well-motivated”, the literature review “properly conducted”, and that our method outperforms classical methods. We will now respond to each of the reviewer’s concerns. (Note: we have uploaded a new version of the manuscript, with modified/added text in red.)
>
> #### **Extension to Other Modalities:**
> We agree that the proposed UncertaINR method is promising, outperforming classical methods on artificial and real benchmark datasets. In this work, we restricted our focus to the well-motivated medical application domain of computed tomography (CT). Our main goal was advancing performance in the low-measurement regime, where uncertainty quantification (UQ)  has critical importance. Namely, we wanted to understand whether INR reconstructions could achieve calibrated UQ and study the effectiveness of various forms of UQ in the INR setting. Note that there are no other INR approaches that provide uncertainty quantification, irrespective of the dataset.
>
> As the reviewer notes, we expect our results to generalize to other domains, such as 3D/4D settings or magnetic resonance imaging (MRI) modalities. However, these different domains have distinct mechanisms and structures, which would require modifications to the UINR architecture. For example, MRI measurements are related to image space by a Fourier rather than Radon transform. Therefore, we leave these extensions as future work, as described in our newly added Section 5: “*Limitations, Broader Impact, and Future Work*”.
>
> #### **Datasets:**
> Regarding datasets, we restricted experiments to the computed tomography setting because the need for uncertainty quantification is well established in this setting and there are also widely used reconstruction datasets. We adopted two of the most popular among these: the artificial Shepp-Logan dataset and the AAPM Grand Challenge datasets. The Shepp-Logan dataset was used in our preliminary ablation study, while our main experiments were performed on the AAPM dataset – in order to enable fair comparisons to CoIL (Sun et al. 2021), which is the most relevant prior work. In the CoIL work, all models are evaluated on a set of 8 images (taken from two different patients) in the AAPM dataset, with artificial noise introduced. We therefore evaluated our method on the same 8 images at a noise level of 40dB SNR.
>
>
> #### **Reed et al. Work Citation:**
> We have updated the ``*Dynamic CT Reconstruction from Limited Views with Implicit Neural Representations and Parametric Motion Fields*'’ citation from arXiv to ICCV, as recommended by the reviewer. We mention this work in our literature review on INRs (Section 2.2), rather than the experimental section. While that work also focuses on CT image reconstruction via INRs, it does not consider uncertainty quantification and is specifically focused on 4D motion estimation.
>
> When the motion estimation component of the Reed et al. work is removed, their base INR model formulation is equivalent to ours. Thus, our current results effectively include a characterization of the Reed et al. methodology, plus our extensions which provide uncertainty estimates. We agree that extending uncertainty quantification to the 4D setting is an interesting direction for future work and have included it in our newly added Section 5: “*Limitations, Broader Impact, and Future Work*”.

---

> > ### Author Response · Authors · 2022-12-30
> > **Response to Reviewer RRpV (Part II)**
> >
> > #### **Number of Test Images:**
> > The reviewer questions whether 5-8 test images are useful/reliable for drawing conclusions about our methods. Thus, we have added Tables 6 and 9 (Appendices F.7 and G.3), reporting average and standard error difference from mean, to demonstrate how our results present a significant relative performance trend between methods.
> >
> > It is also important to note this number of images was not arbitrary, but chosen based on the following considerations. The large ablation study, performed via the Shepp-Logan dataset, involved sweeping over thousands of network architectures  and was primarily used to inform preliminary design choices. This would have been unfeasible with a large set of validation images. Our primary assessment of UncertaINR, via the AAPM dataset, leveraged the exact 8 images used in the CoIL paper (Sun et al. 2021) in order to enable a fair comparison to their method, which is the most relevant work in the field. As can be seen in Appendix G.1 Figure 25, these 8 images are very diverse (taken from two patients, possessing substantially different structures and frequency components), resulting in varying SNRs across the images. Because of this diversity, we believe this dataset does a good job of testing model performance. Furthermore, for our more computationally intensive methods (like HMC), which take 8+ hours to run on GPUs, it was computationally prohibitive to assess on a significantly larger test set. We do, however, note the small dataset size and use of singular real-world dataset as limitations of the work in our newly added “*Limitations, Broader Impact, and Future Work*” Section 5.
> >
> > #### **Measurements vs Training Images:**
> > In direct response to the reviewer’s concern, in each experiment there is only one sinogram training image, which consists of 60 or 120 view angles. To expand upon this, across the paper, we use measurements to refer to the direct radiographic output of a CT scan, which involve X-rays sent through the patient at a specific view angle. A set of these measurements comprises a sinogram image (as illustrated in main paper Figure 1b). Mathematically we denote the sinogram measurements as $S_\phi (r)$, where each $\phi$ denotes a distinct measurement view angle. When we say 60- or 120-views, we are referring to the number of X-ray measurements taken in order to generate the sinogram. The sinogram is the singular “training image” required to learn the INR parameter weights that encode the reconstructed image function. As described in Section 4.2 (Model Comparison), no further training data is needed. Only a handful (<6) of validation images were used to select a good INR architecture, which was then used to reconstruct each image in the dataset.
> >
> > We hope this clarifies our claim that UINR requires substantially less training data than deep learning methods and why we disagree with the claim that “deep learning methods achieve better reconstruction performance on the real dataset”. In fact, we believe the lack of training data is a major advantage of UncertaINR, especially in data-sparse medical settings, over deep learning methods – which require a large number of training images. Additionally, UINR has the following important advantages: 1) uncertainty quantification and 2) superior performance in the low-view regime, which is the ideal regime for medical imaging (fewer sinogram views translate into shorter CT scan times and less patient exposure to harmful radiation).
> >
> > #### **Suggestions:**
> > As the reviewer notes, no single UINR method dominates across the board. As mentioned, HMC achieves decent NLL, but relatively poor ECE compared to other methods. Meanwhile MCD achieves the best reconstruction performance, competitive (if not the best) calibration in terms of both NLL and ECE, and fast convergence times. Therefore, we would single out MCD as the overall top performer.
> >
> > One of the main contributions of this work is, in fact, the surprising finding that MCD performs so well in INR reconstruction and calibration, despite its simplicity and low computational overhead, which is also contrary to common wisdom in the Bayesian DL literature. We try to offer some intuition for this surprising finding in Section 4.3 and Appendices F.6 and G. However, in our newly added Section 5: “*Limitations, Broader Impact, and Future Work*”, we emphasize that it is interesting future work for the Bayesian deep learning community to understand why INR uncertainty quantification appears to deviate from more conventional settings. We have also added a sentence to the manuscript abstract to emphasize that this is an interesting finding of the work.

---

> > > ### Author Response · Authors · 2022-12-30
> > > **Response to Reviewer RRpV (Part III)**
> > >
> > > #### **Supplement:**
> > > We thank the reviewer for their feedback and will try to shorten the supplement. We believe that the material contained in the appendix is a useful, self-contained reference for those unfamiliar with the CT reconstruction literature. Readers already familiar with the literature can simply skip those sections. However, we agree that there is important information we do not want to get lost in an overly large appendix. Therefore, if there are any specific sections that the reviewer believes can/should be removed, please let us know.

---

### Comment · Action_Editors · 2022-12-14
**Discussion phase**

Now that we have three complete reviews, I would ask each of the reviewers to take a look at the other reviews and see if you are in agreement.  We now have a 2 week discussion phase between reviewers and authors.  Looking forward to a productive discussion.

---

> ### Author Response · Authors · 2023-02-06
> **Updates?**
>
> We appreciate the time and feedback of our reviewers. We were wondering if there are any updates or feedback on our rebuttal (posted roughly a month ago)? Thank you very much!

---

> > ### Comment · Action_Editors · 2023-02-06
> > **Review process update**
> >
> > There was indeed a delay getting all decision recommendations, but this is now resolved.  The process is now with me to enter a decision in the coming days.  Thanks for your patience.

---

### Decision · Action_Editors · 2023-02-07

**Recommendation:** Accept with minor revision

**Comment:**

All three reviewers assess the paper as "leaning accept" or "accept."

Reviewer yF4H had no concrete points of improvement in the assessment, and Reviewer 98YR had only positive comments.

Reviewer RRpV felt that some of the appendices are unnecessary and could be cut.  I am more neutral about this point.  I am marking the paper as "Accept with minor revision" but I am also satisfied if the authors feel strongly that they would like to keep the appendices included.  This recommendation should be viewed by the authors as indicating that whether or not to make the recommended change is at their discretion, as the article would in any case pass the threshold for acceptance to TMLR in my opinion.

**Audience:**

Quoting from Reviewer 98YR: " I think especially medical the medical ML community, but also for other INR practitioners who desire UQ, this work would be of great interest."  I agree in the assessment that it is sufficiently of interest to some members of the TMLR audience.

**Claims And Evidence:**

The main claims are benchmarking of calibrated uncertainty quantification in INR models, incorporating also total variation regularization.  Table 1 performs an ablation study, Table 2 a performance assessment, and Figure 3 calibration plots.  Limitations on the claims are appropriately laid out in Section 5.  Relation to previous work is well documented and extensive appendices outline background information, technical details, and extended results.

---

> ### Author Response · Authors · 2023-04-03
> **Camera-Ready Uploaded**
>
> We thank all the reviewers for the reviews, feedback, and decision! For the camera-ready version of the paper, we have updated the manuscript to the final format. We have also created a clean code repository, hosted on Github (https://github.com/bobby-he/uncertainr). Finally, we recorded a talk about the paper, which is hosted on Youtube (https://youtu.be/cD7Wx4F_EjQ).